# Certified Minimax Unlearning with Generalization Rates and Deletion Capacity

**Jiaqi Liu[1], Jian Lou[1,2,†], Zhan Qin[1,†], Kui Ren[1]**
[1]Zhejiang University
[2]ZJU-Hangzhou Global Scientific and Technological Innovation Center
`{jiaqi.liu, jian.lou, qinzhan, kuiren}@zju.edu.cn`
[†]Corresponding authors

## Abstract

We study the problem of $(\epsilon, \delta)$-certified machine unlearning for minimax models. Most of the existing works focus on unlearning from standard statistical learning models that have a single variable and their unlearning steps hinge on the *direct Hessian-based conventional Newton* update. We develop a new $(\epsilon, \delta)$-certified machine unlearning algorithm for minimax models. It proposes a minimax unlearning step consisting of a *total Hessian-based complete Newton* update and the Gaussian mechanism borrowed from differential privacy. To obtain the unlearning certification, our method injects calibrated Gaussian noises by carefully analyzing the "sensitivity" of the minimax unlearning step (i.e., the closeness between the minimax unlearning variables and the retraining-from-scratch variables). We derive the generalization rates in terms of population strong and weak primal-dual risk for three different cases of loss functions, i.e., (strongly-)convex-(strongly-)concave losses. We also provide the deletion capacity to guarantee that a desired population risk can be maintained as long as the number of deleted samples does not exceed the derived amount. With training samples $n$ and model dimension $d$, it yields the order $\mathcal{O}(n/d^{1/4})$, which shows a strict gap over the baseline method of differentially private minimax learning that has $\mathcal{O}(n/d^{1/2})$. In addition, our rates of generalization and deletion capacity match the state-of-the-art results derived previously for standard statistical learning models.

## 1 Introduction

Minimax models have been widely applied in a variety of machine learning applications, including generative adversarial networks [Goodfellow et al., 2014, Arjovsky et al., 2017], adversarially robust learning [Madry et al., 2018, Sinha et al., 2018], and reinforcement learning [Du et al., 2017, Dai et al., 2018]. This is largely credited to the two-variable (i.e., primal and dual variables) model structure of minimax models, which is versatile enough to accommodate such diverse instantiations. As is common in machine learning practice, training a successful minimax model relies crucially on a potentially large corpus of training samples that are contributed by users. This raises privacy concerns for minimax models. Unlike standard statistical learning (STL) models, the privacy studies for minimax models are relatively newer. Most of the existing studies focus on privacy protection during the training phase under the differential privacy (DP) notion [Dwork et al., 2006] and federated minimax learning settings [Sharma et al., 2022]. Recent works in this direction have successfully achieved several optimal generalization performances measured in terms of the population primal-

Table 1: *Summary of Results. Here (S)C means (strongly-)convex loss function, and (S)C-(S)C means (strongly-)convex-(strongly-)concave loss function, PD means Primal-Dual. $n$ is the number of training samples and $d$ is the model dimension.*

| Model | Unlearning Algorithm | Setting | Generalization Measure | Deletion Capacity |
|---|---|---|---|---|
| STL | DP-based [Bassily et al., 2019] | C | Population Excess Risk [Sekhari et al., 2021] | $\mathcal{O}(n/d^{1/2})$ |
| | [Sekhari et al., 2021] | (S)C | | $\mathcal{O}(n/d^{1/4})$ |
| Minimax Learning | DP-based [Zhang et al., 2022a] | SC-SC | Population Strong PD Risk | $\mathcal{O}(n/d^{1/2})$ |
| | DP-based [Bassily et al., 2023] | C-C | | |
| | **Our Work** | (S)C-(S)C | Population Weak or Strong PD Risk | $\mathcal{O}(n/d^{1/4})$ |

dual (PD) risk for DP minimax models specifically [Yang et al., 2022, Zhang et al., 2022a, Bassily et al., 2023, Boob and Guzmán, 2023].

Machine unlearning is an emerging privacy-respecting problem concerning already-trained models (i.e., during the post-training phase) [Cao and Yang, 2015, Guo et al., 2020, Sekhari et al., 2021, Graves et al., 2021, Bourtoule et al., 2021, Li et al., 2021, Shibata et al., 2021, Wu et al., 2022, Cheng et al., 2023, Chen et al., 2023, Tarun et al., 2023, Wu et al., 2023, Ghazi et al., 2023, Wang et al., 2023b]. That is, it removes certain training samples from the trained model upon their users' data deletion requests. It is driven by the right to be forgotten, which is mandated by a growing number of user data protection legislations enacted in recent years. Prominent examples include the European Union's General Data Protection Regulation (GDPR) [Mantelero, 2013], the California Consumer Privacy Act (CCPA), and Canada's proposed Consumer Privacy Protection Act (CPPA). Machine unlearning comes with several desiderata. Besides sufficiently removing the influence of the data being deleted, it should be efficient and avoid the prohibitive computational cost of the baseline method to fully retrain the model on the remaining dataset from scratch. To guarantee the sufficiency of data removal, there are exact machine unlearning methods [Cao and Yang, 2015, Ginart et al., 2019, Brophy and Lowd, 2021, Bourtoule et al., 2021, Ullah et al., 2021, Schelter et al., 2021, Chen et al., 2022b,a, Yan et al., 2022, Di et al., 2023, Xia et al., 2023] and approximate machine unlearning methods [Golatkar et al., 2020a, Wu et al., 2020, Golatkar et al., 2020b, Nguyen et al., 2020, Neel et al., 2021, Peste et al., 2021, Golatkar et al., 2021, Warnecke et al., 2023, Izzo et al., 2021, Mahadevan and Mathioudakis, 2021, Mehta et al., 2022, Zhang et al., 2022c, Wang et al., 2023a, Chien et al., 2023a, Lin et al., 2023] (some can offer the rigorous $(\epsilon, \delta)$-certification [Guo et al., 2020, Sekhari et al., 2021, Suriyakumar and Wilson, 2022, Chien et al., 2023b] inspired by differential privacy). In addition, recent studies also point out the importance of understanding the relationship between the generalization performance and the amount of deleted samples [Sekhari et al., 2021, Suriyakumar and Wilson, 2022]. In particular, they introduce the definition of deletion capacity to formally quantify the number of samples that can be deleted for the after-unlearning model to maintain a designated population risk. However, most existing works so far have focused on machine unlearning for standard statistical learning models with one variable, which leaves it unknown how to design a minimax unlearning method to meet all the desiderata above.

Machine unlearning for minimax models becomes a pressing problem because the trained minimax models also have a heavy reliance on the training data, while the users contributing data are granted the right to be forgotten. In this paper, we study the machine unlearning problem for minimax models under the $(\epsilon, \delta)$-certified machine unlearning framework. We collect in Table 1 the results in this paper and comparisons with baseline methods that are adapted from previous papers to $(\epsilon, \delta)$-certified machine unlearning.

Our main contributions can be summarized as follows.

- *Certified minimax unlearning algorithm:* We develop $(\epsilon, \delta)$-certified minimax unlearning algorithm under the setting of the strongly-convex-strongly-concave loss function. To sufficiently remove the data influence, the algorithm introduces the total Hessian consisting of both direct Hessian and indirect Hessian, where the latter is crucial to account for the inter-dependence between the primal and dual variables in minimax models. It leads to the complete Newton-based minimax

unlearning update. Subsequently, we introduce the Gaussian mechanism from DP to achieve the $(\epsilon, \delta)$-minimax unlearning certification, which requires careful analysis for the closeness between the complete Newton updated variables and the retraining-from-scratch variables.

- *Generalization:* We provide generalization results for our certified minimax unlearning algorithm in terms of the population weak and strong primal-dual risk, which is a common generalization measure for minimax models.

- *Deletion capacity:* We establish the deletion capacity result, which guarantees that our unlearning algorithm can retain the generalization rates for up to $\mathcal{O}(n/d^{1/4})$ deleted samples. It matches the state-of-the-art result under the STL unlearning setting that can be regarded as a special case of our minimax setting.

- *Extension to more general losses:* We extend the certified minimax unlearning to more general loss functions, including convex-concave, strongly-convex-concave, and convex-strongly-concave losses, and provide the corresponding $(\epsilon, \delta)$-certification, population primal-dual risk, and deletion capacity results.

- *Extension with better efficiency:* We develop a more computationally efficient extension, which can also support successive and online deletion requests. It saves the re-computation of the total Hessian matrix during the unlearning phase, where the minimax unlearning update can be regarded as a total Hessian-based infinitesimal jackknife. It also comes with slightly smaller population primal-dual risk though the overall rates of the risk and deletion capacity remain the same.

## 2   Related work

Machine unlearning receives increasing research attention in recent years, mainly due to the growing concerns about the privacy of user data that are utilized for machine learning model training. Since the earliest work by Cao and Yang [2015], a variety of machine unlearning methods have been proposed, which can be roughly divided into two categories: exact unlearning and approximate unlearning.

**Exact machine unlearning.** Methods for exact machine unlearning aim to produce models that perform identically to the models retrained from scratch. Some exact unlearning methods are designed for specific machine learning models like k-means clustering [Ginart et al., 2019] and random forests [Brophy and Lowd, 2021]. SISA [Bourtoule et al., 2021] proposes a general exact unlearning framework based on sharding and slicing the training data into multiple non-overlapping shards and training independently on each shard. During unlearning, SISA retrains only on the shards containing the data to be removed. GraphEraser [Chen et al., 2022b] and RecEraser [Chen et al., 2022a] further extend SISA to unlearning for graph neural networks and recommendation systems, respectively.

**Approximate machine unlearning.** Approximate machine unlearning methods propose to make a tradeoff between the exactness in data removal and computational/memory efficiency. Prior works propose diverse ways to update the model parameter and offer different types of unlearning certification. When it comes to the unlearning update, many existing works consider the Newton update-related unlearning step where the Hessian matrix of the loss function plays a key role [Guo et al., 2020, Golatkar et al., 2020a, Peste et al., 2021, Sekhari et al., 2021, Golatkar et al., 2021, Mahadevan and Mathioudakis, 2021, Suriyakumar and Wilson, 2022, Mehta et al., 2022, Chien et al., 2023b]. This unlearning update is motivated by influence functions [Koh and Liang, 2017]. In order to alleviate the computation of the Hessian, Golatkar et al. [2020a] and Peste et al. [2021] utilize Fisher Information Matrix to approximate the Hessian, mitigating its expensive computation and inversion. Mehta et al. [2022] provide a variant of conditional independence coefficient to select sufficient sets for unlearning, avoiding the need to invert the entire Hessian matrix. ML-forgetting [Golatkar et al., 2021] trains a linear weights set on the core dataset which would not change by standard training and a linear weights set on the user dataset containing data to be forgotten. They use an optimization problem to approximate the forgetting Newton update. Suriyakumar and Wilson [2022] leverage the proximal infinitesimal jackknife as the unlearning step in order to be applied to nonsmooth loss functions. In addition, they can achieve better computational efficiency and are capable of dealing with online delete requests. There are also many other designs achieving different degrees of speedup [Wu et al., 2020, Nguyen et al., 2020, Neel et al., 2021, Zhang et al., 2022c].

Apart from the various designs for the unlearning update, there are also different definitions of certified machine unlearning. Early works like Guo et al. [2020] introduce a certified data-removal

mechanism that adds random perturbations to the loss function at training time. Golatkar et al. [2020a] introduce an information-theoretic-based certified unlearning notion and also add random noise to ensure the certification, which is specific to the Fisher Information Matrix and not general enough. More recently, Sekhari et al. [2021] propose the $(\epsilon, \delta)$-certified machine unlearning definition that does not require introducing additional randomization during training. More essential, Sekhari et al. [2021] points out the importance of providing the generalization performance after machine unlearning. Sekhari et al. [2021], Suriyakumar and Wilson [2022] establish the generalization result in terms of the population risk and derive the deletion capacity guarantee.

However, most of existing works only consider machine unlearning for STL models that minimize a single variable. None of the prior works provide certified machine unlearning pertaining to minimax models, for which the generalization and deletion capacity guarantees are still unknown.

## 3 Preliminaries and Baseline Solution

### 3.1 Minimax Learning

The goal of minimax learning is to optimize the population loss $F(\boldsymbol{\omega}, \boldsymbol{\nu})$, given by

$$\min_{\boldsymbol{\omega} \in \mathcal{W}} \max_{\boldsymbol{\nu} \in \mathcal{V}} F(\boldsymbol{\omega}, \boldsymbol{\nu}) := \mathbb{E}_{z \sim \mathcal{D}}[f(\boldsymbol{\omega}, \boldsymbol{\nu}; z)], \tag{1}$$

where $f : \mathcal{W} \times \mathcal{V} \times \mathcal{Z} \to \mathbb{R}$ is the loss function, $z \in \mathcal{Z}$ is a data instance from the distribution $\mathcal{D}$, $\mathcal{W} \subseteq \mathbb{R}^{d_1}$ and $\mathcal{V} \subseteq \mathbb{R}^{d_2}$ are closed convex domains with regard to primal and dual variables, respectively. Since the data distribution $\mathcal{D}$ is unknown in practice, minimax learning turns to optimize the empirical loss $F_S(\boldsymbol{\omega}, \boldsymbol{\nu})$, given by,

$$\min_{\boldsymbol{\omega} \in \mathcal{W}} \max_{\boldsymbol{\nu} \in \mathcal{V}} F_S(\boldsymbol{\omega}, \boldsymbol{\nu}) := \frac{1}{n} \sum_{i=1}^{n} f(\boldsymbol{\omega}, \boldsymbol{\nu}; z_i), \tag{2}$$

where $S = \{z_1, \cdots, z_n\}$ is the training dataset with $z_i \sim \mathcal{D}$.

We will consider $L$-Lipschitz, $\ell$-smooth and $\mu_{\boldsymbol{\omega}}$-strongly-convex-$\mu_{\boldsymbol{\nu}}$-strongly-concave loss functions, which are described in Assumption 1&2 below and more details can be found in Appendix A.

**Assumption 1.** *For any $z \in \mathcal{Z}$, the function $f(\boldsymbol{\omega}, \boldsymbol{\nu}; z)$ is $L$-Lipschitz and with $\ell$-Lipschitz gradients and $\rho$-Lipschitz Hessians on the closed convex domain $\mathcal{W} \times \mathcal{V}$. Moreover, $f(\boldsymbol{\omega}, \boldsymbol{\nu}; z)$ is convex on $\mathcal{W}$ for any $\boldsymbol{\nu} \in \mathcal{V}$ and concave on $\mathcal{V}$ for any $\boldsymbol{\omega} \in \mathcal{W}$.*

**Assumption 2.** *For any $z \in \mathcal{Z}$, the function $f(\boldsymbol{\omega}, \boldsymbol{\nu}; z)$ satisfies Assumtion 1 and $f(\boldsymbol{\omega}, \boldsymbol{\nu}; z)$ is $\mu_{\boldsymbol{\omega}}$-strongly convex on $\mathcal{W}$ for any $\boldsymbol{\nu} \in \mathcal{V}$ and $\mu_{\boldsymbol{\nu}}$-strongly concave on $\mathcal{V}$ for any $\boldsymbol{\omega} \in \mathcal{W}$.*

Denote a randomized minimax learning algorithm by $A : \mathcal{Z}^n \to \mathcal{W} \times \mathcal{V}$ and its trained variables by $A(S) = (A_{\boldsymbol{\omega}}(S), A_{\boldsymbol{\nu}}(S)) \in \mathcal{W} \times \mathcal{V}$. The generalization performance is a top concern of the trained model variables $(A_{\boldsymbol{\omega}}(S), A_{\boldsymbol{\nu}}(S))$ [Thekumparampil et al., 2019, Zhang et al., 2020, Lei et al., 2021, Farnia and Ozdaglar, 2021, Zhang et al., 2021, 2022b, Ozdaglar et al., 2022], which can be measured by population weak primal-dual (PD) risk or population strong PD risk, as formalized below.

**Definition 1** (**Population Primal-Dual (PD) Risk**). *The population weak PD risk of $A(S)$, $\triangle^w(A_{\boldsymbol{\omega}}(S), A_{\boldsymbol{\nu}}(S))$ and the population strong PD risk of $A(S)$, $\triangle^s(A_{\boldsymbol{\omega}}(S), A_{\boldsymbol{\nu}}(S))$ are defined as*

$$\begin{cases} \triangle^w(A_{\boldsymbol{\omega}}(S), A_{\boldsymbol{\nu}}(S)) = \max_{\boldsymbol{\nu} \in \mathcal{V}} \mathbb{E}[F(A_{\boldsymbol{\omega}}(S), \boldsymbol{\nu})] - \min_{\boldsymbol{\omega} \in \mathcal{W}} \mathbb{E}[F(\boldsymbol{\omega}, A_{\boldsymbol{\nu}}(S))], \\ \triangle^s(A_{\boldsymbol{\omega}}(S), A_{\boldsymbol{\nu}}(S)) = \mathbb{E}[\max_{\boldsymbol{\nu} \in \mathcal{V}} F(A_{\boldsymbol{\omega}}(S), \boldsymbol{\nu}) - \min_{\boldsymbol{\omega} \in \mathcal{W}} F(\boldsymbol{\omega}, A_{\boldsymbol{\nu}}(S))]. \end{cases} \tag{3}$$

**Notations.** We introduce the following notations that will be used in the sequel. For a twice differentiable function $f$ with the arguments $\boldsymbol{\omega} \in \mathcal{W}$ and $\boldsymbol{\nu} \in \mathcal{V}$, we use $\nabla_{\boldsymbol{\omega}} f$ and $\nabla_{\boldsymbol{\nu}} f$ to denote the direct gradient of $f$ w.r.t. $\boldsymbol{\omega}$ and $\boldsymbol{\nu}$, respectively and denote its Jacobian matrix as $\nabla f = [\nabla_{\boldsymbol{\omega}} f; \nabla_{\boldsymbol{\nu}} f]$. We use $\partial_{\boldsymbol{\omega}\boldsymbol{\omega}} f, \partial_{\boldsymbol{\omega}\boldsymbol{\nu}} f, \partial_{\boldsymbol{\nu}\boldsymbol{\omega}} f, \partial_{\boldsymbol{\nu}\boldsymbol{\nu}} f$ to denote the second order partial derivatives w.r.t. $\boldsymbol{\omega}$ and $\boldsymbol{\nu}$, correspondingly and denote its Hessian matrix as $\nabla^2 f = [\partial_{\boldsymbol{\omega}\boldsymbol{\omega}} f, \partial_{\boldsymbol{\omega}\boldsymbol{\nu}} f; \partial_{\boldsymbol{\nu}\boldsymbol{\omega}} f, \partial_{\boldsymbol{\nu}\boldsymbol{\nu}} f]$. We define the total Hessian of the function $f$ w.r.t. $\boldsymbol{\omega}$ and $\boldsymbol{\nu}$: $\mathrm{D}_{\boldsymbol{\omega}\boldsymbol{\omega}} f := \partial_{\boldsymbol{\omega}\boldsymbol{\omega}} f - \partial_{\boldsymbol{\omega}\boldsymbol{\nu}} f \cdot \partial_{\boldsymbol{\nu}\boldsymbol{\nu}}^{-1} f \cdot \partial_{\boldsymbol{\nu}\boldsymbol{\omega}} f$ and $\mathrm{D}_{\boldsymbol{\nu}\boldsymbol{\nu}} f := \partial_{\boldsymbol{\nu}\boldsymbol{\nu}} f - \partial_{\boldsymbol{\nu}\boldsymbol{\omega}} f \cdot \partial_{\boldsymbol{\omega}\boldsymbol{\omega}}^{-1} f \cdot \partial_{\boldsymbol{\omega}\boldsymbol{\nu}} f$ where $\partial_{\boldsymbol{\nu}\boldsymbol{\nu}}^{-1} f$ and $\partial_{\boldsymbol{\omega}\boldsymbol{\omega}}^{-1} f$ are the shorthand of $(\partial_{\boldsymbol{\nu}\boldsymbol{\nu}} f(\cdot))^{-1}$ and $(\partial_{\boldsymbol{\omega}\boldsymbol{\omega}} f(\cdot))^{-1}$, respectively, when $\partial_{\boldsymbol{\nu}\boldsymbol{\nu}} f$ and $\partial_{\boldsymbol{\omega}\boldsymbol{\omega}} f$ are invertible. We also use the shorthand notation $\nabla_{\boldsymbol{\omega}} f(\boldsymbol{\omega}_1, \boldsymbol{\nu}; z) = \nabla_{\boldsymbol{\omega}} f(\boldsymbol{\omega}, \boldsymbol{\nu}; z)|_{\boldsymbol{\omega}=\boldsymbol{\omega}_1}$.

## 3.2 $(\epsilon, \delta)$-Certified Machine Unlearning

An unlearning algorithm $\bar{A}$ for minimax models receives the output of a minimax learning algorithm $A(S)$, the set of delete requests $U \subseteq S$ and some additional memory variables $T(S) \in \mathcal{T}$ as input and returns an updated model $(\boldsymbol{\omega}^u, \boldsymbol{\nu}^u) = (\bar{A}_{\boldsymbol{\omega}}(U, A(S), T(S)), \bar{A}_{\boldsymbol{\nu}}(U, A(S), T(S))) \in \mathcal{W} \times \mathcal{V}$, aiming to remove the influence of $U$. For the memory variables in $T(S)$, it will not contain the entire training set, but instead its size $|T(S)|$ is independent of the training data size $n$. The mapping of an unlearning algorithm can be formulated as $\bar{A} : \mathcal{Z}^m \times \mathcal{W} \times \mathcal{V} \times \mathcal{T} \to \mathcal{W} \times \mathcal{V}$. We now give the notion of $(\epsilon, \delta)$-certified unlearning introduced by Sekhari et al. [2021], which is inspired by the definition of differential privacy [Dwork et al., 2006].

**Definition 2 ($(\epsilon, \delta)$-Certified Unlearning [Sekhari et al., 2021]).** *Let $\Theta$ be the domain of $\mathcal{W} \times \mathcal{V}$. For all $S$ of size $n$, set of delete requests $U \subseteq S$ such that $|U| \leq m$, the pair of learning algorithm $A$ and unlearning algorithm $\bar{A}$ is $(\epsilon, \delta)$-certified unlearning, if $\forall O \subseteq \Theta$ and $\epsilon, \delta > 0$, the following two conditions are satisfied:*

$$\Pr[\bar{A}(U, A(S), T(S)) \in O] \leq e^{\epsilon} \cdot \Pr[\bar{A}(\emptyset, A(S \backslash U), T(S \backslash U)) \in O] + \delta, \quad (4)$$

$$\Pr[\bar{A}(\emptyset, A(S \backslash U), T(S \backslash U)) \in O] \leq e^{\epsilon} \cdot \Pr[\bar{A}(U, A(S), T(S)) \in O] + \delta, \quad (5)$$

*where $\emptyset$ denotes the empty set and $T(S)$ denotes the memory variables available to $\bar{A}$.*

The above definition ensures the indistinguishability between the output distribution of (i) the model trained on the set $S$ and then unlearned with delete requests $U$ and (ii) the model trained on the set $S \backslash U$ and then unlearned with an empty set. Specifically, the unlearning algorithm simply adds perturbations to the output of $A(S \backslash U)$ when the set of delete requests is empty.

**Deletion Capacity.** Under the definition of certified unlearning, Sekhari et al. [2021] introduce the definition of deletion capacity, which formalizes how many samples can be deleted while still maintaining good guarantees on test loss. Here, we utilize the population primal-dual risk defined in Definition 1 instead of the excess population risk utilized for STL models.

**Definition 3 (Deletion capacity, [Sekhari et al., 2021]).** *Let $\epsilon, \delta, \gamma > 0$ and $S$ be a dataset of size $n$ drawn i.i.d from the data distribution $\mathcal{D}$. Let $F(\boldsymbol{\omega}, \boldsymbol{\nu})$ be a minimax model and $U$ be the set of deletion requests. For a pair of minimax learning algorithm $A$ and minimax unlearning algorithm $\bar{A}$ that satisfies $(\epsilon, \delta)$-unlearning, the deletion capacity $m_{\epsilon, \delta, \gamma}^{A, \bar{A}}(d_1, d_2, n)$ is defined as the maximum number of samples $U$ that can be unlearned while still ensuring the population primal-dual (weak PD or strong PD) risk is at most $\gamma$. Let the expectation $\mathbb{E}[\cdot]$ takes over $S \sim \mathcal{D}^n$ and the outputs of the algorithms $A$ and $\bar{A}$. Let $d_1$ denotes the dimension of domain $\mathcal{W}$ and $d_2$ denotes the dimension of domain $\mathcal{V}$, specifically,*

$$m_{\epsilon, \delta, \gamma}^{A, \bar{A}}(d_1, d_2, n) := \max \left\{ m | \triangle \left( \bar{A}_{\boldsymbol{\omega}}(U, A(S), T(S)), \bar{A}_{\boldsymbol{\nu}}(U, A(S), T(S)) \right) \leq \gamma \right\}, \quad (6)$$

*where the ouputs $\bar{A}_{\boldsymbol{\omega}}(U, A(S), T(S))$ and $\bar{A}_{\boldsymbol{\nu}}(U, A(S), T(S))$ of the minimax unlearning algorithm $\bar{A}$ refer to parameter $\boldsymbol{\omega}$ and $\boldsymbol{\nu}$, respectively. $\triangle \left( \bar{A}_{\boldsymbol{\omega}}(U, A(S), T(S)), \bar{A}_{\boldsymbol{\nu}}(U, A(S), T(S)) \right)$ could be the population weak PD risk or population strong PD risk of $\bar{A}(U, A(S), T(S))$.*

We set $\gamma = 0.01$ (or any other small arbitrary constant) throughout the paper.

## 3.3 Baseline Solution: Certified Minimax Unlearning via Differential Privacy

Since Definition 2 is motivated by differential privacy (DP), it is a natural way to use tools from DP for machine unlearning. For a differentially private learning algorithm $A$ with edit distance $m$ in neighboring datasets, the unlearning algorithm $\bar{A}$ simply returns its output $A(S)$ without any changes and is independent of the delete requests $U$ as well as the memory variables $T(S)$, i.e., $\bar{A}(U, A(S), T(S)) = A(S)$.

A number of differentially private minimax learning algorithms can be applied, e.g., Zhang et al. [2022a], Yang et al. [2022], Bassily et al. [2023]. For instance, we can obtain the output $A(S) = (A_{\boldsymbol{\omega}}(S), A_{\boldsymbol{\nu}}(S))$ by calling Algorithm 3 in Zhang et al. [2022a]. Under Assumption 1&2, we then get the population strong PD risk based on [Zhang et al., 2022a, Theorem 4.3] and the group privacy property of DP [Vadhan, 2017, Lemma 7.2.2], as follows,

$$\triangle^s(A_{\boldsymbol{\omega}}(S), A_{\boldsymbol{\nu}}(S)) = \mathcal{O} \left( \frac{\kappa^2}{\mu n} + \frac{m^2 \kappa^2 d \log(m e^{\epsilon} / \delta)}{\mu n^2 \epsilon^2} \right), \quad (7)$$

where we let $\mu = \min\{\mu_{\boldsymbol{\omega}}, \mu_{\boldsymbol{\nu}}\}$, $\kappa = \ell/\mu$, $d = \max\{d_1, d_2\}$, and $m$ be the edit distance between datasets (i.e., the original dataset and the remaining dataset after removing samples to be forgotten).

The algorithm $A$ satisfies $(\epsilon, \delta)$-DP for any set $U \subseteq S$ of size $m$, that is,

$$\Pr[A(S) \in O] \leq e^\epsilon \Pr[A(S \backslash U) \in O] + \delta \quad \text{and} \quad \Pr[A(S \backslash U) \in O] \leq e^\epsilon \Pr[A(S) \in O] + \delta.$$

Since we have $A(S) = \bar{A}(U, A(S), T(S))$ and $A(S \backslash U) = \bar{A}(\emptyset, A(S \backslash U), T(S \backslash U))$, the above privacy guarantee can be converted to the minimax unlearning guarantee in Definition 2, implying that the pair $(A, \bar{A})$ is $(\epsilon, \delta)$-certified minimax unlearning. According to Definition 3, the population strong PD risk in eq.(7) yields the following bound on deletion capacity.

**Theorem 1** (**Deletion capacity of unlearning via DP [Sekhari et al., 2021]**). *Denote* $d = \max\{d_1, d_2\}$. *There exists a polynomial time learning algorithm $A$ and unlearning algorithm $\bar{A}$ for minimax problem of the form $\bar{A}(U, A(S), T(S)) = A(S)$ such that the deletion capacity is:*

$$m_{\epsilon,\delta,\gamma}^{A,\bar{A}}(d_1, d_2, n) \geq \widetilde{\Omega}\left(\frac{n\epsilon}{\sqrt{d \log(e^\epsilon/\delta)}}\right), \tag{8}$$

*where the constant in $\widetilde{\Omega}$-notation depends on the properties of the loss function $f$ (e.g., strongly convexity and strongly concavity parameters, Lipchitz continuity and smoothness parameters).*

However, this DP minimax learning baseline approach provides an inferior deletion capacity. In the following sections, we show that the $d^{1/2}$ in the denominator of eq.(8) can be further reduced to $d^{1/4}$.

# 4 Certified Minimax Unlearning

In this section, we focus on the setting of the strongly-convex-strong-concave loss function. We first provide the intuition for the design of the minimax unlearning step in Sec.4.1, then provide the formal algorithm in Sec.4.2 and a more efficient extension in Sec.4.3 with analysis of minimax unlearning certification, generalization result, and deletion capacity in Sec.4.4. We will provide extensions to more general loss settings in Sec.5. The proofs for the theorems presented in this and the next sections can be found in Appendix B and C, respectively.

## 4.1 Intuition for Minimax Unlearning Update

To begin with, we provide an informal derivation for minimax unlearning update to illustrate its design intuition. Given the training set $S$ of size $n$ and the deletion subset $U \subseteq S$ of size $m$, the aim is to approximate the optimal solution $(\boldsymbol{\omega}_{S\backslash}^*, \boldsymbol{\nu}_{S\backslash}^*)$ of the loss $F_{S\backslash}(\boldsymbol{\omega}, \boldsymbol{\nu})$ on the remaining dataset $S \setminus U$, given by,

$$(\boldsymbol{\omega}_{S\backslash}^*, \boldsymbol{\nu}_{S\backslash}^*) := \arg\min_{\boldsymbol{\omega} \in \mathcal{W}} \max_{\boldsymbol{\nu} \in \mathcal{V}} \{F_{S\backslash}(\boldsymbol{\omega}, \boldsymbol{\nu}) := \frac{1}{n-m} \sum_{z_i \in S \backslash U} f(\boldsymbol{\omega}, \boldsymbol{\nu}; z_i)\}. \tag{9}$$

Meanwhile, we have the optimal solution $(\boldsymbol{\omega}_S^*, \boldsymbol{\nu}_S^*)$ to the original loss $F_S(\boldsymbol{\omega}, \boldsymbol{\nu})$ after minimax learning. Taking unlearning $\boldsymbol{\omega}$ for instance, by using a first-order Taylor expansion for $\nabla_{\boldsymbol{\omega}} F_{S\backslash}(\boldsymbol{\omega}_{S\backslash}^*, \boldsymbol{\nu}_{S\backslash}^*) = 0$ around $(\boldsymbol{\omega}_S^*, \boldsymbol{\nu}_S^*)$, we have

$$\nabla_{\boldsymbol{\omega}} F_{S\backslash}(\boldsymbol{\omega}_S^*, \boldsymbol{\nu}_S^*) + \partial_{\boldsymbol{\omega}\boldsymbol{\omega}} F_{S\backslash}(\boldsymbol{\omega}_S^*, \boldsymbol{\nu}_S^*)(\boldsymbol{\omega}_{S\backslash}^* - \boldsymbol{\omega}_S^*) + \partial_{\boldsymbol{\omega}\boldsymbol{\nu}} F_{S\backslash}(\boldsymbol{\omega}_S^*, \boldsymbol{\nu}_S^*)(\boldsymbol{\nu}_{S\backslash}^* - \boldsymbol{\nu}_S^*) \approx 0. \tag{10}$$

Since $\boldsymbol{\omega}_S^*$ is a minimizer of $F_S(\boldsymbol{\omega}, \boldsymbol{\nu})$, from the first-order optimality condition, we can get $\nabla_{\boldsymbol{\omega}} F_{S\backslash}(\boldsymbol{\omega}_S^*, \boldsymbol{\nu}_S^*) = -\frac{1}{n-m} \sum_{z_i \in U} \nabla_{\boldsymbol{\omega}} f(\boldsymbol{\omega}_S^*, \boldsymbol{\nu}_S^*; z_i)$. Now given an auxiliary function $\mathtt{V}_{S\backslash}(\boldsymbol{\omega}) = \arg\max_{\boldsymbol{\nu} \in \mathcal{V}} F_{S\backslash}(\boldsymbol{\omega}, \boldsymbol{\nu})$ (more best response auxiliary functions are introduced in Appendix A, Definition 8), we have $\boldsymbol{\nu}_{S\backslash}^* = \mathtt{V}_{S\backslash}(\boldsymbol{\omega}_{S\backslash}^*)$. We further get

$$
\begin{aligned}
\boldsymbol{\nu}_{S\backslash}^* - \boldsymbol{\nu}_S^* &= [\mathtt{V}_{S\backslash}(\boldsymbol{\omega}_{S\backslash}^*) - \mathtt{V}_{S\backslash}(\boldsymbol{\omega}_S^*)] + [\mathtt{V}_{S\backslash}(\boldsymbol{\omega}_S^*) - \boldsymbol{\nu}_S^*] \\
&\stackrel{(i)}{\approx} \mathtt{V}_{S\backslash}(\boldsymbol{\omega}_{S\backslash}^*) - \mathtt{V}_{S\backslash}(\boldsymbol{\omega}_S^*) \stackrel{(ii)}{\approx} \left(\frac{d\mathtt{V}_{S\backslash}(\boldsymbol{\omega})}{d\boldsymbol{\omega}}\Big|_{\boldsymbol{\omega}=\boldsymbol{\omega}_S^*}\right)(\boldsymbol{\omega}_{S\backslash}^* - \boldsymbol{\omega}_S^*) \\
&\stackrel{(iii)}{\approx} -\partial_{\boldsymbol{\nu}\boldsymbol{\nu}}^{-1} F_{S\backslash}(\boldsymbol{\omega}_S^*, \boldsymbol{\nu}_S^*) \cdot \partial_{\boldsymbol{\nu}\boldsymbol{\omega}} F_{S\backslash}(\boldsymbol{\omega}_S^*, \boldsymbol{\nu}_S^*) \cdot (\boldsymbol{\omega}_{S\backslash}^* - \boldsymbol{\omega}_S^*),
\end{aligned}
\tag{11}
$$

where the approximate equation $(i)$ leaving out the term $[\mathtt{V}_{S\backslash}(\boldsymbol{\omega}_S^*) - \boldsymbol{\nu}_S^*]$ which is bounded in Appendix A, Lemma 2, and does not affect the overall unlearning guarantee. The approximate equation $(ii)$ is the linear approximation step and is the response Jacobian of the auxiliary function $\mathtt{V}_{S\backslash}(\boldsymbol{\omega})$. The approximate equation $(iii)$ is due to the implicit function theorem. This gives that

$$\partial_{\boldsymbol{\omega\omega}}F_{S\backslash}(\boldsymbol{\omega}_S^*, \boldsymbol{\nu}_S^*)(\boldsymbol{\omega}_{S\backslash}^* - \boldsymbol{\omega}_S^*) + \partial_{\boldsymbol{\omega\nu}}F_{S\backslash}(\boldsymbol{\omega}_S^*, \boldsymbol{\nu}_S^*)(\boldsymbol{\nu}_{S\backslash}^* - \boldsymbol{\nu}_S^*) = \mathtt{D}_{\boldsymbol{\omega\omega}}F_{S\backslash}(\boldsymbol{\omega}_S^*, \boldsymbol{\nu}_S^*)(\boldsymbol{\omega}_{S\backslash}^* - \boldsymbol{\omega}_S^*), \quad (12)$$

which implies the following approximation of $\boldsymbol{\omega}_{S\backslash}^*$:

$$\boldsymbol{\omega}_{S\backslash}^* \approx \boldsymbol{\omega}_S^* + \frac{1}{n-m}[\mathtt{D}_{\boldsymbol{\omega\omega}}F_{S\backslash}(\boldsymbol{\omega}_S^*, \boldsymbol{\nu}_S^*)]^{-1} \sum_{z_i \in U} \nabla_{\boldsymbol{\omega}} f(\boldsymbol{\omega}_S^*, \boldsymbol{\nu}_S^*; z_i). \quad (13)$$

The above informal derivation indicates that the minimax unlearning update relies on the total Hessian to sufficiently remove the data influence [Liu et al., 2023, Zhang et al., 2023], rather than the conventional Hessian that appears in standard statistical unlearning [Guo et al., 2020, Sekhari et al., 2021, Suriyakumar and Wilson, 2022, Mehta et al., 2022]. The update in eq.(13) has a close relation to the complete Newton step in the second-order minimax optimization literature [Zhang et al., 2020], which motivates the complete Newton-based minimax unlearning. However, due to the various approximations in the above informal derivation, we cannot have a certified minimax unlearning guarantee. Below, we will formally derive the upper bound for these approximations in the closeness upper bound analysis. Based on the closeness upper bound, we will introduce the Gaussian mechanism to yield distribution indistinguishably result in the sense of $(\epsilon, \delta)$-certified minimax unlearning.

### 4.2 Proposed Certified Minimax Unlearning

We first provide algorithms under the setting of the smooth and strongly-convex-strongly-concave (SC-SC) loss function as described in Assumptions 1&2.

---

**Algorithm 1** Minimax Learning Algorithm ($A_{sc-sc}$)

**Input:** Dataset $S : \{z_i\}_{i=1}^n \sim \mathcal{D}^n$, loss function: $f(\boldsymbol{\omega}, \boldsymbol{\nu}; z)$.
1: Compute

$$(\boldsymbol{\omega}_S^*, \boldsymbol{\nu}_S^*) \leftarrow \arg\min_{\boldsymbol{\omega}} \max_{\boldsymbol{\nu}} F_S(\boldsymbol{\omega}, \boldsymbol{\nu}) = \frac{1}{n} \sum_{i=1}^n f(\boldsymbol{\omega}, \boldsymbol{\nu}; z_i). \quad (14)$$

**Output:** $(\boldsymbol{\omega}_S^*, \boldsymbol{\nu}_S^*, \mathtt{D}_{\boldsymbol{\omega\omega}}F_S(\boldsymbol{\omega}_S^*, \boldsymbol{\nu}_S^*), \mathtt{D}_{\boldsymbol{\nu\nu}}F_S(\boldsymbol{\omega}_S^*, \boldsymbol{\nu}_S^*))$

---

**Algorithm 2** Certified Minimax Unlearning for Strongly-Convex-Strongly-Concave Loss ($\bar{A}_{sc-sc}$)

**Input:** Delete requests $U : \{z_j\}_{j=1}^m \subseteq S$, output of $A_{sc-sc}(S)$: $(\boldsymbol{\omega}_S^*, \boldsymbol{\nu}_S^*)$, memory variables $T(S)$: $\{\mathtt{D}_{\boldsymbol{\omega\omega}}F_S(\boldsymbol{\omega}_S^*, \boldsymbol{\nu}_S^*), \mathtt{D}_{\boldsymbol{\nu\nu}}F_S(\boldsymbol{\omega}_S^*, \boldsymbol{\nu}_S^*)\}$, loss function: $f(\boldsymbol{\omega}, \boldsymbol{\nu}; z)$, noise parameters: $\sigma_1, \sigma_2$.
1: Compute

$$\mathtt{D}_{\boldsymbol{\omega\omega}}F_{S\backslash}(\boldsymbol{\omega}_S^*, \boldsymbol{\nu}_S^*) = \frac{1}{n-m}\left(n\mathtt{D}_{\boldsymbol{\omega\omega}}F_S(\boldsymbol{\omega}_S^*, \boldsymbol{\nu}_S^*) - \sum_{z_i \in U} \mathtt{D}_{\boldsymbol{\omega\omega}}f(\boldsymbol{\omega}_S^*, \boldsymbol{\nu}_S^*; z_i)\right), \quad (15)$$

$$\mathtt{D}_{\boldsymbol{\nu\nu}}F_{S\backslash}(\boldsymbol{\omega}_S^*, \boldsymbol{\nu}_S^*) = \frac{1}{n-m}\left(n\mathtt{D}_{\boldsymbol{\nu\nu}}F_S(\boldsymbol{\omega}_S^*, \boldsymbol{\nu}_S^*) - \sum_{z_i \in U} \mathtt{D}_{\boldsymbol{\nu\nu}}f(\boldsymbol{\omega}_S^*, \boldsymbol{\nu}_S^*; z_i)\right). \quad (16)$$

2: Define

$$\widehat{\boldsymbol{\omega}} = \boldsymbol{\omega}_S^* + \frac{1}{n-m}[\mathtt{D}_{\boldsymbol{\omega\omega}}F_{S\backslash}(\boldsymbol{\omega}_S^*, \boldsymbol{\nu}_S^*)]^{-1} \sum_{z_i \in U} \nabla_{\boldsymbol{\omega}} f(\boldsymbol{\omega}_S^*, \boldsymbol{\nu}_S^*; z_i), \quad (17)$$

$$\widehat{\boldsymbol{\nu}} = \boldsymbol{\nu}_S^* + \frac{1}{n-m}[\mathtt{D}_{\boldsymbol{\nu\nu}}F_{S\backslash}(\boldsymbol{\omega}_S^*, \boldsymbol{\nu}_S^*)]^{-1} \sum_{z_i \in U} \nabla_{\boldsymbol{\nu}} f(\boldsymbol{\omega}_S^*, \boldsymbol{\nu}_S^*; z_i). \quad (18)$$

3: $\boldsymbol{\omega}^u = \widehat{\boldsymbol{\omega}} + \boldsymbol{\xi}_1$, where $\boldsymbol{\xi}_1 \sim \mathcal{N}(0, \sigma_1 \mathbf{I}_{d_1})$ and $\boldsymbol{\nu}^u = \widehat{\boldsymbol{\nu}} + \boldsymbol{\xi}_2$, where $\boldsymbol{\xi}_2 \sim \mathcal{N}(0, \sigma_2 \mathbf{I}_{d_2})$.
**Output:** $(\boldsymbol{\omega}^u, \boldsymbol{\nu}^u)$.

---

**Minimax Learning algorithm.** We denote our learning algorithm by $A_{sc-sc}$ and the pseudocode is shown in Algorithm 1. Given a dataset $S = \{z_i\}_{i=1}^n$ of size $n$ drawn independently from some distribution $\mathcal{D}$, algorithm $A_{sc-sc}$ computes the optimal solution $(\boldsymbol{\omega}_S^*, \boldsymbol{\nu}_S^*)$ to the empirical risk $F_S(\boldsymbol{\omega}, \boldsymbol{\nu})$. $A_{sc-sc}$ then outputs the point $(\boldsymbol{\omega}_S^*, \boldsymbol{\nu}_S^*)$ as well as the additional memory variables $T(S) := \{\mathtt{D}_{\boldsymbol{\omega\omega}} F_S(\boldsymbol{\omega}_S^*, \boldsymbol{\nu}_S^*), \mathtt{D}_{\boldsymbol{\nu\nu}} F_S(\boldsymbol{\omega}_S^*, \boldsymbol{\nu}_S^*)\}$, which computes and stores the total Hessian of $F_S(\boldsymbol{\omega}, \boldsymbol{\nu})$ at $(\boldsymbol{\omega}_S^*, \boldsymbol{\nu}_S^*)$.

**Minimax Unlearning Algorithm** We denote the proposed certified minimax unlearning algorithm by $\bar{A}_{sc-sc}$ and present its pseudocode in Algorithm 2. Algorithm $\bar{A}_{sc-sc}$ takes the following inputs: the set of delete requests $U = \{z_j\}_{j=1}^m$ of size $m$, the trained minimax model $(\boldsymbol{\omega}_S^*, \boldsymbol{\nu}_S^*)$, and the memory variables $T(S)$. To have the certified minimax unlearning for $\boldsymbol{\omega}$, eq.(15) computes the total Hessian of $F_{S\backslash}(\boldsymbol{\omega}_S^*, \boldsymbol{\nu}_S^*)$ by $\frac{n}{n-m}\mathtt{D}_{\boldsymbol{\omega\omega}} F_S(\boldsymbol{\omega}_S^*, \boldsymbol{\nu}_S^*) - \frac{1}{n-m}\sum_{z_i \in U} \mathtt{D}_{\boldsymbol{\omega\omega}} f(\boldsymbol{\omega}_S^*, \boldsymbol{\nu}_S^*, z_i)$, where the former term can be retrieved from the memory set and the latter is computed on the samples to be deleted; eq.(17) computes the intermediate $\widehat{\boldsymbol{\omega}}$ by the complete Newton step based on the total Hessian $\mathtt{D}_{\boldsymbol{\omega\omega}} F_{S\backslash}(\boldsymbol{\omega}_S^*, \boldsymbol{\nu}_S^*)$; Line 3 injects calibrated Gaussian noise $\boldsymbol{\xi}_1$ to ensure $(\epsilon, \delta)$-certified minimax unlearning. The certified minimax unlearning for $\boldsymbol{\nu}$ is symmetric. We provide detailed analysis for Algorithm 2 including minimax unlearning certification, generalization results, and deletion capacity in Appendix B.1.

## 4.3 Certified Minimax Unlearning without Total Hessian Re-computation

We extend Algorithm 2 and propose Algorithm 3 to reduce the computational cost of Algorithm 2. The complete Newton steps in eq.(19) and eq.(20) utilize the total Hessian $\mathtt{D}_{\boldsymbol{\omega\omega}} F_S(\boldsymbol{\omega}_S^*, \boldsymbol{\nu}_S^*)$ and $\mathtt{D}_{\boldsymbol{\nu\nu}} F_S(\boldsymbol{\omega}_S^*, \boldsymbol{\nu}_S^*)$ that are directly retrieved from the memory, rather than the updated total Hessian $\mathtt{D}_{\boldsymbol{\omega\omega}} F_{S\backslash}(\boldsymbol{\omega}_S^*, \boldsymbol{\nu}_S^*)$ and $\mathtt{D}_{\boldsymbol{\nu\nu}} F_{S\backslash}(\boldsymbol{\omega}_S^*, \boldsymbol{\nu}_S^*)$ used in Algorithm 2. The form in eq.(19) and eq.(20) can also be regarded as the total Hessian extension of the infinitesimal jackknife. In this way, it gets rid of the computationally demanding part of re-evaluating the total Hessian for samples to be deleted, which significantly reduces the computational cost. It turns out to be the same computational complexity as the state-of-the-art certified unlearning method developed for STL models [Suriyakumar and Wilson, 2022]. Moreover, Algorithm 3 can be more appealing for the successive data deletion setting.

---

**Algorithm 3** Efficient Certified Minimax Unlearning ($\bar{A}_{\texttt{efficient}}$)

---

**Input:** Delete requests $U : \{z_j\}_{j=1}^m \subseteq S$, output of $A_{sc-sc}(S)$: $(\boldsymbol{\omega}_S^*, \boldsymbol{\nu}_S^*)$, memory variables $T(S)$: $\{\mathtt{D}_{\boldsymbol{\omega\omega}} F_S(\boldsymbol{\omega}_S^*, \boldsymbol{\nu}_S^*), \mathtt{D}_{\boldsymbol{\nu\nu}} F_S(\boldsymbol{\omega}_S^*, \boldsymbol{\nu}_S^*)\}$, loss function: $f(\boldsymbol{\omega}, \boldsymbol{\nu}; z)$, noise parameters: $\sigma_1, \sigma_2$.
  1: Compute

$$\widetilde{\boldsymbol{\omega}} = \boldsymbol{\omega}_S^* + \frac{1}{n}[\mathtt{D}_{\boldsymbol{\omega\omega}} F_S(\boldsymbol{\omega}_S^*, \boldsymbol{\nu}_S^*)]^{-1} \sum_{z_i \in U} \nabla_{\boldsymbol{\omega}} f(\boldsymbol{\omega}_S^*, \boldsymbol{\nu}_S^*; z_i), \quad (19)$$

$$\widetilde{\boldsymbol{\nu}} = \boldsymbol{\nu}_S^* + \frac{1}{n}[\mathtt{D}_{\boldsymbol{\nu\nu}} F_S(\boldsymbol{\omega}_S^*, \boldsymbol{\nu}_S^*)]^{-1} \sum_{z_i \in U} \nabla_{\boldsymbol{\nu}} f(\boldsymbol{\omega}_S^*, \boldsymbol{\nu}_S^*; z_i). \quad (20)$$

  2: $\widetilde{\boldsymbol{\omega}}^u = \widetilde{\boldsymbol{\omega}} + \boldsymbol{\xi}_1$, where $\boldsymbol{\xi}_1 \sim \mathcal{N}(0, \sigma_1 \mathbf{I}_{d_1})$ and $\widetilde{\boldsymbol{\nu}}^u = \widetilde{\boldsymbol{\nu}} + \boldsymbol{\xi}_2$, where $\boldsymbol{\xi}_2 \sim \mathcal{N}(0, \sigma_2 \mathbf{I}_{d_2})$ .
**Output:** $(\widetilde{\boldsymbol{\omega}}^u, \widetilde{\boldsymbol{\nu}}^u)$.

---

## 4.4 Analysis for Algorithm 3

$(\epsilon, \delta)$**-Certificated Unlearning Guarantee.** The intermediate variables $(\widetilde{\boldsymbol{\omega}}, \widetilde{\boldsymbol{\nu}})$ are distinguishable in distribution from the retraining-from-scratch variables $(\boldsymbol{\omega}_{S\backslash}^*, \boldsymbol{\nu}_{S\backslash}^*)$ because they are deterministic and the Taylor expansion introduces a certain amount of approximation. The following lemma quantifies the closeness between $(\widetilde{\boldsymbol{\omega}}, \widetilde{\boldsymbol{\nu}})$ and $(\boldsymbol{\omega}_{S\backslash}^*, \boldsymbol{\nu}_{S\backslash}^*)$, which can be regarded as the "sensitivity" when applying the Gaussian mechanism.

**Lemma 1** (**Closeness Upper Bound**). *Suppose the loss function $f$ satisfies Assumption 1 and 2, $\|\mathtt{D}_{\boldsymbol{\omega\omega}} F_S(\boldsymbol{\omega}_S^*, \boldsymbol{\nu}_S^*)\| \geq \mu_{\boldsymbol{\omega\omega}}$ and $\|\mathtt{D}_{\boldsymbol{\nu\nu}} F_S(\boldsymbol{\omega}_S^*, \boldsymbol{\nu}_S^*)\| \geq \mu_{\boldsymbol{\nu\nu}}$. Let $\mu = \min\{\mu_{\boldsymbol{\omega}}, \mu_{\boldsymbol{\nu}}, \mu_{\boldsymbol{\omega\omega}}, \mu_{\boldsymbol{\nu\nu}}\}$. Then, we have the closeness bound between $(\widetilde{\boldsymbol{\omega}}, \widetilde{\boldsymbol{\nu}})$ in Line 1 of Algorithm 3 and $(\boldsymbol{\omega}_{S\backslash}^*, \boldsymbol{\nu}_{S\backslash}^*)$ in eq.(9):*

$$\{\|\boldsymbol{\omega}_{S\backslash}^* - \widetilde{\boldsymbol{\omega}}\|, \|\boldsymbol{\nu}_{S\backslash}^* - \widetilde{\boldsymbol{\nu}}\|\} \leq \frac{(8\sqrt{2}L^2\ell^3\rho/\mu^6 + 2\sqrt{2}L\ell^2/\mu^3)m^2}{n^2}. \quad (21)$$

Equipped with Lemma 1, we have the following certified unlearning guarantee by adding Gaussian noise calibrated according to the above closeness result. Due to the minimax structure, our analysis is more involved than the STL case [Sekhari et al., 2021, Suriyakumar and Wilson, 2022].

**Theorem 2** (($\epsilon, \delta$)**-Minimax Unlearning Certification**). *Under the same settings of Lemma 1, our minimax learning algorithm $A_{sc-sc}$ and unlearning algorithm $\bar{A}_{\mathrm{efficient}}$ is ($\epsilon, \delta$)-certified minimax unlearning if we choose*

$$\sigma_1 \text{ and } \sigma_2 = \frac{2(8\sqrt{2}L^2\ell^3\rho/\mu^6 + 2\sqrt{2}L\ell^2/\mu^3)m^2}{n^2\epsilon}\sqrt{2\log(2.5/\delta)}. \tag{22}$$

**Generalization Guarantee.** Theorem 3 below provides the generalization result in terms of the population PD risk for the minimax unlearning algorithm $\bar{A}_{\mathrm{efficient}}$.

**Theorem 3** (**Population Primal-Dual Risk**). *Under the same settings of Lemma 1 and denote $d = \max\{d_1, d_2\}$, the population weak and strong PD risk for the certified minimax unlearning variables $(\widetilde{\boldsymbol{\omega}}^u, \widetilde{\boldsymbol{\nu}}^u)$ returned by Algorithm 3 are*

$$\begin{cases} \triangle^w(\widetilde{\boldsymbol{\omega}}^u, \widetilde{\boldsymbol{\nu}}^u) = \mathcal{O}\left((L^3\ell^3\rho/\mu^6 + L^2\ell^2/\mu^3) \cdot \frac{m^2\sqrt{d\log(1/\delta)}}{n^2\epsilon} + \frac{mL^2}{\mu n}\right), \\ \triangle^s(\widetilde{\boldsymbol{\omega}}^u, \widetilde{\boldsymbol{\nu}}^u) = \mathcal{O}\left((L^3\ell^3\rho/\mu^6 + L^2\ell^2/\mu^3) \cdot \frac{m^2\sqrt{d\log(1/\delta)}}{n^2\epsilon} + \frac{mL^2}{\mu n} + \frac{L^2\ell}{\mu^2 n}\right). \end{cases} \tag{23}$$

**Deletion Capacity.** The population weak and strong PD risk given in Theorem 3 for the output of unlearning algorithms provides the following bound on deletion capacity.

**Theorem 4** (**Deletion Capacity**). *Under the same settings of Lemma 1 and denote $d = \max\{d_1, d_2\}$, the deletion capacity of Algorithm 3 is*

$$m_{\epsilon,\delta,\gamma}^{A,\bar{A}}(d_1, d_2, n) \geq c \cdot \frac{n\sqrt{\epsilon}}{(d\log(1/\delta))^{1/4}}, \tag{24}$$

*where the constant $c$ depends on $L, l, \rho$, and $\mu$ of the loss function $f$.*

## 5 Certified Minimax Unlearning for Convex-Concave Loss Function

We further extend the certified minimax unlearning for the convex-concave loss function. In addition, Appendix C will provide the extension to convex-strongly-concave and strongly-convex-concave loss functions. Give the convex-concave loss function $f(\boldsymbol{\omega}, \boldsymbol{\nu}; z)$, similar to the unlearning for STL models [Sekhari et al., 2021], we define the regularized function as $\widetilde{f}(\boldsymbol{\omega}, \boldsymbol{\nu}; z) = f(\boldsymbol{\omega}, \boldsymbol{\nu}; z) + \frac{\lambda}{2}\|\boldsymbol{\omega}\|^2 - \frac{\lambda}{2}\|\boldsymbol{\nu}\|^2$. Suppose the function $f$ satisfies Assumption 1, then the function $\widetilde{f}$ is $\lambda$-strongly convex in $\boldsymbol{\omega}$, $\lambda$-strongly concave in $\boldsymbol{\nu}$, $(2L+\lambda\|\boldsymbol{\omega}\|+\lambda\|\boldsymbol{\nu}\|)$-Lipschitz, $\sqrt{2}(2\ell+\lambda)$-gradient Lipschitz and $\rho$-Hessian Lipschitz. It suffices to apply the minimax learning and unlearning algorithms in Sec.4 to the regularized loss function with a properly chosen $\lambda$. We denote the learning and unlearning algorithms for convex-concave losses as $A_{c-c}$ and $\bar{A}_{c-c}$. Their implementation details are given in Appendix C. We suppose the SC-SC regularization parameter $\lambda$ satisfies $\lambda < \ell$. Theorem 5 below summarizes guarantees of ($\epsilon, \delta$)-certified unlearning and population primal-dual risk (weak and strong) for Algorithm $\bar{A}_{c-c}$.

**Theorem 5.** *Let Assumption 1 hold and $d = \max\{d_1, d_2\}$. Suppose the parameter spaces $\mathcal{W}$ and $\mathcal{V}$ are bounded so that $\max_{\boldsymbol{\omega}\in\mathcal{W}}\|\boldsymbol{\omega}\| \leq B_{\boldsymbol{\omega}}$ and $\max_{\boldsymbol{\nu}\in\mathcal{V}}\|\boldsymbol{\nu}\| \leq B_{\boldsymbol{\nu}}$. We have,*

(a) ($\epsilon, \delta$)***-Minimax Unlearning Certification:*** *Our minimax learning algorithm $A_{c-c}$ and unlearning algorithm $\bar{A}_{c-c}$ is ($\epsilon, \delta$)-certified minimax unlearning.*

(b) ***Population Weak PD Risk:*** *The population weak PD risk for $(\boldsymbol{\omega}^u, \boldsymbol{\nu}^u)$ by algorithm $\bar{A}_{c-c}$ is*

$$\triangle^w(\boldsymbol{\omega}^u, \boldsymbol{\nu}^u) \leq \mathcal{O}\left((L^3\ell^3\rho/\lambda^6 + L^2\ell^2/\lambda^3) \cdot \frac{m^2\sqrt{d\log(1/\delta)}}{n^2\epsilon} + \frac{mL^2}{\lambda n} + \lambda(B_{\boldsymbol{\omega}}^2 + B_{\boldsymbol{\nu}}^2)\right). \tag{25}$$

*In particular, by setting $\lambda$ below*

$$\lambda = \max\left\{\frac{L}{\sqrt{B_{\boldsymbol{\omega}}^2 + B_{\boldsymbol{\nu}}^2}}\sqrt{\frac{m}{n}}, \left(\frac{L^2\ell^2 m^2\sqrt{d\log(1/\delta)}}{(B_{\boldsymbol{\omega}}^2 + B_{\boldsymbol{\nu}}^2)n^2\epsilon}\right)^{1/4}, \left(\frac{L^3\ell^3\rho m^2\sqrt{d\log(1/\delta)}}{(B_{\boldsymbol{\omega}}^2 + B_{\boldsymbol{\nu}}^2)n^2\epsilon}\right)^{1/7}\right\}, \tag{26}$$

*we have the following population weak PD risk,*

$$\triangle^w(\boldsymbol{\omega}^u, \boldsymbol{\nu}^u) \leq \mathcal{O}\left(c_1\sqrt{\frac{m}{n}} + c_2\left(\frac{d\log(1/\delta)}{\epsilon^2}\right)^{1/8}\sqrt{\frac{m}{n}} + c_3\left(\frac{\sqrt{d\log(1/\delta)}}{\epsilon}\right)^{1/7}\left(\frac{m}{n}\right)^{2/7}\right), \quad (27)$$

*where $c_1, c_2, c_3$ are constants that depend only on $L, l, \rho, B_{\boldsymbol{\omega}}$ and $B_{\boldsymbol{\nu}}$.*

$(c)$ **Population Strong PD Risk:** *The population strong PD risk for $(\boldsymbol{\omega}^u, \boldsymbol{\nu}^u)$ by algorithm $\bar{A}_{c-c}$ is*

$$\triangle^s(\boldsymbol{\omega}^u, \boldsymbol{\nu}^u) \leq \mathcal{O}\left((L^3\ell^3\rho/\lambda^6 + L^2\ell^2/\lambda^3) \cdot \frac{m^2\sqrt{d\log(1/\delta)}}{n^2\epsilon} + \frac{mL^2}{\lambda n} + \frac{L^2\ell}{\lambda^2 n} + \lambda(B_{\boldsymbol{\omega}}^2 + B_{\boldsymbol{\nu}}^2)\right). (28)$$

*In particular, by setting $\lambda$ below*

$$\lambda = \max\left\{\frac{L}{\sqrt{B_{\boldsymbol{\omega}}^2 + B_{\boldsymbol{\nu}}^2}}\sqrt{\frac{m}{n}}, \left(\frac{L^2\ell}{(B_{\boldsymbol{\omega}}^2 + B_{\boldsymbol{\nu}}^2)n}\right), \left(\frac{L^2\ell^2 m^2\sqrt{d\log(1/\delta)}}{(B_{\boldsymbol{\omega}}^2 + B_{\boldsymbol{\nu}}^2)n^2\epsilon}\right)^{1/4}, \right.$$
$$\left.\left(\frac{L^3\ell^3\rho m^2\sqrt{d\log(1/\delta)}}{(B_{\boldsymbol{\omega}}^2 + B_{\boldsymbol{\nu}}^2)n^2\epsilon}\right)^{1/7}\right\}, \quad (29)$$

*we have the following population strong PD risk,*

$$\triangle^s(\boldsymbol{\omega}^u, \boldsymbol{\nu}^u) \leq \mathcal{O}\left(c_1\sqrt{\frac{m}{n}} + c_2\frac{1}{\sqrt[3]{n}} + c_3\left(\frac{d\log(1/\delta)}{\epsilon^2}\right)^{1/8}\sqrt{\frac{m}{n}} + c_4\left(\frac{\sqrt{d\log(1/\delta)}}{\epsilon}\right)^{1/7}\left(\frac{m}{n}\right)^{2/7}\right),$$
$$(30)$$

*where $c_1, c_2, c_3, c_4$ are constants that depend only on $L, l, \rho, B_{\boldsymbol{\omega}}$ and $B_{\boldsymbol{\nu}}$.*

$(d)$ **Deletion Capacity:** *The deletion capacity of Algorithm $\bar{A}_{c-c}$ is*

$$m_{\epsilon,\delta,\gamma}^{A,\bar{A}}(d_1, d_2, n) \geq c \cdot \frac{n\sqrt{\epsilon}}{(d\log(1/\delta))^{1/4}}, \quad (31)$$

*where the constant $c$ depends on the constants $L, l, \rho, B_{\boldsymbol{\omega}}$ and $B_{\boldsymbol{\nu}}$.*

## 6 Conclusion

In this paper, we have studied the certified machine unlearning for minimax models with a focus on the generalization rates and deletion capacity, while existing works in this area largely focus on standard statistical learning models. We have provided a new minimax unlearning algorithm composed of the total Hessian-based complete Newton update and the Gaussian mechanism-based perturbation, which comes with rigorous $(\epsilon, \delta)$-unlearning certification. We have established generalization results in terms of the population weak and strong primal-dual risk and the correspondingly defined deletion capacity results for the strongly-convex-strongly-concave loss functions, both of which match the state-of-the-art results obtained for standard statistical learning models. We have also provided extensions to other loss types like the convex-concave loss function. In addition, we have provided a more computationally efficient extension by getting rid of the total Hessian re-computation during the minimax unlearning phase, which can be more appealing for the successive data deletion setting. Although our bound for deletion capacity is better than that of DP by an order of $d^{1/4}$ and matches the state-of-the-art result established for unlearning under the STL setting, it remains unclear whether this bound is tight or not. In future work, we plan to extend to more general settings like the nonconvex-nonconcave loss function setting.

## Acknowledgments and Disclosure of Funding

We would like to thank Gautam Kamath for his valuable comments on the presentation of the results in the previous version of the paper. Additionally, we extend our thanks to the reviewers and area chair of NeurIPS 2023 for their constructive comments and feedback. This work was supported in part by the National Natural Science Foundation of China (62072395, 62206207, U20A20178), and the National Key Research and Development Program of China (2020AAA0107705, 2021YFB3100300).

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

# A  Additional Definitions and Supporting Lemmas

In this section, we provide additional definitions and supporting lemmas. In the next two sections, Sec.B contains missing proofs in Sec.4 and the online extension to support successive unlearning setting. Sec.C contains missing proofs in Sec.5, as well as detailed algorithm descriptions for the general convex-concave loss function setting.

## A.1  Additional Definitions

We first recall the following standard definitions for the loss function $f(\boldsymbol{\omega}, \boldsymbol{\nu}; z)$ from optimization literature.

**Definition 4** (**Function Lipschitz Continuity**). *The function $f(\boldsymbol{\omega}, \boldsymbol{\nu}; z)$ is $L$-Lipschitz, i.e., there exists a constant $L > 0$ such that for all $\boldsymbol{\omega}, \boldsymbol{\omega}' \in \mathcal{W}$, $\boldsymbol{\nu}, \boldsymbol{\nu}' \in \mathcal{V}$ and $z \in \mathcal{Z}$, it holds that*

$$\|f(\boldsymbol{\omega}, \boldsymbol{\nu}; z) - f(\boldsymbol{\omega}', \boldsymbol{\nu}'; z)\|^2 \leq L^2(\|\boldsymbol{\omega} - \boldsymbol{\omega}'\|^2 + \|\boldsymbol{\nu} - \boldsymbol{\nu}'\|^2). \tag{32}$$

**Definition 5** (**Gradient Lipschitz Continuity**). *The function $f(\boldsymbol{\omega}, \boldsymbol{\nu}; z)$ has $\ell$-Lipschitz gradients, i.e., there exists a constant $\ell > 0$ such that for all $\boldsymbol{\omega}, \boldsymbol{\omega}' \in \mathcal{W}$, $\boldsymbol{\nu}, \boldsymbol{\nu}' \in \mathcal{V}$ and $z \in \mathcal{Z}$, it holds that*

$$\|\nabla f(\boldsymbol{\omega}, \boldsymbol{\nu}; z) - \nabla f(\boldsymbol{\omega}', \boldsymbol{\nu}'; z)\|^2 \leq \ell^2(\|\boldsymbol{\omega} - \boldsymbol{\omega}'\|^2 + \|\boldsymbol{\nu} - \boldsymbol{\nu}'\|^2), \tag{33}$$

*where recall that $\nabla f(\boldsymbol{\omega}, \boldsymbol{\nu}; z) = \begin{bmatrix} \nabla_{\boldsymbol{\omega}} f(\boldsymbol{\omega}, \boldsymbol{\nu}; z) \\ \nabla_{\boldsymbol{\nu}} f(\boldsymbol{\omega}, \boldsymbol{\nu}; z) \end{bmatrix}$.*

**Definition 6** (**Hessian Lipschitz Continuity**). *The function $f(\boldsymbol{\omega}, \boldsymbol{\nu}; z)$ has $\rho$-Lipschitz Hessian, i.e., there exists a constant $\rho > 0$ such that for all $\boldsymbol{\omega}, \boldsymbol{\omega}' \in \mathcal{W}$, $\boldsymbol{\nu}, \boldsymbol{\nu}' \in \mathcal{V}$ and $z \in \mathcal{Z}$, it holds that*

$$\|\nabla^2 f(\boldsymbol{\omega}, \boldsymbol{\nu}; z) - \nabla^2 f(\boldsymbol{\omega}', \boldsymbol{\nu}'; z)\|^2 \leq \rho^2(\|\boldsymbol{\omega} - \boldsymbol{\omega}'\|^2 + \|\boldsymbol{\nu} - \boldsymbol{\nu}'\|^2), \tag{34}$$

*where recall that $\nabla^2 f(\boldsymbol{\omega}, \boldsymbol{\nu}; z) = \begin{bmatrix} \partial_{\boldsymbol{\omega}\boldsymbol{\omega}} f(\boldsymbol{\omega}, \boldsymbol{\nu}; z) & \partial_{\boldsymbol{\omega}\boldsymbol{\nu}} f(\boldsymbol{\omega}, \boldsymbol{\nu}; z) \\ \partial_{\boldsymbol{\nu}\boldsymbol{\omega}} f(\boldsymbol{\omega}, \boldsymbol{\nu}; z) & \partial_{\boldsymbol{\nu}\boldsymbol{\nu}} f(\boldsymbol{\omega}, \boldsymbol{\nu}; z) \end{bmatrix}$.*

**Definition 7** (**Strongly-Convex-Strongly-Concave Objective Function**). *The function $f(\boldsymbol{\omega}, \boldsymbol{\nu}; z)$ is $\mu_{\boldsymbol{\omega}}$-strongly convex on $\mathcal{W}$ and $\mu_{\boldsymbol{\nu}}$-strongly concave on $\mathcal{V}$, i.e., there exist constants $\mu_{\boldsymbol{\omega}} > 0$ and $\mu_{\boldsymbol{\nu}} > 0$ such that for all $\boldsymbol{\omega}, \boldsymbol{\omega}' \in \mathcal{W}$, $\boldsymbol{\nu}, \boldsymbol{\nu}' \in \mathcal{V}$ and $z \in \mathcal{Z}$, it holds that*

$$\begin{cases} f(\boldsymbol{\omega}, \boldsymbol{\nu}; z) \geq f(\boldsymbol{\omega}', \boldsymbol{\nu}; z) + \langle \nabla_{\boldsymbol{\omega}} f(\boldsymbol{\omega}', \boldsymbol{\nu}; z), \boldsymbol{\omega} - \boldsymbol{\omega}' \rangle + \frac{\mu_{\boldsymbol{\omega}}}{2} \|\boldsymbol{\omega} - \boldsymbol{\omega}'\|^2, \\ f(\boldsymbol{\omega}, \boldsymbol{\nu}; z) \leq f(\boldsymbol{\omega}, \boldsymbol{\nu}'; z) + \langle \nabla_{\boldsymbol{\nu}} f(\boldsymbol{\omega}, \boldsymbol{\nu}'; z), \boldsymbol{\nu} - \boldsymbol{\nu}' \rangle - \frac{\mu_{\boldsymbol{\nu}}}{2} \|\boldsymbol{\nu} - \boldsymbol{\nu}'\|^2. \end{cases} \tag{35}$$

**Definition 8** (**Best Response Auxiliary Functions**). *We introduce auxiliary functions $\mathtt{V}_S(\boldsymbol{\omega})$ and $\mathtt{V}_{S\backslash}(\boldsymbol{\omega})$, given by*

$$\mathtt{V}_S(\boldsymbol{\omega}) := \underset{\boldsymbol{\nu} \in \mathcal{V}}{\operatorname{argmax}} F_S(\boldsymbol{\omega}, \boldsymbol{\nu}), \qquad \mathtt{V}_{S\backslash}(\boldsymbol{\omega}) := \underset{\boldsymbol{\nu} \in \mathcal{V}}{\operatorname{argmax}} F_{S\backslash}(\boldsymbol{\omega}, \boldsymbol{\nu}), \tag{36}$$

*and we have $\boldsymbol{\nu}_S^* = \mathtt{V}_S(\boldsymbol{\omega}_S^*)$ and $\boldsymbol{\nu}_{S\backslash}^* = \mathtt{V}_{S\backslash}(\boldsymbol{\omega}_{S\backslash}^*)$. We can similarly introduce $\mathtt{W}_S(\boldsymbol{\nu})$ and $\mathtt{W}_{S\backslash}(\boldsymbol{\nu})$ as*

$$\mathtt{W}_S(\boldsymbol{\nu}) := \underset{\boldsymbol{\omega} \in \mathcal{W}}{\operatorname{argmin}} F_S(\boldsymbol{\omega}, \boldsymbol{\nu}), \qquad \mathtt{W}_{S\backslash}(\boldsymbol{\nu}) := \underset{\boldsymbol{\omega} \in \mathcal{W}}{\operatorname{argmin}} F_{S\backslash}(\boldsymbol{\omega}, \boldsymbol{\nu}), \tag{37}$$

*and we have $\boldsymbol{\omega}_S^* = \mathtt{W}_S(\boldsymbol{\nu}_S^*)$ and $\boldsymbol{\omega}_{S\backslash}^* = \mathtt{W}_{S\backslash}(\boldsymbol{\nu}_{S\backslash}^*)$ by this definition.*

*In addition, we define the primal function $P(\boldsymbol{\omega}) := \max_{\boldsymbol{\nu} \in \mathcal{V}} F_S(\boldsymbol{\omega}, \boldsymbol{\nu}) = F_S(\boldsymbol{\omega}, \mathtt{V}_S(\boldsymbol{\omega}))$, which has gradient $\nabla P(\boldsymbol{\omega}) = \nabla_{\boldsymbol{\omega}} F_S(\boldsymbol{\omega}, \mathtt{V}_S(\boldsymbol{\omega}))$ and Hessian $\nabla_{\boldsymbol{\omega}\boldsymbol{\omega}}^2 P(\boldsymbol{\omega}) = \mathtt{D}_{\boldsymbol{\omega}\boldsymbol{\omega}} F_S(\boldsymbol{\omega}, \mathtt{V}_S(\boldsymbol{\omega}))$ (i.e., the total Hessian of $F_S$). The dual function, its gradient, and Hessian can be similarly defined, e.g., $D(\boldsymbol{\nu}) := \min_{\boldsymbol{\omega} \in \mathcal{W}} F_S(\boldsymbol{\omega}, \boldsymbol{\nu}) = F_S(\mathtt{W}_S(\boldsymbol{\nu}), \boldsymbol{\nu})$.*

## A.2  Supporting Lemmas

The following lemma provides the distance between $\mathtt{V}_S(\boldsymbol{\omega}_S^*)$ and $\mathtt{V}_{S\backslash}(\boldsymbol{\omega}_S^*)$. Similar result can be derived for the distance between $\mathtt{W}_S(\boldsymbol{\nu}_S^*)$ and $\mathtt{W}_{S\backslash}(\boldsymbol{\nu}_S^*)$.

**Lemma 2.** *Under Assumption 1 and Assumption 2, the variables $\mathtt{V}_{S\backslash}(\boldsymbol{\omega}_S^*)$ and $\boldsymbol{\nu}_S^*$ (i.e., $\mathtt{V}_S(\boldsymbol{\omega}_S^*)$) defined in Algorithm 1 satisfy the following distance bound*

$$\|\boldsymbol{\nu}_S^* - \mathtt{V}_{S\backslash}(\boldsymbol{\omega}_S^*)\| \leq \frac{2Lm}{\mu_{\boldsymbol{\nu}}(n - m)}. \tag{38}$$

*Proof.* We observe that

$$(n-m)(F_{S\setminus}(\boldsymbol{\omega}_S^*, \mathtt{V}_{S\setminus}(\boldsymbol{\omega}_S^*)) - F_{S\setminus}(\boldsymbol{\omega}_S^*, \boldsymbol{\nu}_S^*))$$

$$= \sum_{z_i \in S\setminus U} f(\boldsymbol{\omega}_S^*, \mathtt{V}_{S\setminus}(\boldsymbol{\omega}_S^*); z_i) - \sum_{z_i \in S\setminus U} f(\boldsymbol{\omega}_S^*, \boldsymbol{\nu}_S^*; z_i)$$

$$= \sum_{z_i \in S} f(\boldsymbol{\omega}_S^*, \mathtt{V}_{S\setminus}(\boldsymbol{\omega}_S^*); z_i) - \sum_{z_i \in U} f(\boldsymbol{\omega}_S^*, \mathtt{V}_{S\setminus}(\boldsymbol{\omega}_S^*); z_i) - \left( \sum_{z_i \in S} f(\boldsymbol{\omega}_S^*, \boldsymbol{\nu}_S^*; z_i) - \sum_{z_i \in U} (\boldsymbol{\omega}_S^*, \boldsymbol{\nu}_S^*; z_i) \right)$$

$$= n(F_S(\boldsymbol{\omega}_S^*, \mathtt{V}_{S\setminus}(\boldsymbol{\omega}_S^*)) - F_S(\boldsymbol{\omega}_S^*, \boldsymbol{\nu}_S^*)) + \sum_{z_i \in U} f(\boldsymbol{\omega}_S^*, \boldsymbol{\nu}_S^*; z_i) - \sum_{z_i \in U} f(\boldsymbol{\omega}_S^*, \mathtt{V}_{S\setminus}(\boldsymbol{\omega}_S^*); z_i)$$

$$\overset{(i)}{\leq} \sum_{z_i \in U} f(\boldsymbol{\omega}_S^*, \boldsymbol{\nu}_S^*; z_i) - \sum_{z_i \in U} f(\boldsymbol{\omega}_S^*, \mathtt{V}_{S\setminus}(\boldsymbol{\omega}_S^*); z_i) \overset{(ii)}{\leq} mL\|\boldsymbol{\nu}_S^* - \mathtt{V}_{S\setminus}(\boldsymbol{\omega}_S^*)\|,$$

$$(39)$$

where the inequality $(i)$ follows from that $\boldsymbol{\nu}_S^*$ is the maximizer of the function $F_S(\boldsymbol{\omega}, \boldsymbol{\nu})$, thus $F_S(\boldsymbol{\omega}_S^*, \mathtt{V}_{S\setminus}(\boldsymbol{\omega}_S^*)) - F_S(\boldsymbol{\omega}_S^*, \boldsymbol{\nu}_S^*) \leq 0$. The inequality $(ii)$ is due to the fact that the function $f$ is $L$-Lipschitz. Also note that the function $F_{S\setminus}(\boldsymbol{\omega}, \boldsymbol{\nu})$ is $\mu_{\boldsymbol{\nu}}$-strongly concave, thus we have

$$F_{S\setminus}(\boldsymbol{\omega}_S^*, \mathtt{V}_{S\setminus}(\boldsymbol{\omega}_S^*)) - F_{S\setminus}(\boldsymbol{\omega}_S^*, \boldsymbol{\nu}_S^*) \geq \frac{\mu_{\boldsymbol{\nu}}}{2} \|\boldsymbol{\nu}_S^* - \mathtt{V}_{S\setminus}(\boldsymbol{\omega}_S^*)\|^2. \tag{40}$$

Eq.(39) and eq.(40) together give that

$$\frac{\mu_{\boldsymbol{\nu}}(n-m)}{2} \|\boldsymbol{\nu}_S^* - \mathtt{V}_{S\setminus}(\boldsymbol{\omega}_S^*)\|^2 \leq mL\|\boldsymbol{\nu}_S^* - \mathtt{V}_{S\setminus}(\boldsymbol{\omega}_S^*)\|, \tag{41}$$

which implies that $\|\boldsymbol{\nu}_S^* - \mathtt{V}_{S\setminus}(\boldsymbol{\omega}_S^*)\| \leq \frac{2Lm}{\mu_{\boldsymbol{\nu}}(n-m)}$. $\qquad\square$

The following lemma provides the distance between $(\boldsymbol{\omega}_{S\setminus}^*, \boldsymbol{\nu}_{S\setminus}^*)$ and $(\boldsymbol{\omega}_S^*, \boldsymbol{\nu}_S^*)$.

**Lemma 3.** *Under Assumption 1 and Assumption 2, the variables $(\boldsymbol{\omega}_{S\setminus}^*, \boldsymbol{\nu}_{S\setminus}^*)$ defined in eq.(9) and $(\boldsymbol{\omega}_S^*, \boldsymbol{\nu}_S^*)$ defined in Algorithm 1 satisfy the following guarantees*

$$\|\boldsymbol{\omega}_{S\setminus}^* - \boldsymbol{\omega}_S^*\| \leq \frac{2Lm}{\mu_{\boldsymbol{\omega}} n}, \qquad and \qquad \|\boldsymbol{\nu}_{S\setminus}^* - \boldsymbol{\nu}_S^*\| \leq \frac{2Lm}{\mu_{\boldsymbol{\nu}} n}. \tag{42}$$

*Proof.* We begin with the $\boldsymbol{\omega}$-part,

$$n[F_S(\boldsymbol{\omega}_{S\setminus}^*, \boldsymbol{\nu}_{S\setminus}^*) - F_S(\boldsymbol{\omega}_S^*, \boldsymbol{\nu}_{S\setminus}^*)]$$

$$= \sum_{z_i \in S} f(\boldsymbol{\omega}_{S\setminus}^*, \boldsymbol{\nu}_{S\setminus}^*; z_i) - \sum_{z_i \in S} f(\boldsymbol{\omega}_S^*, \boldsymbol{\nu}_{S\setminus}^*; z_i)$$

$$= \sum_{z_i \in S\setminus U} f(\boldsymbol{\omega}_{S\setminus}^*, \boldsymbol{\nu}_{S\setminus}^*; z_i) + \sum_{z_i \in U} f(\boldsymbol{\omega}_{S\setminus}^*, \boldsymbol{\nu}_{S\setminus}^*; z_i) - \sum_{z_i \in S\setminus U} f(\boldsymbol{\omega}_S^*, \boldsymbol{\nu}_{S\setminus}^*; z_i) - \sum_{z_i \in U} f(\boldsymbol{\omega}_S^*, \boldsymbol{\nu}_{S\setminus}^*; z_i)$$

$$= (n-m)[F_{S\setminus}(\boldsymbol{\omega}_{S\setminus}^*, \boldsymbol{\nu}_{S\setminus}^*) - F_{S\setminus}(\boldsymbol{\omega}_S^*, \boldsymbol{\nu}_{S\setminus}^*)] + \sum_{z_i \in U} f(\boldsymbol{\omega}_{S\setminus}^*, \boldsymbol{\nu}_{S\setminus}^*; z_i) - \sum_{z_i \in U} f(\boldsymbol{\omega}_S^*, \boldsymbol{\nu}_{S\setminus}^*; z_i)$$

$$\overset{(i)}{\leq} \sum_{z_i \in U} f(\boldsymbol{\omega}_{S\setminus}^*, \boldsymbol{\nu}_{S\setminus}^*; z_i) - \sum_{z_i \in U} f(\boldsymbol{\omega}_S^*, \boldsymbol{\nu}_{S\setminus}^*; z_i) \overset{(ii)}{\leq} mL\|\boldsymbol{\omega}_{S\setminus}^* - \boldsymbol{\omega}_S^*\|,$$

$$(43)$$

where the inequality $(i)$ holds because $\boldsymbol{\omega}_{S\setminus}^*$ is the minimizer of the function $F_{S\setminus}(\boldsymbol{\omega}, \boldsymbol{\nu})$, thus $F_{S\setminus}(\boldsymbol{\omega}_{S\setminus}^*, \boldsymbol{\nu}_{S\setminus}^*) - F_{S\setminus}(\boldsymbol{\omega}_S^*, \boldsymbol{\nu}_{S\setminus}^*) \leq 0$, and the inequality $(ii)$ follows from the fact that the function $f$ is $L$-Lipschitz. Since the function $F_S(\boldsymbol{\omega}, \boldsymbol{\nu})$ is $\mu_{\boldsymbol{\omega}}$-strongly convex, we further get

$$F_S(\boldsymbol{\omega}_{S\setminus}^*, \boldsymbol{\nu}_{S\setminus}^*) - F_S(\boldsymbol{\omega}_S^*, \boldsymbol{\nu}_{S\setminus}^*) \geq \frac{\mu_{\boldsymbol{\omega}}}{2} \|\boldsymbol{\omega}_{S\setminus}^* - \boldsymbol{\omega}_S^*\|^2. \tag{44}$$

Eq.(43) and eq.(44) together gives that

$$\frac{\mu_{\boldsymbol{\omega}} n}{2} \|\boldsymbol{\omega}_{S\setminus}^* - \boldsymbol{\omega}_S^*\|^2 \leq mL\|\boldsymbol{\omega}_{S\setminus}^* - \boldsymbol{\omega}_S^*\|. \tag{45}$$

Thus, we get $\|\boldsymbol{\omega}_{S\backslash}^* - \boldsymbol{\omega}_S^*\| \leq \frac{2Lm}{\mu_{\boldsymbol{\omega}} n}$.

For the $\boldsymbol{\nu}$-part, we similarly have

$$
\begin{aligned}
&n[F_S(\boldsymbol{\omega}_{S\backslash}^*, \boldsymbol{\nu}_S^*) - F_S(\boldsymbol{\omega}_{S\backslash}^*, \boldsymbol{\nu}_{S\backslash}^*)] \\
&= \sum_{z_i \in S} f(\boldsymbol{\omega}_{S\backslash}^*, \boldsymbol{\nu}_S^*; z_i) - \sum_{z_i \in S} f(\boldsymbol{\omega}_{S\backslash}^*, \boldsymbol{\nu}_{S\backslash}^*; z_i) \\
&= \sum_{z_i \in S\backslash U} f(\boldsymbol{\omega}_{S\backslash}^*, \boldsymbol{\nu}_S^*; z_i) + \sum_{z_i \in U} f(\boldsymbol{\omega}_{S\backslash}^*, \boldsymbol{\nu}_S^*; z_i) - \sum_{z_i \in S\backslash U} f(\boldsymbol{\omega}_{S\backslash}^*, \boldsymbol{\nu}_{S\backslash}^*; z_i) - \sum_{z_i \in U} f(\boldsymbol{\omega}_{S\backslash}^*, \boldsymbol{\nu}_{S\backslash}^*; z_i) \\
&= (n-m)[F_{S\backslash}(\boldsymbol{\omega}_{S\backslash}^*, \boldsymbol{\nu}_S^*) - F_{S\backslash}(\boldsymbol{\omega}_{S\backslash}^*, \boldsymbol{\nu}_{S\backslash}^*)] + \sum_{z_i \in U} f(\boldsymbol{\omega}_{S\backslash}^*, \boldsymbol{\nu}_S^*; z_i) - \sum_{z_i \in U} f(\boldsymbol{\omega}_{S\backslash}^*, \boldsymbol{\nu}_{S\backslash}^*; z_i) \\
&\overset{(i)}{\leq} \sum_{z_i \in U} f(\boldsymbol{\omega}_{S\backslash}^*, \boldsymbol{\nu}_S^*; z_i) - \sum_{z_i \in U} f(\boldsymbol{\omega}_{S\backslash}^*, \boldsymbol{\nu}_{S\backslash}^*; z_i) \overset{(ii)}{\leq} mL\|\boldsymbol{\nu}_S^* - \boldsymbol{\nu}_{S\backslash}^*\|,
\end{aligned}
\tag{46}
$$

where the inequality $(i)$ follows from that $\boldsymbol{\nu}_{S\backslash}^*$ is the maximizer of the function $F_{S\backslash}(\boldsymbol{\omega}, \boldsymbol{\nu})$, thus $F_{S\backslash}(\boldsymbol{\omega}_{S\backslash}^*, \boldsymbol{\nu}_S^*) - F_{S\backslash}(\boldsymbol{\omega}_{S\backslash}^*, \boldsymbol{\nu}_{S\backslash}^*) \leq 0$. The inequality $(ii)$ is due to the fact that the function $f$ is $L$-Lipschitz. In addition, by the strongly-concave assumption of $F_S(\boldsymbol{\omega}, \boldsymbol{\nu})$ is $\mu_{\boldsymbol{\nu}}$, we have

$$
F_S(\boldsymbol{\omega}_{S\backslash}^*, \boldsymbol{\nu}_S^*) - F_S(\boldsymbol{\omega}_{S\backslash}^*, \boldsymbol{\nu}_{S\backslash}^*) \geq \frac{\mu_{\boldsymbol{\nu}}}{2}\|\boldsymbol{\nu}_{S\backslash}^* - \boldsymbol{\nu}_S^*\|^2. \tag{47}
$$

By eq.(46) and eq.(47), we get that

$$
\frac{\mu_{\boldsymbol{\nu}} n}{2}\|\boldsymbol{\nu}_{S\backslash}^* - \boldsymbol{\nu}_S^*\|^2 \leq mL\|\boldsymbol{\nu}_S^* - \boldsymbol{\nu}_{S\backslash}^*\|. \tag{48}
$$

Thus, we have $\|\boldsymbol{\nu}_{S\backslash}^* - \boldsymbol{\nu}_S^*\| \leq \frac{2Lm}{\mu_{\boldsymbol{\nu}} n}$. $\qquad\square$

In the following, we recall several lemmas (i.e., Lemma 4 to Lemma 8) from existing minimax optimization literature for completeness.

**Lemma 4** ([Lin et al., 2020, Lemma 4.3]). *Under Assumption 1 and Assumption 2, for any $\boldsymbol{\omega} \in \mathcal{W}$, the function $V_S(\boldsymbol{\omega})$ is $(\ell/\mu_{\boldsymbol{\nu}})$-Lipschitz.*

*Proof.* By the optimality condition of the function $V_S(\boldsymbol{\omega})$, we have

$$
\begin{aligned}
\langle \nabla_{\boldsymbol{\nu}} F_S(\boldsymbol{\omega}_1, V_S(\boldsymbol{\omega}_1)), V_S(\boldsymbol{\omega}_2) - V_S(\boldsymbol{\omega}_1) \rangle &\leq 0, \\
\langle \nabla_{\boldsymbol{\nu}} F_S(\boldsymbol{\omega}_2, V_S(\boldsymbol{\omega}_2)), V_S(\boldsymbol{\omega}_1) - V_S(\boldsymbol{\omega}_2) \rangle &\leq 0.
\end{aligned}
$$

Summing the two inequalities above yields

$$
\langle \nabla_{\boldsymbol{\nu}} F_S(\boldsymbol{\omega}_1, V_S(\boldsymbol{\omega}_1)) - \nabla_{\boldsymbol{\nu}} F_S(\boldsymbol{\omega}_2, V_S(\boldsymbol{\omega}_2)), V_S(\boldsymbol{\omega}_2) - V_S(\boldsymbol{\omega}_1) \rangle \leq 0. \tag{49}
$$

Since the function $F_S(\boldsymbol{\omega}, \boldsymbol{\nu})$ is $\mu_{\boldsymbol{\nu}}$-strongly concave in $\boldsymbol{\nu}$, we have

$$
\langle \nabla_{\boldsymbol{\nu}} F_S(\boldsymbol{\omega}_1, V_S(\boldsymbol{\omega}_2)) - F_S(\boldsymbol{\omega}_1, V_S(\boldsymbol{\omega}_1)), V_S(\boldsymbol{\omega}_2) - V_S(\boldsymbol{\omega}_1) \rangle + \mu_{\boldsymbol{\nu}}\|V_S(\boldsymbol{\omega}_2) - V_S(\boldsymbol{\omega}_1)\|^2 \leq 0. \tag{50}
$$

By eq.(49) and eq.(50) with the $\ell$-Lipschitz continuity of $\nabla F_S(\boldsymbol{\omega}, \boldsymbol{\nu})$, we further get

$$
\begin{aligned}
\mu_{\boldsymbol{\nu}}\|V_S(\boldsymbol{\omega}_2) - V_S(\boldsymbol{\omega}_1)\|^2 &\leq \langle \nabla_{\boldsymbol{\nu}} F_S(\boldsymbol{\omega}_2, V_S(\boldsymbol{\omega}_2)) - \nabla_{\boldsymbol{\nu}} F_S(\boldsymbol{\omega}_1, V_S(\boldsymbol{\omega}_2)), V_S(\boldsymbol{\omega}_2) - V_S(\boldsymbol{\omega}_1) \rangle \\
&\leq \ell\|\boldsymbol{\omega}_2 - \boldsymbol{\omega}_1\| \cdot \|V_S(\boldsymbol{\omega}_2) - V_S(\boldsymbol{\omega}_1)\|.
\end{aligned}
\tag{51}
$$

Consequently, we have

$$
\|V_S(\boldsymbol{\omega}_2) - V_S(\boldsymbol{\omega}_1)\| \leq \frac{\ell}{\mu_{\boldsymbol{\nu}}}\|\boldsymbol{\omega}_2 - \boldsymbol{\omega}_1\|. \tag{52}
$$

$\qquad\square$

**Remark 1.** *The above lemma can be similarly derived for $W_S$ to obtain that the best response auxiliary function $W_S(\boldsymbol{\nu})$ is $(\ell/\mu_{\boldsymbol{\omega}})$-Lipschitz. In the next three lemmas, we focus on the $\boldsymbol{\omega}$-part and omit the $\boldsymbol{\nu}$-part.*

**Lemma 5** ([Luo et al., 2022, Lemma 3]). *Denote $\kappa_{\boldsymbol{\nu}} = \ell/\mu_{\boldsymbol{\nu}}$. Under Assumption 1 and Assumption 2, for any $\boldsymbol{\omega}, \boldsymbol{\omega}' \in \mathcal{W}$, we have*

$$\|\mathtt{D}_{\boldsymbol{\omega\omega}}F_S(\boldsymbol{\omega}, \mathtt{V}_S(\boldsymbol{\omega})) - \mathtt{D}_{\boldsymbol{\omega\omega}}F_S(\boldsymbol{\omega}', \mathtt{V}_S(\boldsymbol{\omega}'))\| \leq 4\sqrt{2}\kappa_{\boldsymbol{\nu}}^3\rho\|\boldsymbol{\omega} - \boldsymbol{\omega}'\|. \tag{53}$$

**Lemma 6** ([Nesterov and Polyak, 2006, Lemma 1]). *Denote $\kappa_{\boldsymbol{\nu}} = \ell/\mu_{\boldsymbol{\nu}}$. Under Assumption 1 and Assumption 2, for any $\boldsymbol{\omega}, \boldsymbol{\omega}' \in \mathcal{W}$, we have*

$$\|\nabla_{\boldsymbol{\omega}}F_S(\boldsymbol{\omega}, \mathtt{V}_S(\boldsymbol{\omega})) - \nabla_{\boldsymbol{\omega}}F_S(\boldsymbol{\omega}', \mathtt{V}_S(\boldsymbol{\omega}')) - \mathtt{D}_{\boldsymbol{\omega\omega}}F_S(\boldsymbol{\omega}')(\boldsymbol{\omega} - \boldsymbol{\omega}')\| \leq \frac{M}{2}\|\boldsymbol{\omega} - \boldsymbol{\omega}'\|^2, \tag{54}$$

*where $M = 4\sqrt{2}\kappa_{\boldsymbol{\nu}}^3\rho$.*

*Proof.* Recall the definition of the primal function $P(\boldsymbol{\omega}) := \max_{\boldsymbol{\nu}\in\mathcal{V}} F_S(\boldsymbol{\omega}, \boldsymbol{\nu})$ and its gradient $\nabla P(\boldsymbol{\omega}) = \nabla_{\boldsymbol{\omega}}F_S(\boldsymbol{\omega}, \mathtt{V}_S(\boldsymbol{\omega}))$. Due to the optimality of $\mathtt{V}_S$, we have

$$\nabla_{\boldsymbol{\nu}}F_S(\boldsymbol{\omega}, \mathtt{V}_S(\boldsymbol{\omega})) = 0. \tag{55}$$

By taking the total derivative with respect to $\boldsymbol{\omega}$, we get

$$\partial_{\boldsymbol{\nu\omega}}F_S(\boldsymbol{\omega}, \mathtt{V}_S(\boldsymbol{\omega})) + \partial_{\boldsymbol{\nu\nu}}F_S(\boldsymbol{\omega}, \mathtt{V}_S(\boldsymbol{\omega}))\nabla\mathtt{V}_S(\boldsymbol{\omega}) = 0. \tag{56}$$

Taking the total derivative of $\boldsymbol{\omega}$ again on $\nabla P(\boldsymbol{\omega})$, we further have

$$\begin{aligned}
\nabla^2 P(\boldsymbol{\omega}) =& \partial_{\boldsymbol{\omega\omega}}F_S(\boldsymbol{\omega}, \mathtt{V}_S(\boldsymbol{\omega})) + \partial_{\boldsymbol{\omega\nu}}F_S(\boldsymbol{\omega}, \mathtt{V}_S(\boldsymbol{\omega}))\nabla\mathtt{V}_S(\boldsymbol{\omega}) \\
=& \partial_{\boldsymbol{\omega\omega}}F_S(\boldsymbol{\omega}, \mathtt{V}_S(\boldsymbol{\omega})) - \partial_{\boldsymbol{\omega\nu}}F_S(\boldsymbol{\omega}, \mathtt{V}_S(\boldsymbol{\omega}))\partial_{\boldsymbol{\nu\nu}}^{-1}F_S(\boldsymbol{\omega}, \mathtt{V}_S(\boldsymbol{\omega}))\partial_{\boldsymbol{\nu\omega}}F_S(\boldsymbol{\omega}, \mathtt{V}_S(\boldsymbol{\omega})) \\
=& \mathtt{D}_{\boldsymbol{\omega\omega}}F_S(\boldsymbol{\omega}, \mathtt{V}_S(\boldsymbol{\omega})).
\end{aligned} \tag{57}$$

Based on the equality of $\nabla^2 P(\boldsymbol{\omega})$ and $\mathtt{D}_{\boldsymbol{\omega\omega}}F_S(\boldsymbol{\omega}, \mathtt{V}_S(\boldsymbol{\omega}))$ above and Lemma 5, we have

$$\begin{aligned}
& \|\nabla_{\boldsymbol{\omega}}F_S(\boldsymbol{\omega}, \mathtt{V}_S(\boldsymbol{\omega})) - \nabla_{\boldsymbol{\omega}}F_S(\boldsymbol{\omega}', \mathtt{V}_S(\boldsymbol{\omega}')) - \mathtt{D}_{\boldsymbol{\omega\omega}}F_S(\boldsymbol{\omega}')(\boldsymbol{\omega} - \boldsymbol{\omega}')\| \\
=& \|\nabla P(\boldsymbol{\omega}) - \nabla P(\boldsymbol{\omega}') - \nabla^2 P(\boldsymbol{\omega}')(\boldsymbol{\omega} - \boldsymbol{\omega}')\| \\
\leq& \frac{M}{2}\|\boldsymbol{\omega} - \boldsymbol{\omega}'\|^2.
\end{aligned} \tag{58}$$

$\square$

**Lemma 7.** *Under Assumption 1 and Assumption 2, for all $\boldsymbol{\omega} \in \mathcal{W}$ and $\boldsymbol{\nu} \in \mathcal{V}$, we have $\|\mathtt{D}_{\boldsymbol{\omega\omega}}f(\boldsymbol{\omega}, \boldsymbol{\nu}; z)\| \leq \ell + \frac{\ell^2}{\mu_{\boldsymbol{\nu}}}$.*

*Proof.* By the definition of the total Hessian, we have

$$\begin{aligned}
\|\mathtt{D}_{\boldsymbol{\omega\omega}}f(\boldsymbol{\omega}, \boldsymbol{\nu}; z)\| =& \|\partial_{\boldsymbol{\omega\omega}}f(\boldsymbol{\omega}, \boldsymbol{\nu}; z) - \partial_{\boldsymbol{\omega\nu}}f(\boldsymbol{\omega}, \boldsymbol{\nu}; z)\partial_{\boldsymbol{\nu\nu}}^{-1}f(\boldsymbol{\omega}, \boldsymbol{\nu}; z)\partial_{\boldsymbol{\nu\omega}}f(\boldsymbol{\omega}, \boldsymbol{\nu}; z)\| \\
\overset{(i)}{\leq}& \|\partial_{\boldsymbol{\omega\omega}}f(\boldsymbol{\omega}, \boldsymbol{\nu}; z)\| + \|\partial_{\boldsymbol{\omega\nu}}f(\boldsymbol{\omega}, \boldsymbol{\nu}; z)\partial_{\boldsymbol{\nu\nu}}^{-1}f(\boldsymbol{\omega}, \boldsymbol{\nu}; z)\partial_{\boldsymbol{\nu\omega}}f(\boldsymbol{\omega}, \boldsymbol{\nu}; z)\| \\
\overset{(ii)}{\leq}& \ell + \ell \cdot \mu_{\boldsymbol{\nu}}^{-1} \cdot \ell = \ell + \frac{\ell^2}{\mu_{\boldsymbol{\nu}}},
\end{aligned} \tag{59}$$

where the inequality $(i)$ uses the triangle inequality and the inequality $(ii)$ is due to the function $f$ has $\ell$-Lipschitz gradients and $f$ is $\mu_{\boldsymbol{\nu}}$-strongly concave in $\boldsymbol{\nu}$, thus we have $\|\partial_{\boldsymbol{\omega\omega}}f(\boldsymbol{\omega}, \boldsymbol{\nu}; z)\| \leq \ell$, $\|\partial_{\boldsymbol{\omega\nu}}f(\boldsymbol{\omega}, \boldsymbol{\nu}; z)\| \leq \ell$, $\|\partial_{\boldsymbol{\nu\omega}}f(\boldsymbol{\omega}, \boldsymbol{\nu}; z)\| \leq \ell$ and $\|\partial_{\boldsymbol{\nu\nu}}f(\boldsymbol{\omega}, \boldsymbol{\nu}; z)\| \leq \mu_{\boldsymbol{\nu}}^{-1}$. $\square$

**Lemma 8** ([Zhang et al., 2022a, Lemma 4.4]). *Under Assumption 1 and Assumption 2, the population weak PD risk for the minimax learning variables $(\boldsymbol{\omega}_S^*, \boldsymbol{\nu}_S^*)$ returned by Algorithm 1 has*

$$\triangle^w(\boldsymbol{\omega}_S^*, \boldsymbol{\nu}_S^*) \leq \frac{2\sqrt{2}L^2}{\mu n}. \tag{60}$$

**Lemma 9** ([Zhang et al., 2021, Theorem 2]). *Under Assumption 1 and Assumption 2, the population strong PD risk for the minimax learning variables $(\boldsymbol{\omega}_S^*, \boldsymbol{\nu}_S^*)$ returned by Algorithm 1 has*

$$\triangle^s(\boldsymbol{\omega}_S^*, \boldsymbol{\nu}_S^*) \leq \frac{2\sqrt{2}L^2}{n} \cdot \sqrt{\frac{\ell^2}{\mu_{\boldsymbol{\omega}}\mu_{\boldsymbol{\nu}}} + 1} \cdot \left(\frac{1}{\mu_{\boldsymbol{\omega}}} + \frac{1}{\mu_{\boldsymbol{\nu}}}\right) \leq \frac{8L^2\ell}{\mu^2 n}. \tag{61}$$

# B Detailed Algorithm Analysis and Missing Proofs in Section 4

## B.1 Analysis for Algorithm 2

In the following, we provide the analysis for Algorithm 2 in terms of guarantees of $(\epsilon, \delta)$-certified unlearning, population primal-dual risk, and deletion capacity and the corresponding proofs.

**Lemma 10** (**Closeness Upper Bound**). *Suppose the loss function $f$ satisfies Assumption 1 and 2, $\|D_{\boldsymbol{\omega\omega}} F_S(\boldsymbol{\omega}_S^*, \boldsymbol{\nu}_S^*)\| \geq \mu_{\boldsymbol{\omega\omega}}$ and $\|D_{\boldsymbol{\nu\nu}} F_S(\boldsymbol{\omega}_S^*, \boldsymbol{\nu}_S^*)\| \geq \mu_{\boldsymbol{\nu\nu}}$. Let $\mu = \min\{\mu_{\boldsymbol{\omega}}, \mu_{\boldsymbol{\nu}}, \mu_{\boldsymbol{\omega\omega}}, \mu_{\boldsymbol{\nu\nu}}\}$. Then, we have the closeness bound between $(\widehat{\boldsymbol{\omega}}, \widehat{\boldsymbol{\nu}})$ in Line 2 of Algorithm 2 and $(\boldsymbol{\omega}_{S\backslash}^*, \boldsymbol{\nu}_{S\backslash}^*)$ in eq.(9):*

$$\{\|\boldsymbol{\omega}_{S\backslash}^* - \widehat{\boldsymbol{\omega}}\|, \|\boldsymbol{\nu}_{S\backslash}^* - \widehat{\boldsymbol{\nu}}\|\} \leq \frac{(8\sqrt{2}L^2\ell^3\rho/\mu^5 + 8L\ell^2/\mu^2)m^2}{n(\mu n - (\ell + \ell^2/\mu)m)}. \tag{62}$$

*Proof.* Recall that the empirical loss functions $F_{S\backslash}(\boldsymbol{\omega}, \boldsymbol{\nu})$ and $F_S(\boldsymbol{\omega}, \boldsymbol{\nu})$ are

$$F_{S\backslash}(\boldsymbol{\omega}, \boldsymbol{\nu}) := \frac{1}{n-m} \sum_{z_i \in S\backslash U} f(\boldsymbol{\omega}, \boldsymbol{\nu}; z_i), \quad \text{and} \quad F_S(\boldsymbol{\omega}, \boldsymbol{\nu}) := \frac{1}{n} \sum_{z_i \in S} f(\boldsymbol{\omega}, \boldsymbol{\nu}; z_i). \tag{63}$$

We focus on the key term $\nabla_{\boldsymbol{\omega}} F_{S\backslash}(\boldsymbol{\omega}_{S\backslash}^*, \mathsf{V}_S(\boldsymbol{\omega}_{S\backslash}^*)) - \nabla_{\boldsymbol{\omega}} F_{S\backslash}(\boldsymbol{\omega}_S^*, \boldsymbol{\nu}_S^*) - D_{\boldsymbol{\omega\omega}} F_{S\backslash}(\boldsymbol{\omega}_S^*, \boldsymbol{\nu}_S^*)(\boldsymbol{\omega}_{S\backslash}^* - \boldsymbol{\omega}_S^*)$, which has the following conversions

$$\begin{aligned}
&\|\nabla_{\boldsymbol{\omega}} F_{S\backslash}(\boldsymbol{\omega}_{S\backslash}^*, \mathsf{V}_S(\boldsymbol{\omega}_{S\backslash}^*)) - \nabla_{\boldsymbol{\omega}} F_{S\backslash}(\boldsymbol{\omega}_S^*, \boldsymbol{\nu}_S^*) - D_{\boldsymbol{\omega\omega}} F_{S\backslash}(\boldsymbol{\omega}_S^*, \boldsymbol{\nu}_S^*)(\boldsymbol{\omega}_{S\backslash}^* - \boldsymbol{\omega}_S^*)\| \\
=&\|\frac{n}{n-m}[\nabla_{\boldsymbol{\omega}} F_S(\boldsymbol{\omega}_{S\backslash}^*, \mathsf{V}_S(\boldsymbol{\omega}_{S\backslash}^*)) - \nabla_{\boldsymbol{\omega}} F_S(\boldsymbol{\omega}_S^*, \boldsymbol{\nu}_S^*) - D_{\boldsymbol{\omega\omega}} F_S(\boldsymbol{\omega}_S^*, \boldsymbol{\nu}_S^*)(\boldsymbol{\omega}_{S\backslash}^* - \boldsymbol{\omega}_S^*)] \\
&- \frac{1}{n-m} \sum_{z_i \in U} [\nabla_{\boldsymbol{\omega}} f(\boldsymbol{\omega}_{S\backslash}^*, \mathsf{V}_S(\boldsymbol{\omega}_{S\backslash}^*); z_i) - \nabla_{\boldsymbol{\omega}} f(\boldsymbol{\omega}_S^*, \boldsymbol{\nu}_S^*; z_i)] \\
&+ \frac{1}{n-m} \sum_{z_i \in U} D_{\boldsymbol{\omega\omega}} f(\boldsymbol{\omega}_S^*, \boldsymbol{\nu}_S^*; z_i)(\boldsymbol{\omega}_{S\backslash}^* - \boldsymbol{\omega}_S^*)\| \\
\leq& \frac{n}{n-m}\|\nabla_{\boldsymbol{\omega}} F_S(\boldsymbol{\omega}_{S\backslash}^*, \mathsf{V}_S(\boldsymbol{\omega}_{S\backslash}^*)) - \nabla_{\boldsymbol{\omega}} F_S(\boldsymbol{\omega}_S^*, \boldsymbol{\nu}_S^*) - D_{\boldsymbol{\omega\omega}} F_S(\boldsymbol{\omega}_S^*, \boldsymbol{\nu}_S^*)(\boldsymbol{\omega}_{S\backslash}^* - \boldsymbol{\omega}_S^*)\| \\
&+ \frac{1}{n-m} \sum_{z_i \in U} \|\nabla_{\boldsymbol{\omega}} f(\boldsymbol{\omega}_{S\backslash}^*, \mathsf{V}_S(\boldsymbol{\omega}_{S\backslash}^*); z_i) - \nabla_{\boldsymbol{\omega}} f(\boldsymbol{\omega}_S^*, \boldsymbol{\nu}_S^*; z_i)\| \\
&+ \frac{1}{n-m}\|\sum_{z_i \in U} D_{\boldsymbol{\omega\omega}} f(\boldsymbol{\omega}_S^*, \boldsymbol{\nu}_S^*; z_i)(\boldsymbol{\omega}_{S\backslash}^* - \boldsymbol{\omega}_S^*)\|.
\end{aligned} \tag{64}$$

We denote $\kappa_{\boldsymbol{\nu}} = \ell/\mu_{\boldsymbol{\nu}}$. For the first term on the right-hand side of the inequality in eq.(64), we have

$$\begin{aligned}
&\frac{n}{n-m}\|\nabla_{\boldsymbol{\omega}} F_S(\boldsymbol{\omega}_{S\backslash}^*, \mathsf{V}_S(\boldsymbol{\omega}_{S\backslash}^*)) - \nabla_{\boldsymbol{\omega}} F_S(\boldsymbol{\omega}_S^*, \boldsymbol{\nu}_S^*) - D_{\boldsymbol{\omega\omega}} F_S(\boldsymbol{\omega}_S^*, \boldsymbol{\nu}_S^*)(\boldsymbol{\omega}_{S\backslash}^* - \boldsymbol{\omega}_S^*)\| \\
&\overset{(i)}{\leq} \frac{n}{n-m} \cdot 2\sqrt{2}\kappa_{\boldsymbol{\nu}}^3 \rho\|\boldsymbol{\omega}_{S\backslash}^* - \boldsymbol{\omega}_S^*\|^2 \overset{(ii)}{\leq} \frac{8\sqrt{2}\kappa_{\boldsymbol{\nu}}^3 \rho L^2 m^2}{\mu_{\boldsymbol{\omega}}^2 n(n-m)} \leq \frac{8\sqrt{2}\rho L^2\ell^3 m^2}{\mu^5 n(n-m)},
\end{aligned} \tag{65}$$

where the inequality $(i)$ is by Lemma 6 and the inequality $(ii)$ is by Lemma 3.

For the second term on the right-hand side of the inequality in eq.(64), we have

$$\begin{aligned}
&\frac{1}{n-m} \sum_{z_i \in U} \|\nabla_{\boldsymbol{\omega}} f(\boldsymbol{\omega}_{S\backslash}^*, \mathsf{V}_S(\boldsymbol{\omega}_{S\backslash}^*); z_i) - \nabla_{\boldsymbol{\omega}} f(\boldsymbol{\omega}_S^*, \boldsymbol{\nu}_S^*; z_i)\| \\
&\overset{(i)}{\leq} \frac{1}{n-m} \cdot m\ell\sqrt{\|\boldsymbol{\omega}_{S\backslash}^* - \boldsymbol{\omega}_S^*\|^2 + \|\mathsf{V}_S(\boldsymbol{\omega}_{S\backslash}^*) - \mathsf{V}_S(\boldsymbol{\omega}_S^*)\|^2} \\
&\overset{(ii)}{\leq} \frac{1}{n-m} \cdot m\ell\sqrt{\|\boldsymbol{\omega}_{S\backslash}^* - \boldsymbol{\omega}_S^*\|^2 + \kappa_{\boldsymbol{\nu}}^2\|\boldsymbol{\omega}_{S\backslash}^* - \boldsymbol{\omega}_S^*\|^2} \\
&\overset{(iii)}{\leq} \frac{2Llm^2\sqrt{1+\kappa_{\boldsymbol{\nu}}^2}}{\mu_{\boldsymbol{\omega}} n(n-m)} \leq \frac{2\sqrt{2}Ll\kappa_{\boldsymbol{\nu}} m^2}{\mu n(n-m)} \leq \frac{2\sqrt{2}Ll^2 m^2}{\mu^2 n(n-m)},
\end{aligned} \tag{66}$$

where the inequality $(i)$ follows by the fact that the function $\nabla_{\boldsymbol{\omega}} f(\cdot, \cdot)$ is $\ell$-Lipschitz continuous and $\boldsymbol{\nu}_S^* = \mathsf{V}_S(\boldsymbol{\omega}_S^*)$. The inequality $(ii)$ holds because Lemma 4, and the inequality $(iii)$ is by Lemma 3.

For the third term on the right-hand side of the inequality in eq.(64), we have

$$\frac{1}{n-m}\|\sum_{z_i\in U}\mathtt{D}_{\boldsymbol{\omega\omega}}f(\boldsymbol{\omega}_S^*,\boldsymbol{\nu}_S^*;z_i)(\boldsymbol{\omega}_{S\backslash}^*-\boldsymbol{\omega}_S^*)\|$$

$$\leq(\ell+\frac{\ell^2}{\mu_{\boldsymbol{\nu}}})\cdot\frac{2Lm^2}{\mu_{\boldsymbol{\omega}}n(n-m)}\leq\frac{4L\ell^2m^2}{\mu^2n(n-m)},\tag{67}$$

where the first inequality is by Lemma 7. Plugging eq.(65), eq.(66) and eq.(67) into eq.(64), we further get

$$\|\nabla_{\boldsymbol{\omega}}F_{S\backslash}(\boldsymbol{\omega}_{S\backslash}^*,\mathtt{V}_S(\boldsymbol{\omega}_{S\backslash}^*))-\nabla_{\boldsymbol{\omega}}F_{S\backslash}(\boldsymbol{\omega}_S^*,\boldsymbol{\nu}_S^*)-\mathtt{D}_{\boldsymbol{\omega\omega}}F_{S\backslash}(\boldsymbol{\omega}_S^*,\boldsymbol{\nu}_S^*)(\boldsymbol{\omega}_{S\backslash}^*-\boldsymbol{\omega}_S^*)\|$$

$$\leq(8\sqrt{2}L^2\ell^3\rho/\mu^5+8L\ell^2/\mu^2)\frac{m^2}{n(n-m)}.\tag{68}$$

The above derivation yields an upper bound result. In the following, we derive a lower bound result. Let $\mathbf{x}$ be the vector satisfying the following relation,

$$\boldsymbol{\omega}_{S\backslash}^*=\boldsymbol{\omega}_S^*+\frac{1}{n-m}[\mathtt{D}_{\boldsymbol{\omega\omega}}F_{S\backslash}(\boldsymbol{\omega}_S^*,\boldsymbol{\nu}_S^*)]^{-1}\sum_{z_i\in U}\nabla_{\boldsymbol{\omega}}f(\boldsymbol{\omega}_S^*,\boldsymbol{\nu}_S^*;z_i)+\mathbf{x}.\tag{69}$$

Since we have $\nabla_{\boldsymbol{\omega}}F_{S\backslash}(\boldsymbol{\omega}_S^*,\boldsymbol{\nu}_S^*)=-\frac{1}{n-m}\sum_{z_i\in U}\nabla_{\boldsymbol{\omega}}f(\boldsymbol{\omega}_S^*,\boldsymbol{\nu}_S^*;z_i)$ and $\nabla_{\boldsymbol{\omega}}F_{S\backslash}(\boldsymbol{\omega}_{S\backslash}^*,\mathtt{V}_S(\boldsymbol{\omega}_{S\backslash}^*))=0$ due to the optimality of $\boldsymbol{\omega}_{S\backslash}^*$, plugging eq.(69) into eq.(68), we get that

$$\|\mathtt{D}_{\boldsymbol{\omega\omega}}F_{S\backslash}(\boldsymbol{\omega}_S^*,\boldsymbol{\nu}_S^*)\cdot\mathbf{x}\|\leq(8\sqrt{2}L^2\ell^3\rho/\mu^5+8L\ell^2/\mu^2)\frac{m^2}{n(n-m)}.\tag{70}$$

For the left-hand side of eq.(70), with $\|\mathtt{D}_{\boldsymbol{\omega\omega}}F_S(\boldsymbol{\omega}_S^*,\boldsymbol{\nu}_S^*)\|\geq\mu_{\boldsymbol{\omega\omega}}$, we have

$$\|\mathtt{D}_{\boldsymbol{\omega\omega}}F_{S\backslash}(\boldsymbol{\omega}_S^*,\boldsymbol{\nu}_S^*)\cdot\mathbf{x}\|=\frac{1}{n-m}\|[\sum_{z_i\in S\backslash U}\mathtt{D}_{\boldsymbol{\omega\omega}}f(\boldsymbol{\omega}_S^*,\boldsymbol{\nu}_S^*;z_i)]\cdot\mathbf{x}\|$$

$$=\frac{1}{n-m}\|[\sum_{z_i\in S}\mathtt{D}_{\boldsymbol{\omega\omega}}f(\boldsymbol{\omega}_S^*,\boldsymbol{\nu}_S^*;z_i)-\sum_{z_i\in U}\mathtt{D}_{\boldsymbol{\omega\omega}}f(\boldsymbol{\omega}_S^*,\boldsymbol{\nu}_S^*;z_i)]\cdot\mathbf{x}\|$$

$$\geq\frac{1}{n-m}\left(\|n\mathtt{D}_{\boldsymbol{\omega\omega}}F_S(\boldsymbol{\omega}_S^*,\boldsymbol{\nu}_S^*)\|-\|\sum_{z_i\in U}\mathtt{D}_{\boldsymbol{\omega\omega}}f(\boldsymbol{\omega}_S^*,\boldsymbol{\nu}_S^*;z_i)\|\right)\cdot\|\mathbf{x}\|$$

$$\geq\frac{(\mu_{\boldsymbol{\omega\omega}}n-(\ell+\ell^2/\mu_{\boldsymbol{\nu}})m)}{n-m}\|\mathbf{x}\|\geq\frac{(\mu n-(\ell+\ell^2/\mu)m)}{n-m}\|\mathbf{x}\|,\tag{71}$$

where the second inequality is by Lemma 7. Combining eq.(70), eq.(68), and the definition of the vector $\|\mathbf{x}\|$, we get that

$$\|\boldsymbol{\omega}_{S\backslash}^*-\widehat{\boldsymbol{\omega}}\|=\|\mathbf{x}\|\leq\frac{(8\sqrt{2}L^2\ell^3\rho/\mu^5+8L\ell^2/\mu^2)m^2}{n(\mu n-(\ell+\ell^2/\mu)m)}.\tag{72}$$

Symmetrically, we can get that $\|\boldsymbol{\nu}_{S\backslash}^*-\widehat{\boldsymbol{\nu}}\|\leq\frac{(8\sqrt{2}L^2\ell^3\rho/\mu^5+8L\ell^2/\mu^2)m^2}{n(\mu n-(\ell+\ell^2/\mu)m)}$. $\qquad\square$

**Theorem 6** (($\epsilon,\delta$)**-Minimax Unlearning Certification**). *Under the same settings of Lemma 10, our minimax learning algorithm $A_{sc-sc}$ and unlearning algorithm $\bar{A}_{sc-sc}$ is ($\epsilon,\delta$)-certified minimax unlearning if we choose*

$$\sigma_1\text{ and }\sigma_2=\frac{2(8\sqrt{2}L^2\ell^3\rho/\mu^5+8L\ell^2/\mu^2)m^2}{n(\mu n-(\ell+\ell^2/\mu)m)\epsilon}\sqrt{2\log(2.5/\delta)}.\tag{73}$$

*Proof.* Our proof for ($\epsilon,\delta$)-minimax unlearning certification is similar to the one used for the differential privacy guarantee of the Gaussian mechanism [Dwork et al., 2014].

Let ($\boldsymbol{\omega}_S^*,\boldsymbol{\nu}_S^*$) be the output of the learning algorithm $A_{sc-sc}$ trained on dataset $S$ and ($\boldsymbol{\omega}^u,\boldsymbol{\nu}^u$) be the output of the unlearning algorithm $\bar{A}_{sc-sc}$ running with delete requests $U$, the learned

model $(\boldsymbol{\omega}_S^*, \boldsymbol{\nu}_S^*)$, and the memory variables $T(S)$. Then we have $(\boldsymbol{\omega}_S^*, \boldsymbol{\nu}_S^*) = A_{sc-sc}(S)$ and $(\boldsymbol{\omega}^u, \boldsymbol{\nu}^u) = \bar{A}_{sc-sc}(U, A_{sc-sc}(S), T(S))$. We also denote the intermediate variables before adding noise in algorithm $\bar{A}_{sc-sc}$ as $(\widehat{\boldsymbol{\omega}}, \widehat{\boldsymbol{\nu}})$, and we have $\boldsymbol{\omega}^u = \widehat{\boldsymbol{\omega}} + \boldsymbol{\xi}_1$ and $\boldsymbol{\nu}^u = \widehat{\boldsymbol{\nu}} + \boldsymbol{\xi}_2$.

Smilarly, let $(\boldsymbol{\omega}_{S\backslash}^*, \boldsymbol{\nu}_{S\backslash}^*)$ be the output of the learning algorithm $A_{sc-sc}$ trained on dataset $S \setminus U$ and $(\boldsymbol{\omega}_{S\backslash}^u, \boldsymbol{\nu}_{S\backslash}^u)$ be the output of the unlearning algorithm $\bar{A}_{sc-sc}$ running with delete requests $\emptyset$, the learned model $(\boldsymbol{\omega}_{S\backslash}^*, \boldsymbol{\nu}_{S\backslash}^*)$, and the memory variables $T(S \setminus U)$. Then we have $(\boldsymbol{\omega}_{S\backslash}^*, \boldsymbol{\nu}_{S\backslash}^*) = A_{sc-sc}(S \setminus U)$ and $(\boldsymbol{\omega}_{S\backslash}^u, \boldsymbol{\nu}_{S\backslash}^u) = \bar{A}_{sc-sc}(\emptyset, A_{sc-sc}(S \setminus U), T(S \setminus U))$. We also denote the intermediate variables before adding noise in algorithm $\bar{A}_{sc-sc}$ as $(\widehat{\boldsymbol{\omega}}_{S\backslash}, \widehat{\boldsymbol{\nu}}_{S\backslash})$, and we have $\boldsymbol{\omega}_{S\backslash}^u = \widehat{\boldsymbol{\omega}}_{S\backslash} + \boldsymbol{\xi}_1$ and $\boldsymbol{\nu}_{S\backslash}^u = \widehat{\boldsymbol{\nu}}_{S\backslash} + \boldsymbol{\xi}_2$. Note that $\widehat{\boldsymbol{\omega}}_{S\backslash} = \boldsymbol{\omega}_{S\backslash}^*$ and $\widehat{\boldsymbol{\nu}}_{S\backslash} = \boldsymbol{\nu}_{S\backslash}^*$.

We sample the noise $\boldsymbol{\xi}_1 \sim \mathcal{N}(0, \sigma_1 \mathbf{I}_{d_1})$ and $\boldsymbol{\xi}_2 \sim \mathcal{N}(0, \sigma_2 \mathbf{I}_{d_2})$ with the scale:

$$
\begin{cases}
\sigma_1 = \|\boldsymbol{\omega}_{S\backslash}^* - \widehat{\boldsymbol{\omega}}\| \cdot \frac{\sqrt{2\log(2.5/\delta)}}{\epsilon/2} = \|\widehat{\boldsymbol{\omega}}_{S\backslash} - \widehat{\boldsymbol{\omega}}\| \cdot \frac{\sqrt{2\log(2.5/\delta)}}{\epsilon/2}, \\
\sigma_2 = \|\boldsymbol{\nu}_{S\backslash}^* - \widehat{\boldsymbol{\nu}}\| \cdot \frac{\sqrt{2\log(2.5/\delta)}}{\epsilon/2} = \|\widehat{\boldsymbol{\nu}}_{S\backslash} - \widehat{\boldsymbol{\nu}}\| \cdot \frac{\sqrt{2\log(2.5/\delta)}}{\epsilon/2},
\end{cases}
\tag{74}
$$

where $\|\boldsymbol{\omega}_{S\backslash}^* - \widehat{\boldsymbol{\omega}}\|$ and $\|\boldsymbol{\omega}_{S\backslash}^* - \widehat{\boldsymbol{\omega}}\|$ are given in Lemma 10. Then, following the same proof as Dwork et al. [2014, Theorem A.1] together with the composition property of DP [Vadhan, 2017, Lemma 7.2.3], we get that, for any set $O \subseteq \Theta$ where $\Theta := \mathcal{W} \times \mathcal{V}$,

$$
\Pr[(\widehat{\boldsymbol{\omega}}, \widehat{\boldsymbol{\nu}}) \in O] \leq e^\epsilon \Pr[(\widehat{\boldsymbol{\omega}}_{S\backslash}, \widehat{\boldsymbol{\nu}}_{S\backslash}) \in O] + \delta, \text{ and } \Pr[(\widehat{\boldsymbol{\omega}}_{S\backslash}, \widehat{\boldsymbol{\nu}}_{S\backslash}) \in O] \leq e^\epsilon \Pr[(\widehat{\boldsymbol{\omega}}, \widehat{\boldsymbol{\nu}}) \in O] + \delta,
\tag{75}
$$

which implies that the algorithm pair $A_{sc-sc}$ and $\bar{A}_{sc-sc}$ is $(\epsilon, \delta)$-certified minimax unlearning. $\qquad \square$

**Theorem 7** (**Population Primal-Dual Risk**). *Under the same settings of Lemma 10 and denote $d = \max\{d_1, d_2\}$, the population weak and strong PD risk for the certified minimax unlearning variables $(\boldsymbol{\omega}^u, \boldsymbol{\nu}^u)$ returned by Algorithm 2 are*

$$
\begin{cases}
\triangle^w(\boldsymbol{\omega}^u, \boldsymbol{\nu}^u) = \mathcal{O}\left( (L^3 \ell^3 \rho/\mu^6 + L^2 \ell^2/\mu^3) \cdot \frac{m^2 \sqrt{d \log(1/\delta)}}{n^2 \epsilon} + \frac{mL^2}{\mu n} \right), \\
\triangle^s(\boldsymbol{\omega}^u, \boldsymbol{\nu}^u) = \mathcal{O}\left( (L^3 \ell^3 \rho/\mu^6 + L^2 \ell^2/\mu^3) \cdot \frac{m^2 \sqrt{d \log(1/\delta)}}{n^2 \epsilon} + \frac{mL^2}{\mu n} + \frac{L^2 \ell}{\mu^2 n} \right).
\end{cases}
\tag{76}
$$

*Proof.* We begin with the population weak PD risk for the certified minimax unlearning variable $(\boldsymbol{\omega}^u, \boldsymbol{\nu}^u)$, which has the following conversions,

$$
\begin{aligned}
& \triangle^w(\boldsymbol{\omega}^u, \boldsymbol{\nu}^u) \\
=& \max_{\boldsymbol{\nu} \in \mathcal{V}} \mathbb{E}[F(\boldsymbol{\omega}^u, \boldsymbol{\nu})] - \min_{\boldsymbol{\omega} \in \mathcal{W}} \mathbb{E}[F(\boldsymbol{\omega}, \boldsymbol{\nu}^u)] \\
=& \max_{\boldsymbol{\nu} \in \mathcal{V}} \mathbb{E}[F(\boldsymbol{\omega}^u, \boldsymbol{\nu}) - F(\boldsymbol{\omega}_S^*, \boldsymbol{\nu}) + F(\boldsymbol{\omega}_S^*, \boldsymbol{\nu})] - \min_{\boldsymbol{\omega} \in \mathcal{W}} \mathbb{E}[F(\boldsymbol{\omega}, \boldsymbol{\nu}^u) - F(\boldsymbol{\omega}, \boldsymbol{\nu}_S^*) + F(\boldsymbol{\omega}, \boldsymbol{\nu}_S^*)] \\
\leq& \max_{\boldsymbol{\nu} \in \mathcal{V}} \mathbb{E}[F(\boldsymbol{\omega}^u, \boldsymbol{\nu}) - F(\boldsymbol{\omega}_S^*, \boldsymbol{\nu})] + \max_{\boldsymbol{\nu} \in \mathcal{V}} \mathbb{E}[F(\boldsymbol{\omega}_S^*, \boldsymbol{\nu})] \\
& - \min_{\boldsymbol{\omega} \in \mathcal{W}} \mathbb{E}[F(\boldsymbol{\omega}, \boldsymbol{\nu}^u) - F(\boldsymbol{\omega}, \boldsymbol{\nu}_S^*)] - \min_{\boldsymbol{\omega} \in \mathcal{W}} \mathbb{E}[F(\boldsymbol{\omega}, \boldsymbol{\nu}_S^*)] \\
=& \max_{\boldsymbol{\nu} \in \mathcal{V}} \mathbb{E}[F(\boldsymbol{\omega}^u, \boldsymbol{\nu}) - F(\boldsymbol{\omega}_S^*, \boldsymbol{\nu})] + \max_{\boldsymbol{\omega} \in \mathcal{W}} \mathbb{E}[(-F)(\boldsymbol{\omega}, \boldsymbol{\nu}^u) - (-F)(\boldsymbol{\omega}, \boldsymbol{\nu}_S^*)] \\
& + \max_{\boldsymbol{\nu} \in \mathcal{V}} \mathbb{E}[F(\boldsymbol{\omega}_S^*, \boldsymbol{\nu})] - \min_{\boldsymbol{\omega} \in \mathcal{W}} \mathbb{E}[F(\boldsymbol{\omega}, \boldsymbol{\nu}_S^*)] \\
\overset{(i)}{\leq}& \mathbb{E}[L\|\boldsymbol{\omega}^u - \boldsymbol{\omega}_S^*\|] + \mathbb{E}[L\|\boldsymbol{\nu}^u - \boldsymbol{\nu}_S^*\|] + \triangle^w(\boldsymbol{\omega}_S^*, \boldsymbol{\nu}_S^*) \\
\overset{(ii)}{\leq}& \mathbb{E}[L\|\boldsymbol{\omega}^u - \boldsymbol{\omega}_S^*\|] + \mathbb{E}[L\|\boldsymbol{\nu}^u - \boldsymbol{\nu}_S^*\|] + \frac{2\sqrt{2}L^2}{\mu n},
\end{aligned}
\tag{77}
$$

where the inequality $(i)$ holds because the population loss function $F(\boldsymbol{\omega}, \boldsymbol{\nu}) := \mathbb{E}[f(\boldsymbol{\omega}, \boldsymbol{\nu}; z)]$ is $L$-Lipschitz continuous. The inequality $(ii)$ is by Lemma 8.

By recalling the unlearning update step in Algorithm 2, we have

$$
\boldsymbol{\omega}^u = \boldsymbol{\omega}_S^* + \frac{1}{n-m}[\mathrm{D}_{\boldsymbol{\omega}\boldsymbol{\omega}} F_{S\backslash}(\boldsymbol{\omega}_S^*, \boldsymbol{\nu}_S^*)]^{-1} \sum_{z_i \in U} \nabla_{\boldsymbol{\omega}} f(\boldsymbol{\omega}_S^*, \boldsymbol{\nu}_S^*; z_i) + \boldsymbol{\xi}_1,
\tag{78}
$$

where the vector $\boldsymbol{\xi}_1 \in \mathbb{R}^{d_1}$ is drawn independently from $\mathcal{N}(0, \sigma_1^2 \mathbf{I}_{d_1})$. From the relation in eq.(78), we further get

$$
\begin{aligned}
\mathbb{E}[\|\boldsymbol{\omega}^u - \boldsymbol{\omega}_S^*\|] =& \mathbb{E}\left[\left\|\frac{1}{n-m}[\mathrm{D}_{\boldsymbol{\omega\omega}}F_{S\setminus}(\boldsymbol{\omega}_S^*, \boldsymbol{\nu}_S^*)]^{-1} \cdot \sum_{z_i \in U} \nabla_{\boldsymbol{\omega}} f(\boldsymbol{\omega}_S^*, \boldsymbol{\nu}_S^*; z_i) + \boldsymbol{\xi}_1 \right\|\right] \\
\overset{(i)}{\leq}& \frac{1}{n-m}\mathbb{E}\left[\left\|[\mathrm{D}_{\boldsymbol{\omega\omega}}F_{S\setminus}(\boldsymbol{\omega}_S^*, \boldsymbol{\nu}_S^*)]^{-1} \cdot \sum_{z_i \in U} \nabla_{\boldsymbol{\omega}} f(\boldsymbol{\omega}_S^*, \boldsymbol{\nu}_S^*; z_i) \right\|\right] + \mathbb{E}[\|\boldsymbol{\xi}_1\|] \\
\overset{(ii)}{\leq}& \frac{1}{n-m} \cdot \frac{n-m}{(\mu n - \ell(1+\ell/\mu)m)}\mathbb{E}\left[\left\|\sum_{z_i \in U} \nabla_{\boldsymbol{\omega}} f(\boldsymbol{\omega}_S^*, \boldsymbol{\nu}_S^*; z_i) \right\|\right] + \sqrt{\mathbb{E}[\|\boldsymbol{\xi}_1\|^2]} \\
\overset{(iii)}{=}& \frac{1}{(\mu n - \ell(1+\ell/\mu)m)}\mathbb{E}\left[\left\|\sum_{z_i \in U} \nabla_{\boldsymbol{\omega}} f(\boldsymbol{\omega}_S^*, \boldsymbol{\nu}_S^*; z_i) \right\|\right] + \sqrt{d_1}\sigma_1 \\
\overset{(iv)}{\leq}& \frac{mL}{(\mu n - \ell(1+\ell/\mu)m)} + \sqrt{d_1}\sigma_1,
\end{aligned}
\tag{79}
$$

where the inequality $(i)$ is by the triangle inequality and the inequality $(ii)$ follows from the relation in eq.(71), together with the Jensen's inequality to bound $\mathbb{E}[\|\boldsymbol{\xi}_1\|]$. The equality $(iii)$ holds because the vector $\boldsymbol{\xi}_1 \sim \mathcal{N}(0, \sigma_1^2 \mathbf{I}_{d_1})$ and thus we have $\mathbb{E}[\|\boldsymbol{\xi}_1\|^2] = d_1\sigma_1^2$. Furthermore, the inequality $(iv)$ is due to the fact that $f(\boldsymbol{\omega}, \boldsymbol{\nu}; z)$ is $L$-Lipshitz continuous.

Symmetrically, we have

$$
\mathbb{E}[\|\boldsymbol{\nu}^u - \boldsymbol{\nu}_S^*\|] \leq \frac{mL}{(\mu n - \ell(1+\ell/\mu)m)} + \sqrt{d_2}\sigma_2.
\tag{80}
$$

Plugging eq.(79) and eq.(80) into eq.(77) with $d = \max\{d_1, d_2\}$ we get

$$
\triangle^w(\boldsymbol{\omega}^u, \boldsymbol{\nu}^u) \leq \frac{2mL^2}{(\mu n - \ell(1+\ell/\mu)m)} + \sqrt{d}(\sigma_1 + \sigma_2)L + \frac{2\sqrt{2}L^2}{\mu n}.
\tag{81}
$$

With the noise scale $\sigma_1$ and $\sigma_2$ being equal to $\frac{2(8\sqrt{2}L^2\ell^3\rho/\mu^5 + 8L\ell^2/\mu^2)m^2}{n(\mu n - (\ell + \ell^2/\mu)m)\epsilon}\sqrt{2\log(2.5/\delta)}$, we can get our generalization guarantee with population weak PD risk:

$$
\triangle^w(\boldsymbol{\omega}^u, \boldsymbol{\nu}^u) = \mathcal{O}\left((L^3\ell^3\rho/\mu^6 + L^2\ell^2/\mu^3) \cdot \frac{m^2\sqrt{d\log(1/\delta)}}{n^2\epsilon} + \frac{mL^2}{\mu n}\right).
\tag{82}
$$

For the population strong PD risk $\triangle^s(\boldsymbol{\omega}^u, \boldsymbol{\nu}^u)$, similarly, we have

$$
\begin{aligned}
& \mathbb{E}[\max_{\boldsymbol{\nu} \in \mathcal{V}} F(\boldsymbol{\omega}^u, \boldsymbol{\nu}) - \min_{\boldsymbol{\omega} \in \mathcal{W}} F(\boldsymbol{\omega}, \boldsymbol{\nu}^u)] \\
=& \mathbb{E}[\max_{\boldsymbol{\nu} \in \mathcal{V}}(F(\boldsymbol{\omega}^u, \boldsymbol{\nu}) - F(\boldsymbol{\omega}_S^*, \boldsymbol{\nu}) + F(\boldsymbol{\omega}_S^*, \boldsymbol{\nu})) - \min_{\boldsymbol{\omega} \in \mathcal{W}}(F(\boldsymbol{\omega}, \boldsymbol{\nu}^u) - F(\boldsymbol{\omega}, \boldsymbol{\nu}_S^*) + F(\boldsymbol{\omega}, \boldsymbol{\nu}_S^*))] \\
=& \mathbb{E}[\max_{\boldsymbol{\nu} \in \mathcal{V}}(F(\boldsymbol{\omega}^u, \boldsymbol{\nu}) - F(\boldsymbol{\omega}_S^*, \boldsymbol{\nu}) + F(\boldsymbol{\omega}_S^*, \boldsymbol{\nu}))] - \mathbb{E}[\min_{\boldsymbol{\omega} \in \mathcal{W}}(F(\boldsymbol{\omega}, \boldsymbol{\nu}^u) - F(\boldsymbol{\omega}, \boldsymbol{\nu}_S^*) + F(\boldsymbol{\omega}, \boldsymbol{\nu}_S^*))] \\
\leq& \mathbb{E}[\max_{\boldsymbol{\nu} \in \mathcal{V}}(F(\boldsymbol{\omega}^u, \boldsymbol{\nu}) - F(\boldsymbol{\omega}_S^*, \boldsymbol{\nu})) + \max_{\boldsymbol{\nu} \in \mathcal{V}} F(\boldsymbol{\omega}_S^*, \boldsymbol{\nu})] \\
& - \mathbb{E}[\min_{\boldsymbol{\omega} \in \mathcal{W}}(F(\boldsymbol{\omega}, \boldsymbol{\nu}^u) - F(\boldsymbol{\omega}, \boldsymbol{\nu}_S^*)) + \min_{\boldsymbol{\omega} \in \mathcal{W}} F(\boldsymbol{\omega}, \boldsymbol{\nu}_S^*)] \\
=& \mathbb{E}[\max_{\boldsymbol{\nu} \in \mathcal{V}}(F(\boldsymbol{\omega}^u, \boldsymbol{\nu}) - F(\boldsymbol{\omega}_S^*, \boldsymbol{\nu}))] + \mathbb{E}[\max_{\boldsymbol{\nu} \in \mathcal{V}} F(\boldsymbol{\omega}_S^*, \boldsymbol{\nu})] \\
& - \mathbb{E}[\min_{\boldsymbol{\omega} \in \mathcal{W}}(F(\boldsymbol{\omega}, \boldsymbol{\nu}^u) - F(\boldsymbol{\omega}, \boldsymbol{\nu}_S^*))] - \mathbb{E}[\min_{\boldsymbol{\omega} \in \mathcal{W}} F(\boldsymbol{\omega}, \boldsymbol{\nu}_S^*)] \\
=& \mathbb{E}[\max_{\boldsymbol{\nu} \in \mathcal{V}}(F(\boldsymbol{\omega}^u, \boldsymbol{\nu}) - F(\boldsymbol{\omega}_S^*, \boldsymbol{\nu}))] + \mathbb{E}[\max_{\boldsymbol{\omega} \in \mathcal{W}}((-F)(\boldsymbol{\omega}, \boldsymbol{\nu}^u) - (-F)(\boldsymbol{\omega}, \boldsymbol{\nu}_S^*))] \\
& + \mathbb{E}[\max_{\boldsymbol{\nu} \in \mathcal{V}} F(\boldsymbol{\omega}_S^*, \boldsymbol{\nu}) - \min_{\boldsymbol{\omega} \in \mathcal{W}} F(\boldsymbol{\omega}, \boldsymbol{\nu}_S^*)] \\
\overset{(i)}{\leq}& \mathbb{E}[L\|\boldsymbol{\omega}^u - \boldsymbol{\omega}_S^*\|] + \mathbb{E}[L\|\boldsymbol{\nu}^u - \boldsymbol{\nu}_S^*\|] + \triangle^s(\boldsymbol{\omega}_S^*, \boldsymbol{\nu}_S^*) \\
\overset{(ii)}{\leq}& \frac{2mL^2}{(\mu n - \ell(1+\ell/\mu)m)} + \sqrt{d}(\sigma_1 + \sigma_2)L + \frac{8L^2\ell}{\mu^2 n},
\end{aligned}
\tag{83}
$$

where inequality $(i)$ is due to the fact that the population loss function $F(\boldsymbol{\omega}, \boldsymbol{\nu}) := \mathbb{E}[f(\boldsymbol{\omega}, \boldsymbol{\nu}; z)]$ is $L$-Lipschitz continuous. The inequality $(ii)$ uses eq.(79), eq.(80) and Lemma 9. With the same noise scale above, we can get the generalization guarantee in terms of strong PD risk below,

$$\triangle^s(\boldsymbol{\omega}^u, \boldsymbol{\nu}^u) = \mathcal{O}\left((L^3\ell^3\rho/\mu^6 + L^2\ell^2/\mu^3) \cdot \frac{m^2\sqrt{d\log(1/\delta)}}{n^2\epsilon} + \frac{mL^2}{\mu n} + \frac{L^2\ell}{\mu^2 n}\right). \tag{84}$$

$\square$

**Theorem 8** (**Deletion Capacity**). *Under the same settings of Lemma 10 and denote $d = \max\{d_1, d_2\}$, the deletion capacity of Algorithm 2 is*

$$m_{\epsilon,\delta,\gamma}^{A,\bar{A}}(d_1, d_2, n) \geq c \cdot \frac{n\sqrt{\epsilon}}{(d\log(1/\delta))^{1/4}}, \tag{85}$$

*where the constant $c$ depends on $L, l, \rho,$ and $\mu$ of the loss function $f$.*

*Proof.* By the definition of deletion capacity, in order to ensure the population PD risk derived in Theorem 7 is bounded by $\gamma$, it suffices to let:

$$m_{\epsilon,\delta,\gamma}^{A,\bar{A}}(d_1, d_2, n) \geq c \cdot \frac{n\sqrt{\epsilon}}{(d\log(1/\delta))^{1/4}},$$

where the constant $c$ depends on the properties of the loss function $f(\boldsymbol{\omega}, \boldsymbol{\nu}; z)$. $\square$

## B.2 Missing Proofs of Sec.4.4

### B.2.1 Proof of Lemma 1 (Closeness Upper Bound)

*Proof.* By the definition of the functions $F_S(\boldsymbol{\omega}, \boldsymbol{\nu})$ and $F_{S\backslash}(\boldsymbol{\omega}, \boldsymbol{\nu})$, we have

$$\|\nabla_{\boldsymbol{\omega}}F_{S\backslash}(\boldsymbol{\omega}_{S\backslash}^*, \mathtt{V}_S(\boldsymbol{\omega}_{S\backslash}^*)) - \nabla_{\boldsymbol{\omega}}F_{S\backslash}(\boldsymbol{\omega}_S^*, \boldsymbol{\nu}_S^*) - \frac{n}{n-m}\mathtt{D}_{\boldsymbol{\omega\omega}}F_S(\boldsymbol{\omega}_S^*, \boldsymbol{\nu}_S^*)(\boldsymbol{\omega}_{S\backslash}^* - \boldsymbol{\omega}_S^*))\|$$

$$=\|\frac{n}{n-m}[\nabla_{\boldsymbol{\omega}}F_S(\boldsymbol{\omega}_{S\backslash}^*, \mathtt{V}_S(\boldsymbol{\omega}_{S\backslash}^*)) - \nabla_{\boldsymbol{\omega}}F_S(\boldsymbol{\omega}_S^*, \boldsymbol{\nu}_S^*) - \mathtt{D}_{\boldsymbol{\omega\omega}}F_S(\boldsymbol{\omega}_S^*, \boldsymbol{\nu}_S^*)(\boldsymbol{\omega}_{S\backslash}^* - \boldsymbol{\omega}_S^*)]$$

$$-\frac{1}{n-m}\sum_{z_i \in U}[\nabla_{\boldsymbol{\omega}}f(\boldsymbol{\omega}_{S\backslash}^*, \mathtt{V}_S(\boldsymbol{\omega}_{S\backslash}^*); z_i) - \nabla_{\boldsymbol{\omega}}f(\boldsymbol{\omega}_S^*, \boldsymbol{\nu}_S^*; z_i)]\|$$

$$\overset{(i)}{\leq}\frac{n}{n-m}\|\nabla_{\boldsymbol{\omega}}F_S(\boldsymbol{\omega}_{S\backslash}^*, \mathtt{V}_S(\boldsymbol{\omega}_{S\backslash}^*)) - \nabla_{\boldsymbol{\omega}}F_S(\boldsymbol{\omega}_S^*, \boldsymbol{\nu}_S^*) - \mathtt{D}_{\boldsymbol{\omega\omega}}F_S(\boldsymbol{\omega}_S^*, \boldsymbol{\nu}_S^*)(\boldsymbol{\omega}_{S\backslash}^* - \boldsymbol{\omega}_S^*)\| \tag{86}$$

$$+\frac{1}{n-m}\sum_{z_i \in U}\|\nabla_{\boldsymbol{\omega}}f(\boldsymbol{\omega}_{S\backslash}^*, \mathtt{V}_S(\boldsymbol{\omega}_{S\backslash}^*); z_i) - \nabla_{\boldsymbol{\omega}}f(\boldsymbol{\omega}_S^*, \boldsymbol{\nu}_S^*; z_i)\|$$

$$\overset{(ii)}{\leq}\frac{8\sqrt{2}\rho L^2\ell^3 m^2}{\mu^5 n(n-m)} + \frac{2\sqrt{2}Ll^2 m^2}{\mu^2 n(n-m)},$$

where the inequality $(i)$ holds because the triangle inequality and the inequality $(ii)$ uses the results in eq.(65) and eq.(66). Now let $\widetilde{\mathbf{x}}$ be the vector satisfying the following relation,

$$\boldsymbol{\omega}_{S\backslash}^* = \boldsymbol{\omega}_S^* + \frac{1}{n}[\mathtt{D}_{\boldsymbol{\omega\omega}}F_S(\boldsymbol{\omega}_S^*, \boldsymbol{\nu}_S^*)]^{-1}\sum_{z_i \in U}\nabla_{\boldsymbol{\omega}}f(\boldsymbol{\omega}_S^*, \boldsymbol{\nu}_S^*; z_i) + \widetilde{\mathbf{x}}. \tag{87}$$

Since we have $\nabla_{\boldsymbol{\omega}}F_{S\backslash}(\boldsymbol{\omega}_S^*, \boldsymbol{\nu}_S^*) = -\frac{1}{n-m}\sum_{z_i \in U}\nabla_{\boldsymbol{\omega}}f(\boldsymbol{\omega}_S^*, \boldsymbol{\nu}_S^*; z_i)$ and $\nabla_{\boldsymbol{\omega}}F_{S\backslash}(\boldsymbol{\omega}_{S\backslash}^*, \mathtt{V}_S(\boldsymbol{\omega}_{S\backslash}^*)) = 0$ due to the optimality of $\boldsymbol{\omega}_{S\backslash}^*$, plugging the above relation into eq.(86), we get

$$\|\frac{n}{n-m}\mathtt{D}_{\boldsymbol{\omega\omega}}F_S(\boldsymbol{\omega}_S^*, \boldsymbol{\nu}_S^*)\widetilde{\mathbf{x}}\| \leq (8\sqrt{2}L^2\ell^3\rho/\mu^5 + 2\sqrt{2}Ll^2/\mu^2)\frac{m^2}{n(n-m)}. \tag{88}$$

With $\mathtt{D}_{\boldsymbol{\omega\omega}}F_S(\boldsymbol{\omega}_S^*, \boldsymbol{\nu}_S^*) \geq \mu_{\boldsymbol{\omega\omega}}$, we also have

$$\|\frac{n}{n-m}\mathtt{D}_{\boldsymbol{\omega\omega}}F_S(\boldsymbol{\omega}_S^*, \boldsymbol{\nu}_S^*)\widetilde{\mathbf{x}}\| \geq \frac{\mu_{\boldsymbol{\omega\omega}}n}{n-m}\|\widetilde{\mathbf{x}}\| \geq \frac{\mu n}{n-m}\|\widetilde{\mathbf{x}}\|. \tag{89}$$

Combining eq.(89), eq.(88), and the definition of the vector $\|\widetilde{\mathbf{x}}\|$, we get that

$$\|\boldsymbol{\omega}_{S\backslash}^* - \widetilde{\boldsymbol{\omega}}\| = \|\widetilde{\mathbf{x}}\| \leq \frac{(8\sqrt{2}L^2\ell^3\rho/\mu^6 + 2\sqrt{2}Ll^2/\mu^3)m^2}{n^2}. \tag{90}$$

Symmetrically, we can get $\|\boldsymbol{\nu}_{S\backslash}^* - \widetilde{\boldsymbol{\nu}}\| \leq \frac{(8\sqrt{2}L^2\ell^3\rho/\mu^6 + 2\sqrt{2}Ll^2/\mu^3)m^2}{n^2}$. $\square$

### B.2.2 Proof of Theorem 2 ($(\epsilon, \delta)$-Minimax Unlearning Certification)

*Proof.* With the closeness upper bound in Lemma 1 and the given noise scales in eq.(22), the proof is identical to that of Theorem 6. $\square$

### B.2.3 Proof of Theorem 3 (Population Primal-Dual Risk)

*Proof.* We start with the population weak PD risk. By eq.(77), we have

$$\triangle^w(\widetilde{\boldsymbol{\omega}}^u, \widetilde{\boldsymbol{\nu}}^u) \leq \mathbb{E}[L\|\widetilde{\boldsymbol{\omega}}^u - \boldsymbol{\omega}_S^*\|] + \mathbb{E}[L\|\widetilde{\boldsymbol{\nu}}^u - \boldsymbol{\nu}_S^*\|] + \frac{2\sqrt{2}L^2}{\mu n}. \tag{91}$$

By recalling the unlearning step in Algorithm 3, we have

$$\widetilde{\boldsymbol{\omega}}^u = \boldsymbol{\omega}_S^* + \frac{1}{n}[\mathsf{D}_{\boldsymbol{\omega}\boldsymbol{\omega}}F_S(\boldsymbol{\omega}_S^*, \boldsymbol{\nu}_S^*)]^{-1} \sum_{z_i \in U} \nabla_{\boldsymbol{\omega}} f(\boldsymbol{\omega}_S^*, \boldsymbol{\nu}_S^*; z_i) + \boldsymbol{\xi}_1, \tag{92}$$

where the vector $\boldsymbol{\xi}_1 \in \mathbb{R}^{d_1}$ is drawn independently from $\mathcal{N}(0, \sigma_1^2 \mathbf{I}_{d_1})$. From the relation in eq.(78), we further get

$$
\begin{aligned}
\mathbb{E}[\|\widetilde{\boldsymbol{\omega}}^u - \boldsymbol{\omega}_S^*\|] &= \mathbb{E}\left[\left\|\frac{1}{n}[\mathsf{D}_{\boldsymbol{\omega}\boldsymbol{\omega}}F_S(\boldsymbol{\omega}_S^*, \boldsymbol{\nu}_S^*)]^{-1} \cdot \sum_{z_i \in U} \nabla_{\boldsymbol{\omega}} f(\boldsymbol{\omega}_S^*, \boldsymbol{\nu}_S^*; z_i) + \boldsymbol{\xi}_1\right\|\right] \\
&\overset{(i)}{\leq} \frac{1}{n}\mathbb{E}\left[\left\|[\mathsf{D}_{\boldsymbol{\omega}\boldsymbol{\omega}}F_S(\boldsymbol{\omega}_S^*, \boldsymbol{\nu}_S^*)]^{-1} \cdot \sum_{z_i \in U} \nabla_{\boldsymbol{\omega}} f(\boldsymbol{\omega}_S^*, \boldsymbol{\nu}_S^*; z_i)\right\|\right] + \mathbb{E}[\|\boldsymbol{\xi}_1\|] \\
&\overset{(ii)}{\leq} \frac{1}{n} \cdot \mu^{-1}\mathbb{E}\left[\left\|\sum_{z_i \in U} \nabla_{\boldsymbol{\omega}} f(\boldsymbol{\omega}_S^*, \boldsymbol{\nu}_S^*; z_i)\right\|\right] + \sqrt{\mathbb{E}[\|\boldsymbol{\xi}_1\|^2]} \\
&\overset{(iii)}{=} \frac{1}{\mu n}\mathbb{E}\left[\left\|\sum_{z_i \in U} \nabla_{\boldsymbol{\omega}} f(\boldsymbol{\omega}_S^*, \boldsymbol{\nu}_S^*; z_i)\right\|\right] + \sqrt{d_1}\sigma_1 \overset{(iv)}{\leq} \frac{mL}{\mu n} + \sqrt{d_1}\sigma_1,
\end{aligned}
\tag{93}
$$

where the inequality $(i)$ uses the triangle inequality and the inequality $(ii)$ follows by the relation $\mathsf{D}_{\boldsymbol{\omega}\boldsymbol{\omega}}F_S(\boldsymbol{\omega}_S^*, \boldsymbol{\nu}_S^*) \geq \mu_{\boldsymbol{\omega}\boldsymbol{\omega}} \geq \mu$, together with the Jensen's inequality to bound $\mathbb{E}[\|\boldsymbol{\xi}_1\|]$. The equality $(iii)$ holds because the vector $\boldsymbol{\xi}_1 \sim \mathcal{N}(0, \sigma_1^2 \mathbf{I}_{d_1})$ and thus we have $\mathbb{E}[\|\boldsymbol{\xi}_1\|^2] = d_1\sigma_1^2$. And the inequality $(iv)$ is due to the fact that $f(\boldsymbol{\omega}, \boldsymbol{\nu}; z)$ is $L$-Lipschitz continuous. Symmetrically, we have

$$\mathbb{E}[\|\widetilde{\boldsymbol{\nu}}^u - \boldsymbol{\nu}_S^*\|] \leq \frac{mL}{\mu n} + \sqrt{d_2}\sigma_2. \tag{94}$$

Plugging eq.(93) and eq.(94) into eq.(91) with $d = \max\{d_1, d_2\}$ we get

$$\triangle^w(\widetilde{\boldsymbol{\omega}}^u, \widetilde{\boldsymbol{\nu}}^u) \leq \frac{2mL^2}{\mu n} + \sqrt{d}(\sigma_1 + \sigma_2)L + \frac{2\sqrt{2}L^2}{\mu n}. \tag{95}$$

With the noise scale $\sigma_1$ and $\sigma_2$ being equal to $\frac{2(8\sqrt{2}L^2\ell^3\rho/\mu^6 + 2\sqrt{2}L\ell^2/\mu^3)m^2}{n^2\epsilon}\sqrt{2\log(2.5/\delta)}$, we can get our generalization guarantee in terms of population weak PD risk:

$$\triangle^w(\widetilde{\boldsymbol{\omega}}^u, \widetilde{\boldsymbol{\nu}}^u) = \mathcal{O}\left((L^3\ell^3\rho/\mu^6 + L^2\ell^2/\mu^3) \cdot \frac{m^2\sqrt{d\log(1/\delta)}}{n^2\epsilon} + \frac{mL^2}{\mu n}\right). \tag{96}$$

For the population strong PD risk, using an application of eq.(83) with Lemma 9, eq.(93), eq.(94) and the noise scales given in Theorem 2, we can get

$$\triangle^s(\widetilde{\boldsymbol{\omega}}^u, \widetilde{\boldsymbol{\nu}}^u) = \mathcal{O}\left((L^3\ell^3\rho/\mu^6 + L^2\ell^2/\mu^3) \cdot \frac{m^2\sqrt{d\log(1/\delta)}}{n^2\epsilon} + \frac{mL^2}{\mu n} + \frac{L^2\ell}{\mu^2 n}\right). \tag{97}$$

$\square$

### B.2.4 Proof of Theorem 4 (Deletion Capacity)

*Proof.* By the definition of deletion capacity, in order to ensure the population weak or strong PD risk derived in Lemma 3 is bounded by $\gamma$, it suffices to let $m_{\epsilon, \delta, \gamma}^{A, \bar{A}}(d_1, d_2, n) \geq c \cdot \frac{n\sqrt{\epsilon}}{(d\log(1/\delta))^{1/4}}$. $\square$

## B.3 Efficient Online Unlearning Algorithm

To support successive unlearning requests, similar to the STL case Suriyakumar and Wilson [2022], we further provide an efficient and online minimax unlearning algorithm (denoted by $\bar{A}_{\texttt{online}}$). The pseudocode of $\bar{A}_{\texttt{online}}$ is given in Algorithm 4. Its certified minimax unlearning guarantee, generalization, and deletion capacity can be identically yielded as Algorithm 3, which are omitted here.

---

**Algorithm 4** Efficient Online Certified Minimax Unlearning ($\bar{A}_{\texttt{online}}$)

---

**Input:** Delete request $z_k \in S$, early delete requests $U : \{z_j\}_{j=1}^m \subseteq S$, output of $A_{sc-sc}(S)$: $(\boldsymbol{\omega}_S^*, \boldsymbol{\nu}_S^*)$, memory variables $T(S)$: $\{\mathrm{D}_{\boldsymbol{\omega}\boldsymbol{\omega}} F_S(\boldsymbol{\omega}_S^*, \boldsymbol{\nu}_S^*), \mathrm{D}_{\boldsymbol{\nu}\boldsymbol{\nu}} F_S(\boldsymbol{\omega}_S^*, \boldsymbol{\nu}_S^*)\}$, early unlearning variables: $(\widetilde{\boldsymbol{\omega}}_U^u, \widetilde{\boldsymbol{\nu}}_U^u)$, loss function: $f(\boldsymbol{\omega}, \boldsymbol{\nu}; z)$, noise parameters: $\sigma_1, \sigma_2$.
1: Set $\widetilde{\boldsymbol{\omega}}_\emptyset = \boldsymbol{\omega}_S^*$ and $\widetilde{\boldsymbol{\nu}}_\emptyset = \boldsymbol{\nu}_S^*$.
2: Compute

$$\overline{\boldsymbol{\omega}}_{U \cup \{k\}} = \widetilde{\boldsymbol{\omega}}_U + \frac{1}{n} [\mathrm{D}_{\boldsymbol{\omega}\boldsymbol{\omega}} F_S(\boldsymbol{\omega}_S^*, \boldsymbol{\nu}_S^*)]^{-1} \nabla_{\boldsymbol{\omega}} f(\boldsymbol{\omega}_S^*, \boldsymbol{\nu}_S^*; z_k), \tag{98}$$

$$\overline{\boldsymbol{\nu}}_{U \cup \{k\}} = \widetilde{\boldsymbol{\nu}}_U + \frac{1}{n} [\mathrm{D}_{\boldsymbol{\nu}\boldsymbol{\nu}} F_S(\boldsymbol{\omega}_S^*, \boldsymbol{\nu}_S^*)]^{-1} \nabla_{\boldsymbol{\nu}} f(\boldsymbol{\omega}_S^*, \boldsymbol{\nu}_S^*; z_k). \tag{99}$$

3: $\widetilde{\boldsymbol{\omega}}_{U \cup \{k\}}^u = \overline{\boldsymbol{\omega}}_{U \cup \{k\}} + \boldsymbol{\xi}_1$, where $\boldsymbol{\xi}_1 \sim \mathcal{N}(0, \sigma_1 \mathbf{I}_{d_1})$ and $\widetilde{\boldsymbol{\nu}}_{U \cup \{k\}}^u = \overline{\boldsymbol{\nu}}_{U \cup \{k\}} + \boldsymbol{\xi}_2$, where $\boldsymbol{\xi}_2 \sim \mathcal{N}(0, \sigma_2 \mathbf{I}_{d_2})$.

**Output:** $(\widetilde{\boldsymbol{\omega}}_{U \cup \{k\}}^u, \widetilde{\boldsymbol{\nu}}_{U \cup \{k\}}^u)$.

---

# C Detailed Algorithm Descriptions and Missing Proofs in Section 5

## C.1 Minimax Unlearning Algorithm for Smooth Convex-Concave Loss Function

In this section, we provide minimax learning and minimax unlearning algorithms for smooth convex-concave loss functions based on the counterpart algorithms for the SC-SC setting. Given the convex-concave loss function $f(\boldsymbol{\omega}, \boldsymbol{\nu}; z)$, we define the regularized loss function as $\widetilde{f}(\boldsymbol{\omega}, \boldsymbol{\nu}; z) = f(\boldsymbol{\omega}, \boldsymbol{\nu}; z) + \frac{\lambda}{2}\|\boldsymbol{\omega}\|^2 - \frac{\lambda}{2}\|\boldsymbol{\nu}\|^2$. Suppose the function $f$ satisfies Assumption 1, then the function $\widetilde{f}$ is $\lambda$-strongly convex in $\boldsymbol{\omega}$, $\lambda$-strongly concave in $\boldsymbol{\nu}$, $(2L + \lambda\|\boldsymbol{\omega}\| + \lambda\|\boldsymbol{\nu}\|)$-Lipschitz, $\sqrt{2}(2\ell + \lambda)$-gradient Lipschitz and $\rho$-Hessian Lipschitz. Thus, we can apply the minimax learning in Algorithm 1 and unlearning in Algorithm 2 to the regularized loss function with a properly chosen $\lambda$. We denote our learning algorithm by $A_{c-c}$ and unlearning algorithm by $\bar{A}_{c-c}$. The pseudocode is provided in Algorithm 5 and Algorithm 6, respectively. Additionally, we denote the regularized population loss as $\widetilde{F}(\boldsymbol{\omega}, \boldsymbol{\nu}) := \mathbb{E}_{z \sim \mathcal{D}}[\widetilde{f}(\boldsymbol{\omega}, \boldsymbol{\nu}; z)]$ and regularized empirical loss as $\widetilde{F}_S(\boldsymbol{\omega}, \boldsymbol{\nu}) := \frac{1}{n}\sum_{i=1}^n \widetilde{f}(\boldsymbol{\omega}, \boldsymbol{\nu}; z_i)$.

---

**Algorithm 5** Mimimax Learning Algorithm ($A_{c-c}$)

---

**Input:** Dataset $S : \{z_i\}_{i=1}^n \sim \mathcal{D}^n$, loss function: $f(\boldsymbol{\omega}, \boldsymbol{\nu}; z)$, regularization parameter: $\lambda$.
1: Define
$$\widetilde{f}(\boldsymbol{\omega}, \boldsymbol{\nu}; z) = f(\boldsymbol{\omega}, \boldsymbol{\nu}; z) + \frac{\lambda}{2}\|\boldsymbol{\omega}\|^2 - \frac{\lambda}{2}\|\boldsymbol{\nu}\|^2. \tag{100}$$

2: Run the algorithm $A_{sc-sc}$ on the dataset $S$ with loss function $\widetilde{f}$.
**Output:** $(\boldsymbol{\omega}_S^*, \boldsymbol{\nu}_S^*, \mathtt{D}_{\boldsymbol{\omega}\boldsymbol{\omega}}\widetilde{F}_S(\boldsymbol{\omega}_S^*, \boldsymbol{\nu}_S^*), \mathtt{D}_{\boldsymbol{\nu}\boldsymbol{\nu}}\widetilde{F}_S(\boldsymbol{\omega}_S^*, \boldsymbol{\nu}_S^*)) \leftarrow A_{sc-sc}(S, \widetilde{f})$.

---

---

**Algorithm 6** Certified Minimax Unlearning for Convex-Concave Loss ($\bar{A}_{c-c}$)

---

**Input:** Delete requests $U : \{z_j\}_{j=1}^m \subseteq S$, output of $A_{c-c}(S)$: $(\boldsymbol{\omega}_S^*, \boldsymbol{\nu}_S^*)$, memory variables $T(S)$: $\{\mathtt{D}_{\boldsymbol{\omega}\boldsymbol{\omega}}\widetilde{F}_S(\boldsymbol{\omega}_S^*, \boldsymbol{\nu}_S^*), \mathtt{D}_{\boldsymbol{\nu}\boldsymbol{\nu}}\widetilde{F}_S(\boldsymbol{\omega}_S^*, \boldsymbol{\nu}_S^*)\}$, loss function: $f(\boldsymbol{\omega}, \boldsymbol{\nu}; z)$, regularization parameter: $\lambda$, noise parameters: $\sigma_1, \sigma_2$.
1: Define
$$\widetilde{f}(\boldsymbol{\omega}, \boldsymbol{\nu}; z) = f(\boldsymbol{\omega}, \boldsymbol{\nu}; z) + \frac{\lambda}{2}\|\boldsymbol{\omega}\|^2 - \frac{\lambda}{2}\|\boldsymbol{\nu}\|^2. \tag{101}$$

2: Run the algorithm $\bar{A}_{sc-sc}$ with delete requests $U$, learning variables $(\boldsymbol{\omega}_S^*, \boldsymbol{\nu}_S^*)$, memory variables $T(S)$, loss function $\widetilde{f}$ and noise parameters $\sigma_1$ and $\sigma_2$.
**Output:** $(\boldsymbol{\omega}^u, \boldsymbol{\nu}^u) \leftarrow \bar{A}_{sc-sc}(U, (\boldsymbol{\omega}_S^*, \boldsymbol{\nu}_S^*), T(S), \widetilde{f}, \sigma_1, \sigma_2)$.

---

## C.2 Supporting Lemma

**Lemma 11.** *Suppose the function $f(\boldsymbol{\omega}, \boldsymbol{\nu}; z)$ is $L$-Lipschitz continuous. Define the function $\widetilde{f}(\boldsymbol{\omega}, \boldsymbol{\nu}; z)$ as*

$$\widetilde{f}(\boldsymbol{\omega}, \boldsymbol{\nu}; z) = f(\boldsymbol{\omega}, \boldsymbol{\nu}; z) + \frac{\lambda}{2}\|\boldsymbol{\omega}\|^2 - \frac{\lambda}{2}\|\boldsymbol{\nu}\|^2. \tag{102}$$

*Given a dataset $S = \{z_i\}_{i=1}^n$ and denote $(\boldsymbol{\omega}_S^*, \boldsymbol{\nu}_S^*) := \arg\min_{\boldsymbol{\omega} \in \mathcal{W}} \max_{\boldsymbol{\nu} \in \mathcal{V}}\{\widetilde{F}_S(\boldsymbol{\omega}, \boldsymbol{\nu}) := \frac{1}{n}\sum_{i=1}^n \widetilde{f}(\boldsymbol{\omega}, \boldsymbol{\nu}; z_i)\}$. Then, the variables $(\boldsymbol{\omega}_S^*, \boldsymbol{\nu}_S^*)$ satisfy $\|\boldsymbol{\omega}_S^*\| \leq L/\lambda$ and $\|\boldsymbol{\nu}_S^*\| \leq L/\lambda$.*

*Proof.* Due to the optimality of $\boldsymbol{\omega}_S^*$, we have

$$\nabla_{\boldsymbol{\omega}}\widetilde{F}_S(\boldsymbol{\omega}_S^*, \boldsymbol{\nu}_S^*; z) = \frac{1}{n}\sum_{z_i \in S}\nabla_{\boldsymbol{\omega}}\widetilde{f}(\boldsymbol{\omega}_S^*, \boldsymbol{\nu}_S^*; z_i) = 0. \tag{103}$$

Plugging in the definition of the function $\widetilde{f}$ in the above, we get that

$$\frac{1}{n}\sum_{z_i \in S}\nabla_{\boldsymbol{\omega}}f(\boldsymbol{\omega}_S^*, \boldsymbol{\nu}_S^*; z_i) + \lambda\boldsymbol{\omega}_S^* = 0. \tag{104}$$

Then, using the triangle inequality, we have

$$\|\lambda\boldsymbol{\omega}_S^*\| = \|\frac{1}{n}\sum_{z_i \in S}\nabla_{\boldsymbol{\omega}}f(\boldsymbol{\omega}_S^*, \boldsymbol{\nu}_S^*; z_i)\| \le \frac{1}{n}\sum_{z_i \in S}\|\nabla_{\boldsymbol{\omega}}f(\boldsymbol{\omega}_S^*, \boldsymbol{\nu}_S^*; z_i)\| \le L, \tag{105}$$

where the last inequality holds because the function $f$ is $L$-Lipschitz continuous. Thus we have $\|\boldsymbol{\omega}_S^*\| \le L/\lambda$. Similarly, we can get $\|\boldsymbol{\nu}_S^*\| \le L/\lambda$. □

Lemma 11 implies that the empirical optimizer $(\boldsymbol{\omega}_S^*, \boldsymbol{\nu}_S^*)$ returned by Algorithm 6 satisfies $\|\boldsymbol{\omega}_S^*\| \le L/\lambda$ and $\|\boldsymbol{\nu}_S^*\| \le L/\lambda$. Thus our domain of interest are $\mathcal{W} := \{\boldsymbol{\omega}|\|\boldsymbol{\omega}\| \le L/\lambda\}$ and $\mathcal{V} := \{\boldsymbol{\nu}|\|\boldsymbol{\nu}\| \le L/\lambda\}$. Over the set $\mathcal{W} \times \mathcal{V}$, the function $\widetilde{f}(\boldsymbol{\omega}, \boldsymbol{\nu}; z)$ is $4L$-Lipschitz continuous. Also, with $\lambda < \ell$, $\widetilde{f}(\boldsymbol{\omega}, \boldsymbol{\nu}; z)$ has $3\sqrt{2}\ell$-Lipschitz gradients.

## C.3 Proof of Theorem 5 (Certified Minimax Unlearning for Convex-Concave Loss Funcion)

Denote $\widetilde{L} = 4L$ and $\widetilde{\ell} = 3\sqrt{2}\ell$, then the function $\widetilde{f}$ is $\widetilde{L}$-Lipschitz continuous and has $\widetilde{\ell}$-Lipschitz gradients. Let $(\boldsymbol{\omega}_{S\backslash}^*, \boldsymbol{\nu}_{S\backslash}^*)$ be the optimal solution of the loss function $\widetilde{F}_{S\backslash}(\boldsymbol{\omega}, \boldsymbol{\nu})$ on the remaining dataset, i.e.,

$$(\boldsymbol{\omega}_{S\backslash}^*, \boldsymbol{\nu}_{S\backslash}^*) := \arg\min_{\boldsymbol{\omega} \in \mathcal{W}}\max_{\boldsymbol{\nu} \in \mathcal{V}}\{\widetilde{F}_{S\backslash}(\boldsymbol{\omega}, \boldsymbol{\nu}) := \frac{1}{n-m}\sum_{z_i \in S\backslash U}\widetilde{f}(\boldsymbol{\omega}, \boldsymbol{\nu}; z_i)\}. \tag{106}$$

Additionally, we have $\|\mathsf{D}_{\boldsymbol{\omega}\boldsymbol{\omega}}\widetilde{F}_S(\boldsymbol{\omega}_S^*, \boldsymbol{\nu}_S^*)\| \ge \lambda$ and $\|\mathsf{D}_{\boldsymbol{\nu}\boldsymbol{\nu}}\widetilde{F}_S(\boldsymbol{\omega}_S^*, \boldsymbol{\nu}_S^*)\| \ge \lambda$.

**Lemma 12.** *Under the settings of Theorem 5, for any $\lambda > 0$, the population weak and strong PD risk for the minimax learning variables $(\boldsymbol{\omega}_S^*, \boldsymbol{\nu}_S^*)$ returned by Algorithm 5 are*

$$\begin{cases} \max_{\boldsymbol{\nu} \in \mathcal{V}}\mathbb{E}[F(\boldsymbol{\omega}_S^*, \boldsymbol{\nu})] - \min_{\boldsymbol{\omega} \in \mathcal{W}}\mathbb{E}[F(\boldsymbol{\omega}, \boldsymbol{\nu}_S^*)] \le \dfrac{32\sqrt{2}L^2}{\lambda n} + \dfrac{\lambda}{2}(B_{\boldsymbol{\omega}}^2 + B_{\boldsymbol{\nu}}^2), \\[3mm] \mathbb{E}[\max_{\boldsymbol{\nu} \in \mathcal{V}}F(\boldsymbol{\omega}_S^*, \boldsymbol{\nu})] - \min_{\boldsymbol{\omega} \in \mathcal{W}}F(\boldsymbol{\omega}, \boldsymbol{\nu}_S^*)] \le \dfrac{384\sqrt{2}L^2\ell}{\lambda^2 n} + \dfrac{\lambda}{2}(B_{\boldsymbol{\omega}}^2 + B_{\boldsymbol{\nu}}^2). \end{cases} \tag{107}$$

*Proof.* For the function $\widetilde{F}(\boldsymbol{\omega}, \boldsymbol{\nu})$, an application of Lemma 8 gives that

$$\max_{\boldsymbol{\nu} \in \mathcal{V}}\mathbb{E}[\widetilde{F}(\boldsymbol{\omega}_S^*, \boldsymbol{\nu})] - \min_{\boldsymbol{\omega} \in \mathcal{W}}\mathbb{E}[\widetilde{F}(\boldsymbol{\omega}, \boldsymbol{\nu}_S^*)] \le \frac{32\sqrt{2}L^2}{\lambda n}. \tag{108}$$

By the assumption of bounded parameter spaces $\mathcal{W}$ and $\mathcal{V}$ so that $\max_{\boldsymbol{\omega} \in \mathcal{W}}\|\boldsymbol{\omega}\| \le B_{\boldsymbol{\omega}}$ and $\max_{\boldsymbol{\nu} \in \mathcal{V}}\|\boldsymbol{\nu}\| \le B_{\boldsymbol{\nu}}$, we have the following derivations for the population weak PD risk,

$$\begin{aligned} &\max_{\boldsymbol{\nu} \in \mathcal{V}}\mathbb{E}[F(\boldsymbol{\omega}_S^*, \boldsymbol{\nu})] - \min_{\boldsymbol{\omega} \in \mathcal{W}}\mathbb{E}[F(\boldsymbol{\omega}, \boldsymbol{\nu}_S^*)] \\ =&\max_{\boldsymbol{\nu} \in \mathcal{V}}\mathbb{E}\left[\widetilde{F}(\boldsymbol{\omega}_S^*, \boldsymbol{\nu}) - \frac{\lambda}{2}\|\boldsymbol{\omega}_S^*\|^2 + \frac{\lambda}{2}\|\boldsymbol{\nu}\|^2\right] - \min_{\boldsymbol{\omega} \in \mathcal{W}}\mathbb{E}\left[\widetilde{F}(\boldsymbol{\omega}, \boldsymbol{\nu}_S^*) - \frac{\lambda}{2}\|\boldsymbol{\omega}\|^2 + \frac{\lambda}{2}\|\boldsymbol{\nu}_S^*\|^2\right] \\ \le&\max_{\boldsymbol{\nu} \in \mathcal{V}}\mathbb{E}\left[\widetilde{F}(\boldsymbol{\omega}_S^*, \boldsymbol{\nu}) + \frac{\lambda}{2}\|\boldsymbol{\nu}\|^2\right] - \min_{\boldsymbol{\omega} \in \mathcal{W}}\mathbb{E}\left[\widetilde{F}(\boldsymbol{\omega}, \boldsymbol{\nu}_S^*) - \frac{\lambda}{2}\|\boldsymbol{\omega}\|^2\right] \\ \le&\max_{\boldsymbol{\nu} \in \mathcal{V}}\mathbb{E}\left[\widetilde{F}(\boldsymbol{\omega}_S^*, \boldsymbol{\nu})\right] - \min_{\boldsymbol{\omega} \in \mathcal{W}}\mathbb{E}\left[\widetilde{F}(\boldsymbol{\omega}, \boldsymbol{\nu}_S^*)\right] + \max_{\boldsymbol{\nu} \in \mathcal{V}}\mathbb{E}\left[\frac{\lambda}{2}\|\boldsymbol{\nu}\|^2\right] + \max_{\boldsymbol{\omega} \in \mathcal{W}}\mathbb{E}\left[\frac{\lambda}{2}\|\boldsymbol{\omega}\|^2\right] \\ \le&\frac{32\sqrt{2}L^2}{\lambda n} + \frac{\lambda}{2}(B_{\boldsymbol{\omega}}^2 + B_{\boldsymbol{\nu}}^2). \end{aligned} \tag{109}$$

Similarly, an application of Lemma 9 gives that

$$\mathbb{E}[\max_{\boldsymbol{\nu} \in \mathcal{V}}\widetilde{F}(\boldsymbol{\omega}_S^*, \boldsymbol{\nu}) - \min_{\boldsymbol{\omega} \in \mathcal{W}}\widetilde{F}(\boldsymbol{\omega}, \boldsymbol{\nu}_S^*)] \le \frac{128\sqrt{2}L^2(2\ell + \lambda)}{\lambda^2 n}. \tag{110}$$

And we can get the population strong PD risk with the following conversions,

$$\mathbb{E}[\max_{\boldsymbol{\nu}\in\mathcal{V}} F(\boldsymbol{\omega}_S^*, \boldsymbol{\nu}) - \min_{\boldsymbol{\omega}\in\mathcal{W}} F(\boldsymbol{\omega}, \boldsymbol{\nu}_S^*)]$$

$$=\mathbb{E}\left[\max_{\boldsymbol{\nu}\in\mathcal{V}} \left(\widetilde{F}(\boldsymbol{\omega}_S^*, \boldsymbol{\nu}) - \frac{\lambda}{2}\|\boldsymbol{\omega}_S^*\|^2 + \frac{\lambda}{2}\|\boldsymbol{\nu}\|^2\right) - \min_{\boldsymbol{\omega}\in\mathcal{W}} \left(\widetilde{F}(\boldsymbol{\omega}, \boldsymbol{\nu}_S^*) - \frac{\lambda}{2}\|\boldsymbol{\omega}\|^2 + \frac{\lambda}{2}\|\boldsymbol{\nu}_S^*\|^2\right)\right]$$

$$\leq\mathbb{E}\left[\max_{\boldsymbol{\nu}\in\mathcal{V}} \left(\widetilde{F}(\boldsymbol{\omega}_S^*, \boldsymbol{\nu}) + \frac{\lambda}{2}\|\boldsymbol{\nu}\|^2\right) - \min_{\boldsymbol{\omega}\in\mathcal{W}} \left(\widetilde{F}(\boldsymbol{\omega}, \boldsymbol{\nu}_S^*) - \frac{\lambda}{2}\|\boldsymbol{\omega}\|^2\right)\right]$$

$$=\mathbb{E}\left[\max_{\boldsymbol{\nu}\in\mathcal{V}} \left(\widetilde{F}(\boldsymbol{\omega}_S^*, \boldsymbol{\nu}) + \frac{\lambda}{2}\|\boldsymbol{\nu}\|^2\right)\right] - \mathbb{E}\left[\min_{\boldsymbol{\omega}\in\mathcal{W}} \left(\widetilde{F}(\boldsymbol{\omega}, \boldsymbol{\nu}_S^*) - \frac{\lambda}{2}\|\boldsymbol{\omega}\|^2\right)\right]$$

$$\leq\mathbb{E}[\max_{\boldsymbol{\nu}\in\mathcal{V}} \widetilde{F}(\boldsymbol{\omega}_S^*, \boldsymbol{\nu}) + \max_{\boldsymbol{\nu}\in\mathcal{V}} \frac{\lambda}{2}\|\boldsymbol{\nu}\|^2] + \mathbb{E}[\max_{\boldsymbol{\omega}\in\mathcal{W}}(-\widetilde{F})(\boldsymbol{\omega}, \boldsymbol{\nu}_S^*) + \max_{\boldsymbol{\omega}\in\mathcal{W}} \frac{\lambda}{2}\|\boldsymbol{\omega}\|^2]$$

$$=\mathbb{E}[\max_{\boldsymbol{\nu}\in\mathcal{V}} \widetilde{F}(\boldsymbol{\omega}_S^*, \boldsymbol{\nu}) - \min_{\boldsymbol{\omega}\in\mathcal{W}} \widetilde{F}(\boldsymbol{\omega}, \boldsymbol{\nu}_S^*)] + \mathbb{E}[\max_{\boldsymbol{\nu}\in\mathcal{V}} \frac{\lambda}{2}\|\boldsymbol{\nu}\|^2] + \mathbb{E}[\max_{\boldsymbol{\omega}\in\mathcal{W}} \frac{\lambda}{2}\|\boldsymbol{\omega}\|^2]$$

$$\leq\frac{128\sqrt{2}L^2(2\ell+\lambda)}{\lambda^2 n} + \frac{\lambda}{2}(B_{\boldsymbol{\omega}}^2 + B_{\boldsymbol{\nu}}^2) \leq \frac{384\sqrt{2}L^2\ell}{\lambda^2 n} + \frac{\lambda}{2}(B_{\boldsymbol{\omega}}^2 + B_{\boldsymbol{\nu}}^2). \tag{111}$$

$\square$

**Lemma 13** (**Closeness Upper Bound**). *Under the settings of Theorem 5, we have the closeness bound between the intermediate variables $(\widehat{\boldsymbol{\omega}}, \widehat{\boldsymbol{\nu}})$ in Algorithm 6 and $(\boldsymbol{\omega}_{S\backslash}^*, \boldsymbol{\nu}_{S\backslash}^*)$ in eq.(106):*

$$\{\|\boldsymbol{\omega}_{S\backslash}^* - \widehat{\boldsymbol{\omega}}\|, \|\boldsymbol{\nu}_{S\backslash}^* - \widehat{\boldsymbol{\nu}}\|\} \leq \frac{(8\sqrt{2}\widetilde{L}^2\widetilde{\ell}^3\rho/\lambda^5 + 8\widetilde{L}\widetilde{\ell}^2/\lambda^2)m^2}{n(\lambda n - (\widetilde{\ell} + \widetilde{\ell}^2/\lambda)m)}. \tag{112}$$

*Proof.* Since we now run the algorithms $A_{sc-sc}$ and $\bar{A}_{sc-sc}$ with the regularized loss function $\widetilde{f}$, the proof is identical to that of Lemma 10. $\square$

Equipped with the supporting lemmas above, the proof of Theorem 5 can be separated into the proofs of the following three lemmas.

**Lemma 14** (**Minimax Unlearning Certification**). *Under the settings of Theorem 5, our minimax learning algorithm $A_{c-c}$ and unlearning algorithm $\bar{A}_{c-c}$ is $(\epsilon, \delta)$-certified minimax unlearning if we choose*

$$\sigma_1 \text{ and } \sigma_2 = \frac{2(8\sqrt{2}\widetilde{L}^2\widetilde{\ell}^3\rho/\lambda^5 + 8\widetilde{L}\widetilde{\ell}^2/\lambda^2)m^2}{n(\lambda n - (\widetilde{\ell} + \widetilde{\ell}^2/\lambda)m)\epsilon} \sqrt{2\log(2.5/\delta)}. \tag{113}$$

*Proof.* With the closeness upper bound in Lemma 13 and the given noise scales in eq.(113), the proof is identical to that of Theorem 6. $\square$

**Lemma 15** (**Population Primal-Dual Risk**). *Under the settings of Theorem 5, the population weak and strong PD risk for $(\boldsymbol{\omega}^u, \boldsymbol{\nu}^u)$ returned by Algorithm 6 are*

$$\begin{cases} \triangle^w(\boldsymbol{\omega}^u, \boldsymbol{\nu}^u) \leq \mathcal{O}\left((L^3\ell^3\rho/\lambda^6 + L^2\ell^2/\lambda^3) \cdot \frac{m^2\sqrt{d\log(1/\delta)}}{n^2\epsilon} + \frac{mL^2}{\lambda n} + \lambda(B_{\boldsymbol{\omega}}^2 + B_{\boldsymbol{\nu}}^2)\right), \\ \triangle^s(\boldsymbol{\omega}^u, \boldsymbol{\nu}^u) \leq \mathcal{O}\left((L^3\ell^3\rho/\lambda^6 + L^2\ell^2/\lambda^3) \cdot \frac{m^2\sqrt{d\log(1/\delta)}}{n^2\epsilon} + \frac{mL^2}{\lambda n} + \frac{L^2\ell}{\lambda^2 n} + \lambda(B_{\boldsymbol{\omega}}^2 + B_{\boldsymbol{\nu}}^2)\right). \end{cases} \tag{114}$$

*Proof.* For the population weak PD risk, an application of eq.(77) together with Lemma 12 gives that

$$\triangle^w(\boldsymbol{\omega}^u, \boldsymbol{\nu}^u) \leq \mathbb{E}[L\|\boldsymbol{\omega}^u - \boldsymbol{\omega}_S^*\|] + \mathbb{E}[L\|\boldsymbol{\nu}^u - \boldsymbol{\nu}_S^*\|] + \frac{32\sqrt{2}L^2}{\lambda n} + \frac{\lambda}{2}(B_{\boldsymbol{\omega}}^2 + B_{\boldsymbol{\nu}}^2). \tag{115}$$

According to Algorithm 6, we have the unlearning update step

$$\boldsymbol{\omega}^u = \boldsymbol{\omega}_S^* + \frac{1}{n-m}[\mathsf{D}_{\boldsymbol{\omega}\boldsymbol{\omega}}\widetilde{F}_{S\backslash}(\boldsymbol{\omega}_S^*, \boldsymbol{\nu}_S^*)]^{-1} \sum_{z_i \in U} \nabla_{\boldsymbol{\omega}}\widetilde{f}(\boldsymbol{\omega}_S^*, \boldsymbol{\nu}_S^*; z_i) + \boldsymbol{\xi}_1, \tag{116}$$

where $\widetilde{F}_{S\backslash}(\boldsymbol{\omega}_S^*, \boldsymbol{\nu}_S^*) := \frac{1}{n-m}\sum_{z_i \in S\backslash U} \widetilde{f}(\boldsymbol{\omega}, \boldsymbol{\nu}; z_i)$. From the relation above, we further get

$$
\begin{aligned}
&\mathbb{E}[\|\boldsymbol{\omega}^u - \boldsymbol{\omega}_S^*\|] \\
&=\mathbb{E}\left[\left\|\frac{1}{n-m}[\mathrm{D}_{\boldsymbol{\omega\omega}}\widetilde{F}_{S\backslash}(\boldsymbol{\omega}_S^*, \boldsymbol{\nu}_S^*)]^{-1}\cdot\sum_{z_i \in U}\nabla_{\boldsymbol{\omega}}\widetilde{f}(\boldsymbol{\omega}_S^*, \boldsymbol{\nu}_S^*; z_i) + \boldsymbol{\xi}_1\right\|\right] \\
&\overset{(i)}{\le}\frac{1}{n-m}\mathbb{E}\left[\left\|[\mathrm{D}_{\boldsymbol{\omega\omega}}\widetilde{F}_{S\backslash}(\boldsymbol{\omega}_S^*, \boldsymbol{\nu}_S^*)]^{-1}\cdot\sum_{z_i \in U}\nabla_{\boldsymbol{\omega}}\widetilde{f}(\boldsymbol{\omega}_S^*, \boldsymbol{\nu}_S^*; z_i)\right\|\right] + \mathbb{E}[\|\boldsymbol{\xi}_1\|] \\
&\overset{(ii)}{\le}\frac{1}{n-m}\cdot\frac{n-m}{(\lambda n - \widetilde{\ell}(1+\widetilde{\ell}/\lambda)m)}\mathbb{E}\left[\left\|\sum_{z_i \in U}\nabla_{\boldsymbol{\omega}}\widetilde{f}(\boldsymbol{\omega}_S^*, \boldsymbol{\nu}_S^*; z_i)\right\|\right] + \sqrt{\mathbb{E}[\|\boldsymbol{\xi}_1\|^2]} \\
&\overset{(iii)}{=}\frac{1}{(\lambda n - \widetilde{\ell}(1+\widetilde{\ell}/\lambda)m)}\mathbb{E}\left[\left\|\sum_{z_i \in U}\nabla_{\boldsymbol{\omega}}\widetilde{f}(\boldsymbol{\omega}_S^*, \boldsymbol{\nu}_S^*; z_i)\right\|\right] + \sqrt{d_1}\sigma_1 \\
&\overset{(iv)}{\le}\frac{1}{(\lambda n - \widetilde{\ell}(1+\widetilde{\ell}/\lambda)m)}\mathbb{E}\left[\sum_{z_i \in U}\left(\|\nabla_{\boldsymbol{\omega}}f(\boldsymbol{\omega}_S^*, \boldsymbol{\nu}_S^*; z_i)\| + \lambda\|\boldsymbol{\omega}_S^*\| + \lambda\|\boldsymbol{\nu}_S^*\|\right)\right] + \sqrt{d_1}\sigma_1 \\
&\overset{(vi)}{\le}\frac{3mL}{(\lambda n - \widetilde{\ell}(1+\widetilde{\ell}/\lambda)m)} + \sqrt{d_1}\sigma_1,
\end{aligned}
\tag{117}
$$

where the inequality $(i)$ uses the triangle inequality and the inequality $(ii)$ follows from an application of eq.(71), together with the Jensen's inequality to bound $\mathbb{E}[\|\boldsymbol{\xi}_1\|]$. The equality $(iii)$ holds because the vector $\boldsymbol{\xi}_1 \sim \mathcal{N}(0, \sigma_1^2\mathbf{I}_{d_1})$ and thus we have $\mathbb{E}[\|\boldsymbol{\xi}_1\|^2] = d_1\sigma_1^2$. The inequality $(iv)$ uses the definition of the function $\widetilde{f}$ and the triangle inequality. The inequality $(vi)$ is due to the fact that $f(\boldsymbol{\omega}, \boldsymbol{\nu}; z)$ is $L$-Lipshitz continuous and Lemma 11. Symmetrically, we have

$$
\mathbb{E}[\|\boldsymbol{\nu}^u - \boldsymbol{\nu}_S^*\|] \le \frac{3mL}{(\lambda n - \widetilde{\ell}(1+\widetilde{\ell}/\lambda)m)} + \sqrt{d_2}\sigma_2.
\tag{118}
$$

Plugging eq.(117) and eq.(118) into eq.(115) with noise scales given in Lemma 14, we can get our generalization guarantee in terms of population weak PD risk:

$$
\triangle^w(\boldsymbol{\omega}^u, \boldsymbol{\nu}^u) \le \mathcal{O}\left((L^3\ell^3\rho/\lambda^6 + L^2\ell^2/\lambda^3)\cdot\frac{m^2\sqrt{d\log(1/\delta)}}{n^2\epsilon} + \frac{mL^2}{\lambda n} + \lambda(B_{\boldsymbol{\omega}}^2 + B_{\boldsymbol{\nu}}^2)\right).
\tag{119}
$$

Similarly, using an application of eq.(83) together with Lemma 12, Lemma 14, eq.(117) and eq.(118), we can get the following population strong PD risk:

$$
\triangle^s(\boldsymbol{\omega}^u, \boldsymbol{\nu}^u) \le \mathcal{O}\left((L^3\ell^3\rho/\lambda^6 + L^2\ell^2/\lambda^3)\cdot\frac{m^2\sqrt{d\log(1/\delta)}}{n^2\epsilon} + \frac{mL^2}{\lambda n} + \frac{L^2\ell}{\lambda^2 n} + \lambda(B_{\boldsymbol{\omega}}^2 + B_{\boldsymbol{\nu}}^2)\right).
\tag{120}
$$

$\square$

**Lemma 16** (**Deletion Capacity**). *Under the settings of Theorem 5, the deletion capacity of Algorithm 6 is*

$$
m_{\epsilon,\delta,\gamma}^{A,\bar{A}}(d_1, d_2, n) \ge c\cdot\frac{n\sqrt{\epsilon}}{(d\log(1/\delta))^{1/4}},
\tag{121}
$$

*where the constant $c$ depends on $L, l, \rho, B_{\boldsymbol{\omega}}$ and $B_{\boldsymbol{\nu}}$.*

*Proof.* By the definition of deletion capacity, in order to ensure the population PD risk derived in Lemma 15 is bounded by $\gamma$, it suffices to let $m_{\epsilon,\delta,\gamma}^{A,\bar{A}}(d_1, d_2, n) \ge c\cdot\frac{n\sqrt{\epsilon}}{(d\log(1/\delta))^{1/4}}$. $\square$

### C.4 Minimax Unlearning Algorithm for Smooth Convex-Strongly-Concave Loss Function

In this section, we briefly discuss the extension to the smooth C-SC setting. The SC-C setting is symmetric and thus omitted here.

Given the loss function $f(\boldsymbol{\omega}, \boldsymbol{\nu}; z)$ that satisfies Assumption 1 with $\mu_{\boldsymbol{\nu}}$-strong concavity in $\boldsymbol{\nu}$, we define the regularized function as $\widetilde{f}(\boldsymbol{\omega}, \boldsymbol{\nu}; z) = f(\boldsymbol{\omega}, \boldsymbol{\nu}; z) + \frac{\lambda}{2}\|\boldsymbol{\omega}\|^2$. Our minimax learning and minimax unlearning algorithms for C-SC loss function $f$ denoted by $A_{c-sc}$ and $\bar{A}_{c-sc}$ are given in Algorithm 7 and Algorithm 8 respectively. Additionally, we denote the regularized population loss by $\widetilde{F}(\boldsymbol{\omega}, \boldsymbol{\nu}) := \mathbb{E}_{z \sim \mathcal{D}}[\widetilde{f}(\boldsymbol{\omega}, \boldsymbol{\nu}; z)]$ and regularized empirical loss by $\widetilde{F}_S(\boldsymbol{\omega}, \boldsymbol{\nu}) := \frac{1}{n}\sum_{i=1}^n \widetilde{f}(\boldsymbol{\omega}, \boldsymbol{\nu}; z_i)$.

---

**Algorithm 7** Mimimax Learning Algorithm ($A_{c-sc}$)

---

**Input:** Dataset $S : \{z_i\}_{i=1}^n \sim \mathcal{D}^n$, loss function: $f(\boldsymbol{\omega}, \boldsymbol{\nu}; z)$, regularization parameter: $\lambda$.
  1: Define

$$\widetilde{f}(\boldsymbol{\omega}, \boldsymbol{\nu}; z) = f(\boldsymbol{\omega}, \boldsymbol{\nu}; z) + \frac{\lambda}{2}\|\boldsymbol{\omega}\|^2. \tag{122}$$

  2: Run the algorithm $A_{sc-sc}$ on the dataset $S$ with loss function $\widetilde{f}$.
**Output:** $(\boldsymbol{\omega}_S^*, \boldsymbol{\nu}_S^*, \mathtt{D}_{\boldsymbol{\omega\omega}}\widetilde{F}_S(\boldsymbol{\omega}_S^*, \boldsymbol{\nu}_S^*), \mathtt{D}_{\boldsymbol{\nu\nu}}\widetilde{F}_S(\boldsymbol{\omega}_S^*, \boldsymbol{\nu}_S^*)) \leftarrow A_{sc-sc}(S, \widetilde{f})$.

---

---

**Algorithm 8** Certified Minimax Unlearning for Convex-Strongly-Concave Loss ($\bar{A}_{c-sc}$)

---

**Input:** Delete requests $U : \{z_j\}_{j=1}^m \subseteq S$, output of $A_{c-sc}(S)$: $(\boldsymbol{\omega}_S^*, \boldsymbol{\nu}_S^*)$, memory variables $T(S)$: $\{\mathtt{D}_{\boldsymbol{\omega\omega}}\widetilde{F}_S(\boldsymbol{\omega}_S^*, \boldsymbol{\nu}_S^*), \mathtt{D}_{\boldsymbol{\nu\nu}}\widetilde{F}_S(\boldsymbol{\omega}_S^*, \boldsymbol{\nu}_S^*)\}$, loss function: $f(\boldsymbol{\omega}, \boldsymbol{\nu}; z)$, regularization parameter: $\lambda$, noise parameters: $\sigma_1, \sigma_2$.
  1: Define

$$\widetilde{f}(\boldsymbol{\omega}, \boldsymbol{\nu}; z) = f(\boldsymbol{\omega}, \boldsymbol{\nu}; z) + \frac{\lambda}{2}\|\boldsymbol{\omega}\|^2. \tag{123}$$

  2: Run the algorithm $\bar{A}_{sc-sc}$ with delete requests $U$, learning variables $(\boldsymbol{\omega}_S^*, \boldsymbol{\nu}_S^*)$, memory variables $T(S)$, loss function $\widetilde{f}$ and noise parameters $\sigma_1$ and $\sigma_2$.
**Output:** $(\boldsymbol{\omega}^u, \boldsymbol{\nu}^u) \leftarrow \bar{A}_{sc-sc}(U, (\boldsymbol{\omega}_S^*, \boldsymbol{\nu}_S^*), T(S), \widetilde{f}, \sigma_1, \sigma_2)$.

---

Note that the function $\widetilde{f}(\boldsymbol{\omega}, \boldsymbol{\nu}; z)$ is $\lambda$-strongly convex in $\boldsymbol{\omega}$, $\mu_{\boldsymbol{\nu}}$-strongly concave in $\boldsymbol{\nu}$, $(\widetilde{L} := 2L + \lambda\|\boldsymbol{\omega}\|)$-Lipschitz, $(\widetilde{\ell} := \sqrt{2}(2\ell + \lambda))$-gradient Lipschitz and $\rho$-Hessian Lipschitz. We also have $\|\mathtt{D}_{\boldsymbol{\omega\omega}}\widetilde{F}_S(\boldsymbol{\omega}_S^*, \boldsymbol{\nu}_S^*)\| \geq \lambda$. Let $(\boldsymbol{\omega}_{S\backslash}^*, \boldsymbol{\nu}_{S\backslash}^*)$ be the optimal solution of the loss function $\widetilde{F}_{S\backslash}(\boldsymbol{\omega}, \boldsymbol{\nu})$ on the remaining dataset, i.e.,

$$(\boldsymbol{\omega}_{S\backslash}^*, \boldsymbol{\nu}_{S\backslash}^*) := \arg\min_{\boldsymbol{\omega} \in \mathcal{W}} \max_{\boldsymbol{\nu} \in \mathcal{V}}\{\widetilde{F}_{S\backslash}(\boldsymbol{\omega}, \boldsymbol{\nu}) := \frac{1}{n-m}\sum_{z_i \in S\backslash U}\widetilde{f}(\boldsymbol{\omega}, \boldsymbol{\nu}; z_i)\}. \tag{124}$$

An application of Lemma 11 implies that the empirical optimizer $(\boldsymbol{\omega}_S^*, \boldsymbol{\nu}_S^*)$ returned by Algorithm 8 satisfies $\|\boldsymbol{\omega}_S^*\| \leq L/\lambda$. Thus our domain of interest are $\mathcal{W} := \{\boldsymbol{\omega}|\|\boldsymbol{\omega}\| \leq L/\lambda\}$. Over the set $\mathcal{W}$, the function $\widetilde{f}(\boldsymbol{\omega}, \boldsymbol{\nu}; z)$ is $3L$-Lipschitz continuous. Suppose the strongly-convex regularization parameter $\lambda$ satisfies $\lambda < \ell$, then $\widetilde{f}(\boldsymbol{\omega}, \boldsymbol{\nu}; z)$ has $3\sqrt{2}\ell$-Lipschitz gradients.

The corresponding theoretical results are given below.

**Lemma 17** (**Closeness Upper Bound**). *Let Assumption 1 hold. Assume the function $f(\boldsymbol{\omega}, \boldsymbol{\nu}; z)$ is $\mu_{\boldsymbol{\nu}}$-strongly concave in $\boldsymbol{\nu}$ and $\|\mathtt{D}_{\boldsymbol{\nu\nu}}\widetilde{F}_S(\boldsymbol{\omega}_S^*, \boldsymbol{\nu}_S^*)\| \geq \mu_{\boldsymbol{\nu\nu}}$. Let $\mu = \min\{\mu_{\boldsymbol{\nu}}, \mu_{\boldsymbol{\nu\nu}}\}$. Then, we have the closeness bound between the intermediate variables $(\widehat{\boldsymbol{\omega}}, \widehat{\boldsymbol{\nu}})$ in Algorithm 8 and $(\boldsymbol{\omega}_{S\backslash}^*, \boldsymbol{\nu}_{S\backslash}^*)$ in eq.(124):*

$$\begin{cases} \|\boldsymbol{\omega}_{S\backslash}^* - \widehat{\boldsymbol{\omega}}\| \leq \left(\frac{8\sqrt{2}\widetilde{L}^2\widetilde{\ell}^3\rho}{\lambda^2\mu^3} + \frac{8\widetilde{L}\widetilde{\ell}^2}{\lambda\mu}\right) \cdot \frac{m^2}{n(\lambda n - (\widetilde{\ell} + \widetilde{\ell}^2/\mu)m)}, \\ \|\boldsymbol{\nu}_{S\backslash}^* - \widehat{\boldsymbol{\nu}}\| \leq \left(\frac{8\sqrt{2}\widetilde{L}^2\widetilde{\ell}^3\rho}{\lambda^3\mu^2} + \frac{8\widetilde{L}\widetilde{\ell}^2}{\lambda\mu}\right) \cdot \frac{m^2}{n(\mu n - (\widetilde{\ell} + \widetilde{\ell}^2/\lambda)m)}. \end{cases} \tag{125}$$

*Proof.* Since we now run the algorithms $A_{sc-sc}$ and $\bar{A}_{sc-sc}$ with the regularized loss function $\widetilde{f}$, the proof is identical to that of Lemma 10. □

**Lemma 18** (**Minimax Unlearning Certification**). *Under the settings of Lemma 17, our minimax learning algorithm $A_{c-sc}$ and unlearning algorithm $\bar{A}_{c-sc}$ is $(\epsilon, \delta)$-certified minimax unlearning if*

*we choose*

$$\begin{cases} \sigma_1 = \big(\frac{8\sqrt{2}\widetilde{L}^2\widetilde{\ell}^3\rho}{\lambda^2\mu^3} + \frac{8\widetilde{L}\widetilde{\ell}^2}{\lambda\mu}\big) \cdot \frac{2m^2\sqrt{2\log(2.5/\delta)}}{n(\lambda n - (\widetilde{\ell} + \widetilde{\ell}^2/\mu)m)\epsilon}, \\ \sigma_2 = \big(\frac{8\sqrt{2}\widetilde{L}^2\widetilde{\ell}^3\rho}{\lambda^3\mu^2} + \frac{8\widetilde{L}\widetilde{\ell}^2}{\lambda\mu}\big) \cdot \frac{2m^2\sqrt{2\log(2.5/\delta)}}{n(\mu n - (\widetilde{\ell} + \widetilde{\ell}^2/\lambda)m)\epsilon}. \end{cases} \tag{126}$$

*Proof.* With the closeness upper bound in Lemma 17 and the given noise scales in eq.(126), the proof is identical to that of Theorem 6. □

**Lemma 19** (**Population Weak PD Risk**). *Under the same settings of Lemma 17, suppose the parameter space $\mathcal{W}$ is bounded so that $\max_{\boldsymbol{\omega} \in \mathcal{W}} \|\boldsymbol{\omega}\| \leq B_{\boldsymbol{\omega}}$, the population weak PD risk for the certified minimax unlearning variables $(\boldsymbol{\omega}^u, \boldsymbol{\nu}^u)$ returned by Algorithm 8 is*

$$\triangle^w(\boldsymbol{\omega}^u, \boldsymbol{\nu}^u) \leq \mathcal{O}\bigg(\big(\frac{L^3\ell^3\rho}{\lambda^3\mu^3} + \frac{L^2\ell^2}{\lambda^2\mu} + \frac{L^2\ell^2}{\lambda\mu^2}\big) \cdot \frac{m^2\sqrt{d\log(1/\delta)}}{n^2\epsilon} + \frac{mL^2}{\lambda n} + \frac{mL^2}{\mu n} + \lambda B_{\boldsymbol{\omega}}^2\bigg), \tag{127}$$

*where $d = \max\{d_1, d_2\}$. In particular, by setting the regularization parameter $\lambda$ as:*

$$\lambda = \max\bigg\{\frac{L}{B_{\boldsymbol{\omega}}}\sqrt{\frac{m}{n}}, \frac{L\ell m}{B_{\boldsymbol{\omega}}\mu n}\big(\frac{\sqrt{d\log(1/\delta)}}{\epsilon}\big)^{1/2}, \big(\frac{L^2\ell^2 m^2\sqrt{d\log(1/\delta)}}{B_{\boldsymbol{\omega}}^2\mu n^2\epsilon}\big)^{1/3}, \\ \big(\frac{L^3\ell^3\rho m^2\sqrt{d\log(1/\delta)}}{B_{\boldsymbol{\omega}}^2\mu^3 n^2\epsilon}\big)^{1/4}\bigg\}, \tag{128}$$

*we have the following population weak PD risk:*

$$\triangle^w(\boldsymbol{\omega}^u, \boldsymbol{\nu}^u) \leq \mathcal{O}\bigg(c_1\sqrt{\frac{m}{n}} + c_2\frac{m}{n} + c_3\big(\frac{\sqrt{d\log(1/\delta)}}{\epsilon}\big)^{1/2}\frac{m}{n} \\ + c_4\big(\frac{\sqrt{d\log(1/\delta)}}{\epsilon}\big)^{1/3}\big(\frac{m}{n}\big)^{2/3} + c_5\big(\frac{\sqrt{d\log(1/\delta)}}{\epsilon}\big)^{1/4}\sqrt{\frac{m}{n}}\bigg), \tag{129}$$

*where $c_1, c_2, c_3, c_4$ and $c_5$ are constants that depend only on $L, l, \rho, \mu$ and $B_{\boldsymbol{\omega}}$.*

*Proof.* An application of [Zhang et al., 2021, Theorem 1] gives that

$$\max_{\boldsymbol{\nu} \in \mathcal{V}} \mathbb{E}[\widetilde{F}(\boldsymbol{\omega}_S^*, \boldsymbol{\nu})] - \min_{\boldsymbol{\omega} \in \mathcal{W}} \mathbb{E}[\widetilde{F}(\boldsymbol{\omega}, \boldsymbol{\nu}_S^*)] \leq \frac{18\sqrt{2}L^2}{n}\big(\frac{1}{\lambda} + \frac{1}{\mu}\big). \tag{130}$$

Using the relation above with an application of eq.(77) and eq.(109), we have

$$\triangle^w(\boldsymbol{\omega}^u, \boldsymbol{\nu}^u) \leq \mathbb{E}[L\|\boldsymbol{\omega}^u - \boldsymbol{\omega}_S^*\|] + \mathbb{E}[L\|\boldsymbol{\nu}^u - \boldsymbol{\nu}_S^*\|] + \frac{18\sqrt{2}L^2}{n}\big(\frac{1}{\lambda} + \frac{1}{\mu}\big) + \frac{\lambda B_{\boldsymbol{\omega}}^2}{2}. \tag{131}$$

By an application of eq.(117), we further get

$$\mathbb{E}[\|\boldsymbol{\omega}^u - \boldsymbol{\omega}_S^*\|] \leq \frac{2mL}{\lambda n - \widetilde{\ell}(1 + \widetilde{\ell}/\mu)m} + \sqrt{d_1}\sigma_1, \tag{132}$$

and

$$\mathbb{E}[\|\boldsymbol{\nu}^u - \boldsymbol{\nu}_S^*\|] \leq \frac{2mL}{\mu n - \widetilde{\ell}(1 + \widetilde{\ell}/\lambda)m} + \sqrt{d_2}\sigma_2. \tag{133}$$

Plugging eq.(132) and eq.(133) into eq.(131) with noise scales given in Lemma 18, we can get our generalization guarantee:

$$\triangle^w(\boldsymbol{\omega}^u, \boldsymbol{\nu}^u) \leq \mathcal{O}\bigg(\big(\frac{L^3\ell^3\rho}{\lambda^3\mu^3} + \frac{L^2\ell^2}{\lambda^2\mu} + \frac{L^2\ell^2}{\lambda\mu^2}\big) \cdot \frac{m^2\sqrt{d\log(1/\delta)}}{n^2\epsilon} + \frac{mL^2}{\lambda n} + \frac{mL^2}{\mu n} + \lambda B_{\boldsymbol{\omega}}^2\bigg), \tag{134}$$

where $d = \max\{d_1, d_2\}$. □

**Lemma 20** (**Population Strong PD Risk**). *Under the same settings of Lemma 19, the population strong PD risk for $(\boldsymbol{\omega}^u, \boldsymbol{\nu}^u)$ returned by Algorithm 8 is*

$$\triangle^s(\boldsymbol{\omega}^u, \boldsymbol{\nu}^u) \leq \mathcal{O}\bigg( \Big( \frac{L^3\ell^3\rho}{\lambda^3\mu^3} + \frac{L^2\ell^2}{\lambda^2\mu} + \frac{L^2\ell^2}{\lambda\mu^2} \Big) \cdot \frac{m^2\sqrt{d\log(1/\delta)}}{n^2\epsilon} + \frac{mL^2}{\lambda n} + \frac{mL^2}{\mu n} + \frac{L^2\ell}{\lambda^{3/2}\mu^{1/2}n} + \frac{L^2\ell}{\lambda^{1/2}\mu^{3/2}n} + \lambda B_{\boldsymbol{\omega}}^2 \bigg), \tag{135}$$

*where $d = \max\{d_1, d_2\}$. In particular, by setting the regularization parameter $\lambda$ as:*

$$\lambda = \max\bigg\{ \frac{L}{B_{\boldsymbol{\omega}}}\sqrt{\frac{m}{n}}, \frac{L\ell m}{B_{\boldsymbol{\omega}}\mu n}\Big( \frac{\sqrt{d\log(1/\delta)}}{\epsilon} \Big)^{1/2}, \Big( \frac{L^2\ell^2 m^2\sqrt{d\log(1/\delta)}}{B_{\boldsymbol{\omega}}^2\mu n^2\epsilon} \Big)^{1/3},$$
$$\Big( \frac{L^3\ell^3\rho m^2\sqrt{d\log(1/\delta)}}{B_{\boldsymbol{\omega}}^2\mu^3 n^2\epsilon} \Big)^{1/4}, \Big( \frac{L^2\ell}{B_{\boldsymbol{\omega}}^2\mu^{1/2}n} \Big)^{2/5}, \frac{1}{\mu}\Big( \frac{L^2\ell}{B_{\boldsymbol{\omega}}^2 n} \Big)^{2/3} \bigg\}. \tag{136}$$

*we have the following population strong PD risk:*

$$\triangle^s(\boldsymbol{\omega}^u, \boldsymbol{\nu}^u) \leq \mathcal{O}\bigg( c_1\sqrt{\frac{m}{n}} + c_2\frac{m}{n} + c_3\Big( \frac{\sqrt{d\log(1/\delta)}}{\epsilon} \Big)^{1/2}\frac{m}{n} + c_4\Big( \frac{\sqrt{d\log(1/\delta)}}{\epsilon} \Big)^{1/3}\Big( \frac{m}{n} \Big)^{2/3}$$
$$+ c_5\Big( \frac{\sqrt{d\log(1/\delta)}}{\epsilon} \Big)^{1/4}\sqrt{\frac{m}{n}} + c_6\frac{1}{n^{2/5}} + c_7\frac{1}{n^{2/3}} \bigg), \tag{137}$$

*where $c_1, c_2, c_3, c_4, c_5, c_6$ and $c_7$ are constants that depend only on $L, l, \rho, \mu$ and $B_{\boldsymbol{\omega}}$.*

*Proof.* An application of Lemma 9 gives that

$$\mathbb{E}[\max_{\boldsymbol{\nu}\in\mathcal{V}} \widetilde{F}(\boldsymbol{\omega}_S^*, \boldsymbol{\nu}) - \min_{\boldsymbol{\omega}\in\mathcal{W}} \widetilde{F}(\boldsymbol{\omega}, \boldsymbol{\nu}_S^*)] \leq \frac{36\sqrt{2}L^2(2\ell+\lambda)}{n}\left( \frac{1}{\lambda^{3/2}\mu^{1/2}} + \frac{1}{\lambda^{1/2}\mu^{3/2}} \right)$$
$$\leq \frac{108\sqrt{2}L^2\ell}{n}\left( \frac{1}{\lambda^{3/2}\mu^{1/2}} + \frac{1}{\lambda^{1/2}\mu^{3/2}} \right). \tag{138}$$

Using an application of eq.(83) and eq.(111), together with eq.(138), eq.(132), eq.(133) and the noise scales given in Lemma 18, we have

$$\triangle^s(\boldsymbol{\omega}^u, \boldsymbol{\nu}^u) \leq \mathcal{O}\bigg( \Big( \frac{L^3\ell^3\rho}{\lambda^3\mu^3} + \frac{L^2\ell^2}{\lambda^2\mu} + \frac{L^2\ell^2}{\lambda\mu^2} \Big) \cdot \frac{m^2\sqrt{d\log(1/\delta)}}{n^2\epsilon} + \frac{mL^2}{\lambda n} + \frac{mL^2}{\mu n} + \frac{L^2\ell}{\lambda^{3/2}\mu^{1/2}n} + \frac{L^2\ell}{\lambda^{1/2}\mu^{3/2}n} + \lambda B_{\boldsymbol{\omega}}^2 \bigg). \tag{139}$$

$\square$

**Lemma 21** (**Deletion Capacity**). *Under the same settings as Lemma 19, the deletion capacity of Algorithm 3 is*

$$m_{\epsilon,\delta,\gamma}^{A,\bar{A}}(d_1, d_2, n) \geq c \cdot \frac{n\sqrt{\epsilon}}{(d\log(1/\delta))^{1/4}}, \tag{140}$$

*where the constant $c$ depends on $L, l, \rho, \mu$ and $B_{\boldsymbol{\omega}}$ and $d = \max\{d_1, d_2\}$.*

*Proof.* By the definition of deletion capacity, in order to ensure the population PD risk derived in Lemma 19 or Lemma 20 is bounded by $\gamma$, it suffices to let $m_{\epsilon,\delta,\gamma}^{A,\bar{A}}(d_1, d_2, n) \geq c \cdot \frac{n\sqrt{\epsilon}}{(d\log(1/\delta))^{1/4}}$. $\square$

