# OpenReview forum: "Certified Minimax Unlearning with Generalization Rates and Deletion Capacity"
_NeurIPS.cc/2023/Conference — NeurIPS 2023 poster_

### Official Review · Reviewer_kPH1 · 2023-06-14

**Soundness:** 4 excellent
**Presentation:** 3 good
**Contribution:** 3 good
**Rating:** 7
**Confidence:** 4

**Summary:**

This paper proposes using hessian-based updates to solve the unlearning problem for strongly-convex-strongly-concave minimax learners. They also provide theoretical analyses of the error of such an unlearning algorithm as well as its sensitivity to deleted data.

**Strengths:**

1. This paper is well-written and structured. The narrative is very easy to follow, and the analysis is strong.
2. The unlearning update is simple.
3. While in a restricted setting, the analysis is coherent.
4. Motivation is obviously, well done, given the connections to privacy.

This paper operates in a restricted setting with some small issues with the presentation. The issues with presentation are fixable, and I expect the authors will go through and better annotate their notation for the camera-ready version. Moreover, the setting is restricted and not practical for unlearning in modern settings. However, this paper achieves its goal and is a good step in the right direction. I, therefore, advocate for this paper's acceptance.

**Weaknesses:**

1. There is a notational mistake for Equation (5). You are missing a closing parenthesis.
2. In equation 6, you should specify that $m$ is the size of $U$.
3. You miswrote Assumption 2 in Lemma 2.
4. I feel some work is needed on the motivation of the setting of this paper. I will discuss more of this in limitations. However, this paper seems only to work when the loss is strongly-convex-strongly-concave loss functions (or suffers bad constants for transforming in the c,c case). However, unlearning is meant to be a more practical tool for modern learning, where this condition rarely holds.




**Questions:**

1. What is $\mathcal{T}$? I understand that it is the set of memory variables, but that should be more formally defined, and I don't find that it is formally defined elsewhere in the paper.
2. It seems that most of the variables in the statement of Lemma 2 are difficult to understand. It is only after going through the proof that I am able to understand. This should be made more clear. For example, what is $\rho$ or $l$ here? This suggestion follows for correcting Theorem 5.
3. I would be interested if the authors of this paper have thought about weakening the strongly-convex-strongly-concave conditions instead assuming gradient-dominance holds (see "Solving Robust MDPs through No Regret Dynamics" or "Global Optimality Guarantees For Policy Gradient Methods" for examples).
4. Have this paper's authors considered including some small toy experiments to verify their theoretical results? It would be interesting to see if they can achieve strong unlearning for linear classifiers with regularization under this unlearning framework.

I also believe this paper is missing a relevant citation: "Accelerated Perceptrons and Beyond" analyzes the generalization/margin of minimax learners in linear or convex-concave settings. This, however, will not impact my score.

**Limitations:**

Yes, they have sufficiently addressed the limitations of their works.

---

> ### Author Rebuttal · Authors · 2023-08-09
>
> We thank the reviewer for his/her careful reading of our manuscript and numerous constructive remarks and questions.
>
> **Notational mistake, size of $U$, and miswrote Assumption 2 in Lemma 2:**
>
> Thank you for your careful reading and pointing out these issues. We will fix them and thoroughly proofread our paper to avoid other potential representational issues.
>
> **Regarding the SC-SC/C-C assumptions:**
>
> Yes, we acknowledge that the SC-SC/C-C assumptions can limit the applicability of our proposed method. However,  we would like to emphasize that existing certified unlearning methods, especially those focused on establishing theoretical results, have largely focused on convex/strongly convex settings. As a nascent research field, we expect to develop advanced certified unlearning methods that rely on relaxed assumptions to be applicable to more general tasks in the future. That being said, it does not diminish the significance of our current work as the first certified minimax unlearning algorithms.
>
>
> **Regarding the set of memory variables $T$:**
>
> The set of memory variables $T$ corresponds to variables that can be computed based on a trained model and memorized to facilitate the unlearning algorithms. When it comes to our certified minimax unlearning algorithm (e.g., Algorithm 2), this set contains the total Hessian matrices with respect to $\omega$ and $\nu$, which are inputs to Algorithm 2. We will formally define the set of memory variables for each algorithm in our paper, as suggested by the reviewer.
>
> **Regarding making variables in Lemma 2 and Theorem 5 more clear:**
>
> Thank you for pointing it out. We will provide all necessary descriptions to clearly introduce the variables that appear in all lemmas and theorems.
>
> **I would be interested if the authors of this paper have thought about weakening the strongly-convex-strongly-concave conditions instead assuming gradient-dominance holds (see "Solving Robust MDPs through No Regret Dynamics" or "Global Optimality Guarantees For Policy Gradient Methods" for examples):**
>
> We appreciate your suggestion and agree that it is a valuable direction for future research. We sincerely thank the reviewer for pointing us in this future direction.
>
> **Regarding inclusion of some small toy experiments:**
>
> We appreciate your suggestion and agree that including toy experiments could strengthen our paper. We will consider machine unlearning under the pre-training and fine-tuning setting described in [1]. In this setting, all network parameters are fixed except for the loss layer, which will be fine-tuned on a dataset. During the fine-tuning stage, we effectively consider a SC-SC/C-C model. We will test the performance of our proposed methods with deletion requests made only from this dataset.
>
> [1] Golatkar, Aditya, et al. "Mixed-privacy forgetting in deep networks." CVPR'21.
>
> **Missing a relevant citation:**
>
> Thank you for pointing us to this missing reference. We will introduce and cite it in our paper.

---

> > ### Comment · Reviewer_kPH1 · 2023-08-10
> > **Thank you for your rebuttal**
> >
> > I appreciate the author's rebuttal. I maintain my score and vote to accept.

---

### Official Review · Reviewer_fuZ1 · 2023-06-28

**Soundness:** 3 good
**Presentation:** 4 excellent
**Contribution:** 3 good
**Rating:** 6
**Confidence:** 4

**Summary:**

Machine unlearning is a privacy-inspired area to remove certain training samples of users’ data from well-trained model making those samples uninfluential while the unlearning approach does not cause the model to be retrained from scratch (not cause comprehensive computational cost) to achieve the baseline relied on the rest of datasets. The conventional machine unlearning usually applies for standard statistical learning models with one variable, but there is limited work for minimax models. Borrowing from Gaussian mechanism of differential privacy, this work also focuses on epsilon-delta certification of machine unlearning. Moreover, generalization rates and deletion capacity (number of deleted samples) are also very crucial to study in this framework.

**Strengths:**

1. Machine unlearning is a quite new field for privacy of machine learning. This work smoothly introduces the field with related work (a lot of citations) and preliminaries. Also it well states the reason minimax model is important for machine unlearning.
2. Certified minimax unlearning can be well extended to more general loss functions with reasonable generalization rates and deletion capacity results.
3. This work supports successive and online deletion requests, which is computationally efficient for unlearning phase with its minimax unlearning update.

**Weaknesses:**

1. Although this work focuses more on theoretical part of unlearning, it is still better to provide some preliminary results on some datasets. At least in one paragraph of intro or conclusion, authors can discuss scenarios or applications how to make the certified minimax unlearning practical.
2. This work does not discuss the relationships of certified unlearning between STL model and minimax model theoretically (How are they different).

**Questions:**

1. Compared to other approaches for privacy of machine learning like differential privacy, what is the advantage of machine unlearning for the privacy? Since this work focuses on minimax model, the question could narrow down the specific model.
2. On Line 98, is it intentionally “extract” or should it be “exact”?

**Limitations:**

1. As weakness 1, authors can consider some implications for the applications by adding a paragraph. Like in the beginning of the introduction, how does this work help GAN, ARL or RL?
2. As weakness 2, this work emphasizes how important the certified unlearning for minimax model, but this work does not talk too much about the connection to unlearning for STL models.
3. Instead of putting every proof in the appendix, it is better to add one or two sentences for each theorem or lemma about the clue to prove (what methods will you use? How will you prove?) to convince readers who are interested in the topic but not familiar.

---

> ### Author Rebuttal · Authors · 2023-08-09
>
> We thank the reviewer for his/her careful reading of our manuscript and numerous constructive remarks and questions.
>
> **Regarding 1) providing some preliminary results on some datasets and 2) discussing scenarios or applications:**
>
> Thank you for the valuable suggestions. For 1), we will consider the machine unlearning under the pre-training and fine-tuning setting described in [1]. In this setting, all network parameters are fixed except for the loss layer, which will be fine-tuned on a dataset. During the fine-tuning stage, we effectively consider a SC-SC/C-C model. We will test the performance of our proposed methods with deletion requests made only from this dataset. For 2), we will add discussions about the applications (e.g., machine unlearning for GAN, fairness learning, RL, adversarial training) in the introduction section.
>
> [1] Golatkar, Aditya, et al. "Mixed-privacy forgetting in deep networks." CVPR'21.
>
> **Regarding differences of certified unlearning between SLT model and minimax model theoretically:**
>
> Thank you for your valuable suggestion and constructive comments. Due to the character limit, we provide detailed discussions of their differences in General Response. We will revise the introduction section and Sec 4.1 by adding these discussions to clearly convey the differences and challenges, as suggested by the reviewer.
>
> **Regarding advantage over differential privacy:**
>
> The main advantage of certified unlearning over differential privacy is its better deletion capacity. This has two interpretations: First, for a given level of generalization performance, certified unlearning supports a larger number of data deletions while maintaining that generalization performance. Second, for a given number of deletions, certified unlearning yields better generalization performance than differential privacy. The intuition behind this is that certified unlearning has smaller sensitivity than differential privacy, and therefore requires less random perturbation to maintain utility.
>
> **Consider some implications for the applications by adding a paragraph. Like in the beginning of the introduction, how does this work help GAN, ARL or RL:**
>
> Thank you for your constructive comment. We will add a paragraph to the introduction section discussing how our certified minimax unlearning algorithm can be applied to existing minimax models to facilitate effective unlearning with certified unlearning endurance, along with generalization and performance guarantees.
>
> **Regarding adding one or two sentences for each theorem or lemma about the clue to prove:**
>
> Thank you for the valuable suggestion. We will provide the essential ideas and clues used to derive our main results including Lemma 1 to 3 and Theorem 2 to 4.
>
> **Miswrote "exact" on Line 98:**
>
> Thank you for your careful reading and pointing out the miswriting. We will fix it and thoroughly proofread our paper to address other potential representational issues.

---

> > ### Comment · Reviewer_fuZ1 · 2023-08-13
> > **Thanks for your rebuttal**
> >
> > Authors answered all my concerns and I agreed with those explanations. Thank you!

---

### Official Review · Reviewer_sP3P · 2023-07-05

**Soundness:** 3 good
**Presentation:** 2 fair
**Contribution:** 2 fair
**Rating:** 6
**Confidence:** 4

**Summary:**

The papers proposes a Newton-based differentially private algorithm for stochastic minimax problems and analyse the generalization rate and deletion capacity for the algorithms.
They analyse the generalization bound for weak primal-dual risk in SC-SC, C-SC and SC-C cases. Also the deletion capacity they derive is $O(n/d^{1/4})$, which is better than the baseline result $O(n/d^{1/2})$. The results of generalization bound and deletion capacity match the bound in the pure minimazation case.

**Strengths:**

The paper is the first one analysing certified machine unlearning for minimax problems and provide certain generalization and deletion capacity guarantee.

**Weaknesses:**

There are several weaknesses and issues:
1. The reviewer is not convinced that weak primal-dual risk is enough to capture the behavior of generalization for minimax problems. In recent literature, people begin to study the strong primal-dual risk for minimax problems even without strong convexity.
For example, the authors in ``What is a Good Metric to Study Generalization of Minimax Learners'' derive the result for generalization of strong primal dual  risk in convex-concave setting.
Therefore, the reviewer thinks the paper can be improved if the authors can extend their result to strong PD risk.
2. Suggestions for writing: The certified machine unlearning and deletion capacity might not be well-known to general audiums. However, in the abstract and begining of the paper, the authors use too many specific terminology for machine unlearning, deletion, etc. It is not friendly to the audiums. The reviewer suggests the authors can discuss more motivation for this paper and give more explanations for the intuition of these terminologies.
3. Missing reference: ''Uniform convergence and generalization for nonconvex stochastic minimax problems'',   ''What is a Good Metric to Study Generalization of Minimax Learners''.

**Questions:**

None

---

> ### Author Rebuttal · Authors · 2023-08-09
>
> We thank the reviewer for his/her careful reading of our manuscript and numerous constructive remarks and questions.
>
> **Regarding strong primal-dial risk vs weak primal-dual risk:**
>
> Thank you for the valuable suggestion. We have derived new population performance results in terms of the strong primal-dual risk for our minimax unlearning algorithm, as can be found in General Response and the attached PDF.
>
> **Regarding specific terminology for machine unlearning and deletion in the abstract and beginning of the paper:**
>
> Thank you for your valuable suggestions. We will revise the introduction section to include the following content.
>
> 1) Introduce unlearning-related terminologies and their intuition: We will provide clear definitions and explanations of key unlearning-related terms and concepts to help readers better understand the motivation and significance of our work.
>
> 2) Motivation of our algorithm: We will discuss the challenges and limitations of existing unlearning methods and explain how our algorithm and analysis address these challenges to provide effective certified minimax unlearning algorithms with generalization and deletion capacity guarantees.
>
> 3) Meaning and significance of our results: We will provide a clear and concise summary of our main results and their implications, as well as meanings to potential applications.
>
> **Missing references:**
>
> Thank you for pointing us to these references. We will introduce and cite them in our paper.

---

> > ### Comment · Reviewer_sP3P · 2023-08-18
> >
> > I appreciate the authors for addressing most of my inquiries and providing a generalization bound for strong PD-risk in the strongly-convex case. I'd like to update my rating to 6. However, this paper also examines the convex-concave scenario, lacking a generalization bound for strong PD risk in that case. The authors' response only furnishes a proof for the strongly convex case. In the paper "What is a Good Metric...," the authors present a generalization bound for strong PD risk in the convex-concave situation. I'm curious whether the findings from this paper can be extended to the convex-concave case.

---

> > > ### Author Response · Authors · 2023-08-19
> > > **Generalization bound for strong PD risk in the convex-concave situation**
> > >
> > > Thank you very much for engaging with our rebuttal and raising the initial score. We have also extended the  strong primal-dual risk analysis to the general convex concave case, which was not presented in the previous rebuttal due to limited space. In this follow-up response, we provide this result and its sketched proof.
> > >
> > > With an application of Lemma 23 in the attached pdf, we first extend Lemma 10 in terms of population strong primal-dual risk: $\mathbb E[\max_{\nu\in\mathcal V}F(\omega^\*\_S,\nu)-\min_{\omega\in\mathcal W}F(\omega,\nu^\*\_S)] \leq \frac{128\sqrt 2 L^2(2\ell+\lambda)}{\lambda^2 n}+\frac{\lambda}{2}(B_\omega^2+B_\nu^2)$. Then, together with eq.(133) we have that $\triangle^s(\omega^u,\nu^u)
> > > \leq \mathbb E[L\\|\omega^u-\omega^\*\_S\\|]+\mathbb E[L\\|\nu^u-\nu^\*\_S\\|]+\frac{128\sqrt 2 L^2(2\ell+\lambda)}{\lambda^2 n}+\frac{\lambda}{2}(B_\omega^2+B_\nu^2)$. Plugging eq.(96) and eq.(97) into the equation above with noise scales given in Lemma 12, we can get the generalization guarantee for population strong PD risk in the convex-concave situation: $\triangle^s(\omega^u,\nu^u) \leq O \bigg( (L^3 \ell^3 \rho / \lambda^6 + L^2 \ell^2/\lambda^3)\cdot \frac{m^2 \sqrt{d\log(1/\delta)}}{n^2 \epsilon} + \frac{mL^2}{\lambda n} + \frac{L^2\ell}{\lambda^2 n} + \lambda(B_{\omega}^2 + B_{\nu}^2) \bigg)$. In order to ensure the population strong PD risk is bounded by $\gamma$, it suffices to let the deletion capacity $m ^ { A,\bar A} _ {\epsilon,\delta,\gamma}(d_1,d_2,n) \geq c \cdot \frac{n\sqrt{\epsilon}}{(d\log(1/\delta))^{1/4}}$.
> > >
> > > Thank you again and we will add these new results in the final version of our paper.

---

### Official Review · Reviewer_R8EP · 2023-07-05

**Soundness:** 3 good
**Presentation:** 3 good
**Contribution:** 3 good
**Rating:** 6
**Confidence:** 3

**Summary:**

This paper studies the problem of machine learning for the minimax model. By using Newton's step update with the Hessian information of the leftover data and Gaussian Mechanism, the proposed method improves the deletion capacity from $O(n/d^{1/2})$ to $O(n/d^{1/4})$.

**Strengths:**

- Given the rise of GAN and the increasing concern for the privacy of participants in datasets used by GAN/ML models, minimax unlearning is an important problem to study and understand.

- The proposed algorithm shows improvement in both strongly convex-strongly convex and the more general convex-concave settings.

- The improvement of the deletion capacity from $O(n/d^{1/2})$ to $O(n/d^{1/4})$ could be quite significant for practical problems with high-dimensional features.

- The efficient algorithm that dispenses the need for recomputing some of the Hessian information is a nice touch since computing the Hessian matrix can be very computationally expensive.

- The paper is well-written overall.

**Weaknesses:**

- Some of the technical details are used before being defined properly. For example, in equation (7), $\kappa$ is defined as $\kappa = l/\mu$ even though $l$ and $\mu$ haven't been defined yet. These quantities are later mentioned in section 4 but it would be better for the readers if those quantities are defined right away.



**Questions:**

- I'm a bit confused about the intuition part where $\nu^\star_{S\}- \nu^\star  \approx -\partial_{\nu\nu}^{-1}F_{S\}(w_S^\star,\nu_S^\star)\dot\partial_{\nu w}^{-1}F_{S\}(w_S^\star,\nu_S^\star)(w^\star_{S\}- w^\star)$.  Can the author explain how we can use linear approximation and the implicit function theorem to get this result? Also why do we want to leave out the second term in eq. 60?

- Is there any benefit in doing the update in Algorithm 2 compared to Algorithm 3? As the authors have shown in the proofs of Lemma 1 and Lemma 19, the analyses are basically identical so I'm just curious if there's any good thing about the update of Algorithm 1 and why we want to do it in the first place?

- Seems like there's no improvement in the result in SC-SC case compared to C-C case. Is this the artifact of the algorithm or is this something that usually happens in minimax learning?

---

> ### Author Rebuttal · Authors · 2023-08-09
>
> We thank the reviewer for his/her careful reading of our manuscript and numerous constructive remarks and questions.
>
> **Some of the technical details are used before being defined properly:**
>
> Thank you for your valuable suggestion. We will move Assumptions 1 and 2 from Sec 4 to Sec 3.3.
> Additionally, we will carefully check all definitions of terms in the paper to ensure that they are defined before being used.
>
> **Regarding the intuition part in Sec 4.1:**
>
> We apologize for causing the confusion. We will provide more detailed explanations of the intuition and the derivation of Fact 1 in our paper, as follows.
>
> For the linear approximation and the implicit function theorem to get $\nu^\*\_{S^{\setminus}} - \nu\_{S}^\* \approx -\partial^{-1}\_{\nu\nu} F\_{S^{\setminus}}(\omega\_S^\*,\nu\_S^\*) \partial\_{\nu\omega} F\_{S^{\setminus}}(\omega\_S^\*,\nu\_S^\*) (\omega\_{S^{\setminus}}^\* - \omega\_{S}^\*)$ is as follows. Following Eq.(61), we have $\nu^\*\_{S^{\setminus}} - \nu^\*\_{S} \approx V\_{S^{\setminus}}(\omega^\*\_{S^{\setminus}}) - V\_{S^{\setminus}}(\omega ^\*\_{S}) \approx \Big{(}\frac{d V\_{S^{\setminus}}(\omega^\*\_{S})}{d \omega} \Big{|}\_{\omega = \omega^\*\_{S}}\Big{)}\cdot(\omega^\*\_{S^{\setminus}} - \omega^\*\_{S})$, where the second ''$\approx$'' is the linear approximation step and is the response Jacobian of the auxiliary function $V$ defined in Definition 8. Next, by implicit function theorem, we further have $\Big{(}\frac{d V\_{S^{\setminus}}(\omega^\*\_{S})}{d \omega} \Big{|}\_{\omega = \omega^\*\_{S}}\Big{)} = -\partial^{-1}\_{\nu\nu} F\_{S^{\setminus}}(\omega\_S^\*,\nu\_S^\*) \partial\_{\nu\omega} F\_{S^{\setminus}}(\omega\_S^\*,\nu\_S^\*) $, which leads to the result. We emphasize that this only provides the intuition to introduce the total Hessian (rather than the conventional Hessian) in the minimax unlearning, which is not meant to be rigorous. We rigorously derive the $(\epsilon,\delta)$-certified minimax unlearning algorithm based on the one-step complete Newton update and Gaussian mechanism. To do so, it requires to bound the closeness upper bound in Lemma 1, which is more involved than the certified machine unlearning for standard learning models.
>
> For the purpose of leaving out $V\_{S^{\setminus}}(\omega^\*\_{S})-\nu\_S^\*$: This is based on the observation that we already have the direct and partial Hessian of $F\_{S^{\setminus}}$ in Eq.(10), both are for the loss $F\_{S^{\setminus}}$. Thus, when dealing with the remaining term $\nu^\*\_{S^{\setminus}} - \nu\_{S}^\*$, we want to convert it into a form having strong correlations with $F_{S^{\setminus}}$ (rather than $F_{S}$). First, according to Definition 8, we notice that  $\nu^\*\_{S^{\setminus}} - \nu\_{S}^\*$ can be equivalently represented as $V\_{S^{\setminus}}(\omega^\*\_{S^{\setminus}}) - V\_{S}(\omega^\*\_{S})$, where the term $V\_{S}(\omega^\*\_{S})$ does not related to $F_{S^{\setminus}}$. We further convert it to $[V\_{S^{\setminus}}(\omega^\*\_{S^{\setminus}}) -V\_{S^{\setminus}}(\omega^\*\_{S})] + [V\_{S^{\setminus}}(\omega^\*\_{S}) - V\_{S}(\omega^\*\_{S})]$. Then, the first square bracket belongs to the total Hessian of $F_{S^{\setminus}}$ after conversions, while the second bracket still has $V\_{S}(\omega^\*\_{S})$ that does not have relationship with $F_{S^{\setminus}}$ (instead it is related to $F_{S}$). We have shown that the second bracket can be bounded in Lemma 2 and the increased approximation does not affect the overall certified unlearning guarantee and the order of generalization and deletion capacity. As a result, this term can be safely left out.
>
> **Why we want Algorithm 2:**
>
> We choose to present Algorithm 2 in detail first in the paper mainly due to the following two considerations.
> First, the minimax unlearning update in Algorithm 2 is a direct consequence of the intuition in Sec 4.1 to introduce the total Hessian of $F_{S^{\setminus}}(\omega,\nu)$ (rather than the conventional Hessian in prior SLT unlearning updates). Thus, Algorithm 2 has a closer relation to the intuition than Algorithm 3, which uses the total Hessian of $F_{S}(\omega,\nu)$ and does not need to update the total Hessian. This cannot be directly derived from the intuition and it is only after formally bounding the closeness upper bound and deriving the generalization and deletion capacity results that we can ensure that replacing $F_{S^{\setminus}}(\omega,\nu)$ with $F_{S}(\omega,\nu)$ still have the same utility but better efficiency.
>
> Second, sequentially introducing Algorithms 2 and 3 is aligned with the chronological development of certified unlearning methods in SLT, which facilitates direct comparison between certified unlearning for minimax models and SLT models. That is, early works on certified unlearning for SLT proposed to use the conventional Hessian of $F_{S^{\setminus}}(\omega)$ first, while later works discovered that replacing it with the conventional Hessian of $F_{S}(\omega)$ can still yield certified unlearning guarantee as well as generalization and deletion capacity guarantees. Thus, we chose to present Algorithms 2 and 3 according to this chronological development so that both minimax unlearning updates can be directly compared with their counterparts in SLT unlearning updates.
>
> **Regarding improvement in results in SC-SC case compared to C-C case:**
>
> The generalization performance of the C-C case is worse than the SC-SC case, where the former is $O(\sqrt{\frac{m}{n}} + (\frac{m}{n})^{2/7})$ while the latter is $O(\frac{1}{n} + \frac{m^2}{n^2})$. In terms of the deletion capacity, the two cases indeed have the same order of deletion capacity $\frac{n}{d^{1/4}}$, which indicates that both cases can support the same amount of samples to be deleted. This order of deletion capacity matches that of SLT unlearning, which can be regarded as a special case of minimax unlearning. Therefore, it does not seem to be an artifact of our certified minimax unlearning algorithm.

---

> > ### Comment · Reviewer_R8EP · 2023-08-19
> >
> > Thanks for the response! I will keep my score.

---

### Official Review · Reviewer_DtDD · 2023-07-10

**Soundness:** 2 fair
**Presentation:** 3 good
**Contribution:** 2 fair
**Rating:** 6
**Confidence:** 4

**Summary:**

The paper studies approximate unlearning for minimax problems. They design learning and unlearning procedures and provide bounds on deletion capacity in terms of generalization performance (weak gap). Akin to minimization (statistical learning), the deletion capacity for strongly convex-strongly concave setting, is shown to be, $n/d^{1/4}$, where $n$ is the number of samples and $d$, dimension. The authors also provide extensions to non strongly convex/concave settings and efficient updates.

**Strengths:**

The problem of machine unlearning has gathered a lot of interest recently, owing to various privacy regulations. Further, the minimax formulation, is widely applicable, especially in robust adversarial learning and reinforcement learning. The paper is the first to study unlearning for minimax settings. Therefore, the topic of the work is natural and timely.

**Weaknesses:**

1. The paper very closely follows the outline and techniques in prior work of Sekhari et. al. The extensions, to non-strongly convex settings (via regularization) and efficient updates also use techniques directly from prior work. If there are additional challenges due to the minimax (as opposed to min) structure, then I don't think they are communicated well in the write-up. The only relevant section is Section 4.1 "Intuition for Minimax Unlearning Update", however, I found it too raw to convey the intuition -- for instance, how does Eqn. 10 follow? In the current state of the write-up, it is difficult to evaluate if there are significant challenges overcome in relation to prior works.

2. Comparison between Section 4.3 and 5.2: It seems to me that both are in the same setting, and achieve the same guarantees, yet Section 5.2 provides a more efficient update. If indeed Section 5.2 is a strict improvement over 4.3, then what is the point of devoting considerable space to the weaker result in Section 4.3.  The authors should re-organize and present the strongest result in the main paper. The space should be used to explain the challenges compared to the minimization setting. If this is not the case, then please explain the differences.

3. Strong gap vs weak gap: The generalization performance considered in the paper is the weak primal-dual gap. In the non-private setting, "strong" primal-dual gap (as opposed to weak) is what is typically considered. Further, as explained in Bassily et al. 2023, the strong gap criterion has game-theoretic interpretation and motivation, and moreover, the weak and strong gaps may be arbitrarily apart. Seemingly, the consideration of the weak gap in the paper primarily stems from challenges of studying strong gap under privacy settings, until recently. However, given that the work Bassily et al. 2023 has established optimal rates for strong gap under privacy, can the authors, perhaps borrowing techniques from the aforementioned work, also provide guarantees in terms of the strong gap?

**Questions:**

1. Please answer the question posed in weakness 1.
2. Please answer the question posed in weakness 2.
3. Please answer the question posed in weakness 3.

**Limitations:**

1 The work is limited to the approximate unlearning setting, as opposed to exact unlearning.
2. The authors study generalization performance in terms of weak gap as opposed to strong gap.

---

> ### Author Rebuttal · Authors · 2023-08-09
>
> We thank the reviewer for his/her careful reading of our manuscript and numerous constructive remarks and questions.
>
> **Regarding additional challenges due to the minimax structure and comparison to prior works:**
>
> Thank you for your valuable suggestion and constructive comments. Due to the character limit, we provide explanations of additional challenges due to the minimax structure in General Response. We will revise Sec 4.1 by adding these explanations to clearly convey the challenges, as suggested by the reviewer.
>
> **Regarding Equation (10):**
>
> The left hand of Equation (10) contains two terms. The first term is the direct Hessian (also known as the conventional Hessian) of $F_{S^{\setminus}}(\omega,\nu)$, which has already appeared in the unlearning update for SLT models. The second term is unique to the minimax unlearning, which captures the inter-influence between the min and max variables and leads to the total Hessian to be used in the minimax unlearning update. In detail, the second term spells out as follows. To get
> $\nu^\*\_{S^{\setminus}} - \nu_{S}^\* \approx -\partial^{-1}\_{\nu\nu} F_{S^{\setminus}}(\omega_S^\*,\nu_S^\*) \partial_{\nu\omega} F_{S^{\setminus}}(\omega_S^\*,\nu_S^\*) (\omega_{S^{\setminus}}^\* - \omega_{S}^\*)$, we begin with $\nu^\*\_{S^{\setminus}} - \nu^\*\_{S} \approx V\_{S^{\setminus}}(\omega^\*\_{S^{\setminus}}) - V\_{S^{\setminus}}(\omega ^\*\_{S}) \approx \Big{(}\frac{d V\_{S^{\setminus}}(\omega^\*\_{S})}{d \omega} \Big{|}\_{\omega = \omega^\*\_{S}}\Big{)}\cdot(\omega^\*\_{S^{\setminus}} - \omega^\*\_{S})$, where the second ''$\approx$'' is the linear approximation step and is the response Jacobian of the auxiliary function $V$ defined in Definition 8. Next, by implicit function theorem, we further have $\Big{(}\frac{d V\_{S^{\setminus}}(\omega^\*\_{S})}{d \omega} \Big{|}\_{\omega = \omega^\*\_{S}}\Big{)} = -\partial^{-1}\_{\nu\nu} F\_{S^{\setminus}}(\omega\_S^\*,\nu\_S^\*) \partial\_{\nu\omega} F\_{S^{\setminus}}(\omega_S^\*,\nu_S^\*) $
> , which leads to the result. We emphasize that this only provides the intuition to introduce the total Hessian (rather than the conventional Hessian) in the minimax unlearning, which is not meant to be rigorous. We rigorously derive the $(\epsilon,\delta)$-certified minimax unlearning algorithm based on the one-step complete Newton update and Gaussian mechanism.
>
> We will move the above derivation from Appendix to Sec 4.1, as suggested by the reviewer.
>
> **Regarding the comparison between Section 4.3 and 5.2 (Algorithm 2 and Algorithm 3):**
>
> We thank the reviewer for the valuable suggestion. Algorithm 3 in Sec 5.2 is indeed a strict improvement over Algorithm 2 in Sec 4.3, with better efficiency and similar guarantees in terms of generalization and deletion capacity. We choose to present Algorithm 2 in detail first in the paper mainly due to the following two considerations.
> First, the minimax unlearning update in Algorithm 2 is a direct consequence of the intuition in Sec 4.1 to introduce the total Hessian of $F_{S^{\setminus}}(\omega,\nu)$ (rather than the conventional Hessian in prior SLT unlearning updates). Thus, Algorithm 2 has a closer relation to the intuition than Algorithm 3, which uses the total Hessian of $F_{S}(\omega,\nu)$ and does not need to update the total Hessian. This cannot be directly derived from the intuition and it is only after formally bounding the closeness upper bound and deriving the generalization and deletion capacity results that we can ensure that replacing $F_{S^{\setminus}}(\omega,\nu)$ with $F_{S}(\omega,\nu)$ still have the same utility but better efficiency.
>
> Second, sequentially introducing Algorithms 2 and 3 is aligned with the chronological development of certified unlearning methods in SLT, which facilitates direct comparison between certified unlearning for minimax models and SLT models. That is, early works on certified unlearning for SLT proposed to use the conventional Hessian of $F_{S^{\setminus}}(\omega)$ first, while later works discovered that replacing it with the conventional Hessian of $F_{S}(\omega)$ can still yield certified unlearning guarantee as well as generalization and deletion capacity guarantees. Thus, we chose to present Algorithms 2 and 3 according to this chronological development so that both minimax unlearning updates can be directly compared with their counterparts in SLT unlearning updates.
>
> We will revise our manuscript according to the reviewer's constructive comments. Specifically, we will: 1) relegate the detailed analysis for Algorithm 2 in Sec 4.3 to the appendix; 2) Move Algorithm 3 and its analysis from Sec 5.2 to Sec 4.3; 3) Highlight the additional challenges between certified minimax unlearning over SLT unlearning in Sec. 4.1; 4) Add discussions in Sec 4.2 to compare the differences between minimax unlearning and SLT unlearning and provide discussions to motivate the changes made to Algorithm 2 to yield Algorithm 3.
>
>
> **Regarding Strong gap vs weak gap:**
>
> Thank you for the valuable suggestion. We have derived new population performance results in terms of the strong primal-dual risk for our minimax unlearning algorithm, as can be found in General Response and the attached PDF.
>
> **Regarding limited to the approximate unlearning setting, as opposed to exact unlearning:**
>
> In this paper, we focus on the approximate unlearning that comes with $(\epsilon,\delta)$-certified unlearning guarantee, generalization, and deletion capacity guarantees. We will add discussions in the future work part about the potential for developing minimax unlearning methods under the exact unlearning setting.

---

> > ### Comment · Reviewer_DtDD · 2023-08-21
> > **Thanks!**
> >
> > I thank the authors for their detailed response, and in particular their derivation of guarantee on strong gap. I encourage the authors to include this, and the other parts of their response, to the revised paper. I increase my score to 6.

---

### Author Rebuttal · Authors · 2023-08-09

In this general response, we would like to first thank all reviewers for their careful review and valuable comments. Next, we provide the responses to two common comments that are shared by at least two reviewers. We provide the remaining point-to-point responses to each reviewer in our individual responses.

**Regarding the comparison of certified unlearning between minimax models and SLT, and additional challenges overcome in relation to prior works (Reviewers DtDD and fuZ1):**

Comparing the certified unlearning between minimax models and SLT, there are three main differences:

1) Different designs of the unlearning update: minimax unlearning introduces the total Hessian (i.e., direct Hessian plus indirect Hessian) to sufficiently capture the data influence for unlearning, which is tailor-made to the minimax structure that has the inter-influencing min and max variables. In contrast, SLT unlearning requires only the direct Hessian (also known as the conventional Hessian).
2) Different calibrations of the random perturbation: In order to properly calibrate the random perturbation to guarantee $(\epsilon,\delta)$-certified unlearning, it is essential to analyze the closeness upper bound (e.g., Lemma 1), which signifies the magnitude of the randomized perturbation. Three factors make the analysis of the closeness upper bounds very different between minimax unlearning and SLT unlearning: i) extra approximation terms are introduced during the derivation of the minimax unlearning update formula; ii) the minimax unlearning update utilizes the total Hessian-based unlearning update rather than the conventional Hessian; iii) the auxiliary functions (as defined in Definition 8) that arise due to the minimax structure appear in the analysis.
3) Different generalization performance metrics: SLT unlearning uses the excess population risk to measure generalization performance. In our minimax unlearning, we provide weak primal-dual risk in our paper. Additionally, we derive new generalization results in terms of the strong primal-dual risk during the rebuttal period.

Due to these differences, certified minimax unlearning presents additional challenges:
1) it is nontrivial to design the certified minimax unlearning update scheme, which requires a series of conversions and proper approximations to come up with a closed-form update in the form of the total Hessian-based complete Newton step. This challenge corresponds to Equations (9) to (11) in the paper and the development of Fact 1 in Appendix A.2.

2) it is challenging to derive the closeness upper bound to guarantee certified minimax unlearning, which requires dealing with i) the extra approximation terms left out during the minimax unlearning update design, ii) the total Hessian that has different characteristics compared to the conventional Hessian, and iii) best response auxiliary functions that are unique to the minimax structure and do not appear in STL unlearning.

3) it was previously unknown whether it was possible to achieve ideal generalization performance and deletion capacity for certified minimax unlearning. Our work is the first to show that the rates of generalization and deletion capacity match the state-of-the-art rates derived previously for SLT unlearning, which are special cases of minimax unlearning. This indicates that certified minimax unlearning can indeed achieve ideal generalization performance and deletion capacity.

**Regarding strong primal-dual risk (Reviewers DtDD and sP3P):**

Thank you for your valuable suggestion. During the rebuttal period, we have derived new generalization performance results for our certified minimax unlearning algorithms in terms of strong primal-dual risk. In the general response below, we summarize our main results derived. In the attached PDF file, we provide more detailed results, along with a sketch of the proof highlighting the main differences in derivation compared to the weak primal-dual risk counterpart. Due to space limits, we provide the new results for Algorithm 2 as an example in this response. We will provide the counterpart results for all proposed algorithms to our paper.
Denote $d=\max\\{d_1,d_2\\}$, under the same settings of Lemma 1, the population strong primal-dual risk for the certified minimax unlearning variables $(\omega^u,\nu^u)$ returned by Algorithm 2 is $ \triangle^s (\omega^u,\nu^u) = O \left( (L^3 \ell^3 \rho / \mu^6 + L^2 \ell^2/\mu^3)\cdot \frac{m^2 \sqrt{d\log(1/\delta)}}{n^2 \epsilon} + \frac{mL^2}{\mu n} + \frac{L^2\ell}{\mu^2 n}\right)$. Define the deletion capacity $m^{A,\bar A}_{\epsilon,\delta,\gamma}(d_1,d_2,n)$ as the maximum number of samples $U$ that can be unlearned while still ensuring the population strong primal-dual risk is at most $\gamma$.Then it suffices to let $m ^ { A,\bar A} _ {\epsilon,\delta,\gamma}(d_1,d_2,n) \geq c \cdot \frac{n\sqrt{\epsilon}}{(d\log(1/\delta))^{1/4}}$, where the constant $c$ depends on $L, l, \rho,$ and $\mu$ of the loss function $f$.

---

### Decision · Program_Chairs · 2023-09-21

**Decision:**

Accept (poster)

**Comment:**

The paper studies the problem of unlearning in the (epsilon, delta)-unlearning model. It provides a new algorithm and shows that with n training samples and models with d parameters, one can unlearn up to O(n/d^{1/4}) samples, which is strictly better than the previous known result of O(n/sqrt{d}). Unlearning has attracted a lot of interest in recent years and the reviewers agree that the paper makes a nice contribution to the field. I would appreciate it if authors could incorporate reviewer suggestions in the camera-ready version.